# Spatially resolved phosphoproteomics reveals fibroblast growth factor receptor recycling-driven regulation of autophagy and survival

Joanne Watson [1,2,7], Harriet R. Ferguson [2,7], Rosie M. Brady[3], Jennifer Ferguson[2], Paul Fullwood[2], Hanyi Mo[1], Katherine H. Bexley [2], David Knight[4], Gareth Howell[5], Jean-Marc Schwartz [1], Michael P. Smith [2] ✉ & Chiara Francavilla [2,6] ✉

Receptor Tyrosine Kinase (RTK) endocytosis-dependent signalling drives cell proliferation and motility during development and adult homeostasis, but is dysregulated in diseases, including cancer. The recruitment of RTK signalling partners during endocytosis, specifically during recycling to the plasma membrane, is still unknown. Focusing on Fibroblast Growth Factor Receptor 2b (FGFR2b) recycling, we reveal FGFR signalling partners proximal to recycling endosomes by developing a Spatially Resolved Phosphoproteomics (SRP) approach based on APEX2-driven biotinylation followed by phosphorylated peptides enrichment. Combining this with traditional phosphoproteomics, bioinformatics, and targeted assays, we uncover that FGFR2b stimulated by its recycling ligand FGF10 activates mTOR-dependent signalling and ULK1 at the recycling endosomes, leading to autophagy suppression and cell survival. This adds to the growing importance of RTK recycling in orchestrating cell fate and suggests a therapeutically targetable vulnerability in ligand-responsive cancer cells. Integrating SRP with other systems biology approaches provides a powerful tool to spatially resolve cellular signalling.

Endocytosis is the process by which surface molecules, including Receptor Tyrosine Kinases (RTKs), undergo internalisation from the plasma membrane into the early endosome within seconds of ligand binding, followed by direct recycling to the plasma membrane, sorting to the lysosome via the late endosome for degradation, or sorting to the recycling endosomes for recycling to the plasma membrane[1–3]. In addition to controlling receptor availability at the cell surface[4], recycling is critical for regulating signalling duration and output[1,5–10]. For instance, we and others have established a link between the recycling of RTKs, such as fibroblast and epidermal growth factor receptors (FGFR and EGFR) or the recycling of integrins and the sustained signalling activation that regulates cell motility[11–13]. Thus, recycling

[1]Division of Evolution, Infection and Genomics, School of Biological Science, Faculty of Biology Medicine and Health (FBMH), The University of Manchester, M139PT Manchester, UK. [2]Division of Molecular and Cellular Function, School of Biological Science, Faculty of Biology Medicine and Health (FBMH), The University of Manchester, M139PT Manchester, UK. [3]Division of Cancer Sciences, School of Medical Science, Faculty of Biology Medicine and Health (FBMH), The University of Manchester, Manchester M20 4GJ, UK. [4]Bio-MS Core Research Facility, FBMH, The University of Manchester, M139PT Manchester, UK. [5]Flow Cytometry Core Research Facility, FBMH, The University of Manchester, M139PT Manchester, UK. [6]Manchester Breast Centre, Manchester Cancer Research Centre, The University of Manchester, M139PT Manchester, UK. [7]These authors contributed equally: Joanne Watson, Harriet R. Ferguson. ✉e-mail: michael.smith-8@manchester.ac.uk; chiara.francavilla@manchester.ac.uk

and its known regulators (e.g. RAB11, TTP and RCP)[1,8,12,13] maintain homoeostasis in health and their dysregulation leads to several diseases, including cancer, diabetes, viral infections, and neurodegeneration[8,14,15]. However, we still do not know which RTK signalling partners are recruited in close proximity to the recycling endosomes during recycling and, thereby how they modulate downstream cellular responses.

The view of endocytosis as a way to attenuate signalling by receptor down-modulation and by controlling receptor availability at the cell surface has been challenged by data linking endocytosis to the propagation of RTK signalling from endosomes[6,7,9]. For instance, EGFR signalling from early endosomes leads to AKT phosphorylation and cell survival[16], suggesting that EGFR internalisation is required for the full spectrum of signalling activation downstream of EGFR. Scaffold proteins recruited to early endosomes regulate a certain branch of signalling, exemplified by the p14/MP1 complex engaging the kinase ERK-MAPK[17,18]. Another example of the crucial role of endosomes as signalling regulators is the recruitment of the LAMTOR complex to the lysosomes, which regulates the kinase mTOR in response to nutrients and growth factors with consequences for signalling, cell growth, metabolism, and autophagy[19]. However, much less is known about the role of recycling endosomes as signalling platform compared to early endosomes or lysosomes[20,21]. This knowledge would allow for specifically modulating cellular signalling. For instance, depleting the recycling adaptor RCP in cancer cells not only switches EGFR trafficking from recycling to degradation but also decreases cell proliferation and migration[12]. Recently, we found a reciprocal regulation between FGFR2b and EGFR signalling outputs which (i) occurs at the recycling endosomes; (ii) leads to FGFR2b-dependent phosphorylation of EGFR on threonine 696 (T693) and of the cell cycle regulator CDK1 on T161; (iii) regulates cell cycle progression[22]. This data suggests that the recycling endosomes can integrate and propagate signals, prompting us to further investigate which FGFR2b signalling partners are specifically recruited in close proximity to the recycling endosomes.

The FGFR family is a useful model for studying the contribution of trafficking to signalling outputs[23]. There are four FGFRs, with FGFR1-3 having splice-variants denoted as b and c isoforms, and 21 FGF ligands, with each FGFR/FGF pair regulating signalling specificity in a context-dependent manner during development, in maintaining adult homoeostasis, and in several diseases such as cancer[24–26]. One stark example of such functional selectivity is given by FGFR2b, which is expressed on epithelial cells[24,25,27]. Stimulation of FGFR2b with FGF7 induced receptor degradation in contrast to stimulation with FGF10, which resulted in recycling of FGFR2b via RAB11-positive recycling endosomes[13,22,28]. These two different trafficking routes of FGFR2b were associated with different phosphorylation dynamics within the signalling cascade and an increase in cell proliferation and proliferation/migration, respectively[13,22]. Therefore, the duration and location of FGFR signalling must be strictly regulated to modulate the appropriate cellular outputs[23,25].

Here, to investigate FGFR2b signalling partners at the recycling endosomes, we developed a Mass Spectrometry (MS)-based phosphoproteomics approach, which allowed us to distinguish sites globally phosphorylated upon FGF10 binding to FGFR2b from those sites specifically phosphorylated in the proximity of the recycling endosomes during receptor recycling. This spatially resolved phosphoproteomics (SRP) approach is based on proximity-dependent biotinylation, which has been recently developed to profile the interactome of internalised receptors such as G-protein coupled receptors (GPCRs) and stress granules[29–31]. Proximity-dependent biotinylation occurs when a bait protein, tagged with a biotin ligase such as BioID or a peroxidase such as APEX2, encounters other proteins within the labelling radius of 10 or 20 nm, respectively[32,33]. Combined with biotin enrichment using streptavidin beads and MS analysis, interactomes of bait proteins can be identified for any subcellular compartment[34,35].

Using peroxidases like APEX2 is preferable for investigating short-acting, dynamic processes, due to their short labelling time of 1 min. This approach was successfully used to define the interactors of GPCRs and of selective autophagy receptor-dependent cargoes[29,36] and was, therefore, our choice. However, we expanded the method by adding a phosphorylated peptide enrichment step after the biotin enrichment of proteins using streptavidin beads and after protein digestion. This strategy has allowed us to uncover FGFR2b signalling partners localised at the recycling endosomes and to study their impact on FGFR2b responses. To dissect the spatially restricted signalling modules regulated by FGF10/FGFR2b during recycling, we combined the SRP approach with traditional quantitative phosphoproteomics of epithelial cells in which FGFR2b recycling was blocked[22]. We found that FGFR2b signalling localised at the recycling endosomes during recycling regulates mTOR-ULK1 signalling with functional consequences for autophagy and cell survival.

## Results

### Inhibiting FGFR2b trafficking alters the phosphoproteome

To investigate changes in FGFR2b signalling during recycling, we have previously analysed the phosphoproteome of cells stimulated with the recycling ligand FGF10[13,22]. Here, we examined the effect of FGFR2b trafficking impairment on the FGF10-stimulated phosphoproteome of the epithelial cell lines HeLa, stably expressing FGFR2b (HeLa_FGFR2b^ST) and T47D, which express endogenous FGFR2b[13,22]. We transiently expressed (more than 80% of positive cells) Dynamin_K44A-eGFP (dominant negative Dynamin, DnDNM2) or eGFP-RAB11_S25N (dominant negative RAB11, DnRAB11), which are known to inhibit FGFR2b internalisation and recycling to the plasma membrane, respectively, in response to FGF10 stimulation for 40 min[22]. At this time point, FGFR2b was localised in the recycling endosomes in cells expressing wild-type eGFP-RAB11 (wild-type RAB11 and wtRAB11) (Fig. 1a, b)[13,22]. We also stimulated cells with FGF10 for 120 min to study the fate of FGFR2b at a longer time point. As shown for FGFR1[37], FGFR2b co-localised with the marker of early endosomes EEA1, and with DnRAB11 in cells expressing DnRAB11 and was not found at the plasma membrane upon 40 or 120 min stimulation with FGF10 (Fig. 1a). These findings suggest that FGFR2b is trapped in EEA1/DnRAB11-positive vesicles. When cells express DnDNM2 (green in the eight bottom panels of Fig. 1a), FGFR2b (red) was detected at the plasma membrane at all time points and was never detected in the cytoplasm, thus indicating a lack of internalisation. Furthermore, there was no co-localisation between FGFR2b and the marker of early endosomes EEA1 in the cytoplasm. The results of this experiment are quantified in Fig. 1b. In conclusion, expressing DnDNM2 and DnRAB11 impair FGFR2b trafficking and will be used here to study trafficking-dependent changes in FGFR2b signalling in response to FGF10. Immunoblot analysis of cells stimulated for early time points (up to 40 min) with FGF10 to replicate the trafficking assay showed that impeding FGFR2b trafficking did not alter FGFR2b activation or the phosphorylation of ERK1/2 downstream of FGF10 (Fig. 1c and Supplementary Fig 1). Therefore, we used Mass Spectrometry (MS)-based quantitative phosphoproteomics to comprehensively investigate changes in FGFR2b signalling beyond phosphorylation of FGFR and ERK when FGF10-dependent FGFR2b trafficking was impaired. We stimulated both HeLa cells expressing FGFR2b and either eGFP (as control, HeLa-FGFR2b GFP), DnRAB11 (HeLa-FGFR2b DnRAB11) or DnDNM2 (HeLa-FGFR2b DnDNM2) and T47D transiently expressing wtRAB11, DnRAB11 or DnDNM2 with FGF10 for 40 min and analysed the proteome and the phosphoproteome by MS (Fig. 2a and Supplementary Figs. 2, 3 and Supplementary Data 1–4). Firstly, we checked that the transient expression of dominant negative proteins did not alter the cellular proteome using Pearson correlation, which was indeed high among all the experimental conditions (Supplementary Fig. 2a, j and Supplementary Data 1, 3). The quality of the 7620 and 8075

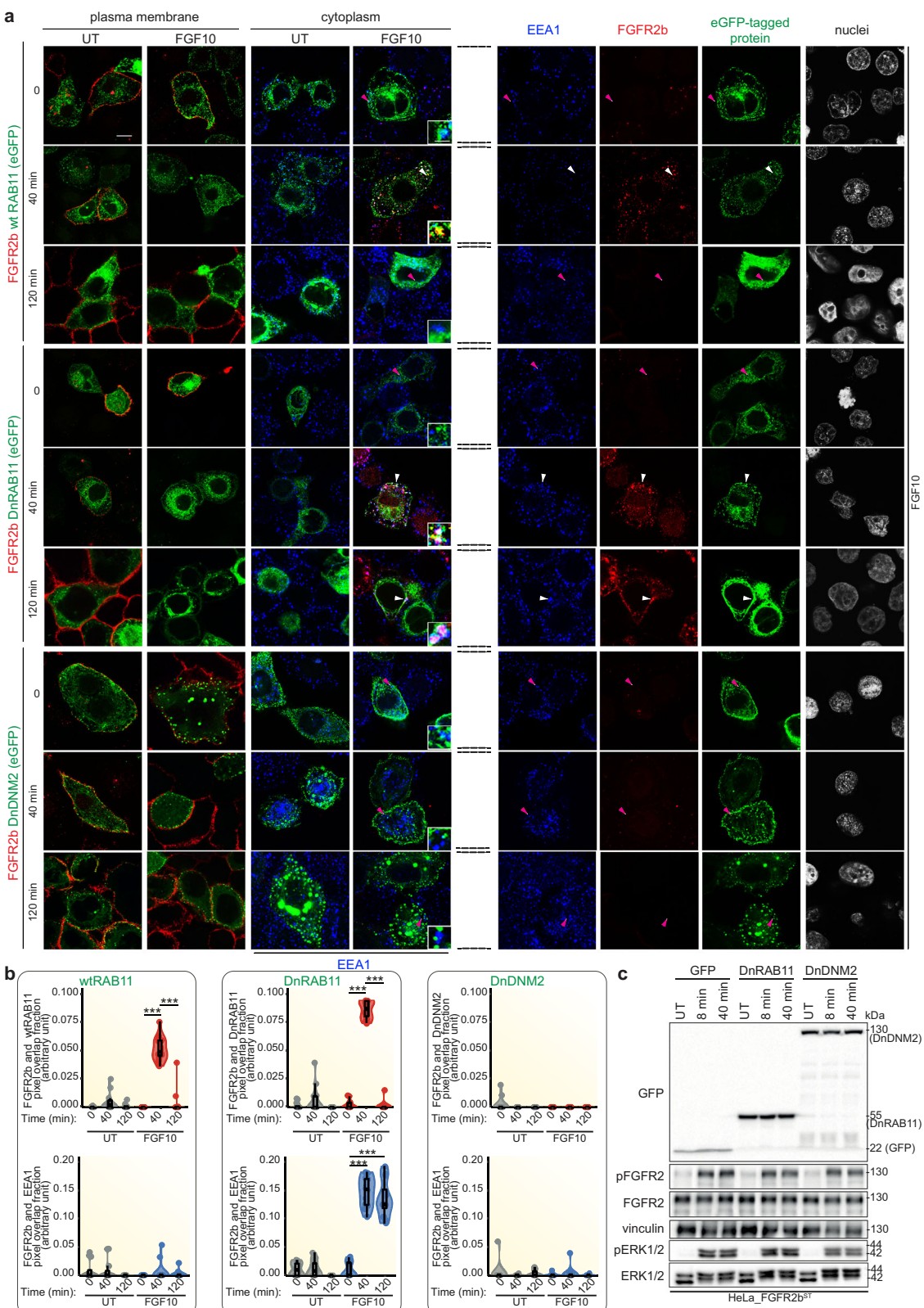

phosphorylated sites quantified in HeLa-FGFR2b and T47D, respectively was consistent with previous publications[22] (Supplementary Fig. 2b–g, k–p and Supplementary Data 2, 4). As HeLa-FGFR2b and T47D expressed different levels of FGFR2b and their proteome and phosphoproteome did not correlate (Supplementary Fig. 3), we first focused on the HeLa-FGFR2b datasets and then used the results to interrogate the T47D datasets. Principal component analysis (PCA)

separated the MS runs based on the experimental conditions (Fig. 2b), suggesting that the location of FGFR2b during FGF10-dependent FGFR2b trafficking affects global signalling activation. To characterise this, we utilised Fuzzy c-means clustering of phosphorylated sites significantly dysregulated across the four conditions (ANOVA, $p$ value <0.0001) and identified 11 clusters (Fig. 2c, d and Supplementary Data 2). We focused on phosphorylated sites either regulated by FGF10

**Fig. 1 | FGFR2b activation is not affected by receptor subcellular localisation.**
**a** Representative confocal image of the presence of FGFR2b (red) in the cytoplasm
and of FGFR2b recycling to the plasma membrane in HeLa cells stably transfected
with FGFR2b-HA (HeLa_FGFR2b$^{ST}$), expressing eGFP-RAB11a (wtRAB11), dominant
negative eGFP-RAB11a_S25N (DnRAB11), or dominant negative dynamin-2_K44A-
eGFP (DnDNM2) (green), and treated with FGF10 for 40 and 120 min. UT, treatment
with vehicle as control). Early endosome antigen 1 (EEA1), blue. Scale bar, 5 µm.
Zoomed images of the regions indicated by the arrowheads (scale bar, 50 µm) and
single channels for FGF10-stimulated cells for 0, 40 and 120 min. are shown in the
inset and on the right after the broken lines, respectively. White arrowheads indi-
cate co-localisation and pink arrowheads indicate a lack of co-localisation.

**b** Quantification of the co-localisation of stimulated FGFR2b (red pixels) with GFP-
tagged proteins (green pixels) indicated by red-green pixel overlap fraction (left
panel). Quantification of the co-localisation of FGFR2b (red pixels) with EEA1 (blue
pixels) indicated by red-blue pixel overlap fraction (right panel). Representative
images are shown in 1a. Values represent median ± SD from $N = 3$ independent
biological replicates where we analysed between 2 and 5 cells for each $N$; ***$p$ value
<0.0005 (one-sided students $t$-test)[22]. **c** Immunoblot analysis ($N \geq 3$ independent
biological replicates) with the indicated antibodies of HeLa_FGFR2b$^{ST}$ cells
expressing GFP, DnRAB11 or DnDNM2 treated with FGF10 for 8- and 40-min. UT,
treatment with vehicle as control. Source data are provided as Source Data file.

but not affected by the expression of DnRAB11 or DnDNM2 (clusters 3
and 4), sites regulated by FGF10 but dysregulated by the expression of
DnDNM2 (clusters 5 and 8), or sites regulated by FGF10 but dysregu-
lated by the expression of both DnRAB11 and DnDNM2 (clusters 9 and
11), hereby defined as the membrane, the internalisation, and the
recycling response clusters, respectively (Fig. 2d). Next, we used the 11
clusters identified in the HeLa-FGFR2b phosphoproteome as a training
dataset to identify clusters with corresponding patterns of regulation
in the T47D phosphoproteome, identifying the three response clusters
corresponding to the membrane, internalisation-dependent and
recycling-dependent signalling (Fig. 2d, e and Supplementary Data 3).
We, therefore, concluded that inhibiting FGFR2b trafficking affects the
regulation of global signalling pathways regardless of FGFR2b levels
and of the overall proteome, indicating the fundamental role of recy-
cling in regulating specific branches of FGFR2b signalling. Indeed,
over-representation analysis (ORA) of KEGG pathways identified MAPK
signalling as regulated downstream of FGFR2b activation regardless of
receptor localisation, whereas mTOR signalling was identified as a
pathway enriched for proteins dysregulated in the recycling response
cluster common to both HeLa-FGFR2b and T47D cells (highlighted in
red in Fig. 2f). However, we did not find any signalling pathways spe-
cifically enriched upon inhibition of FGFR2b internalisation only (cells
expressing DnDNM2 and stimulated with FGF10) (Fig. 2f). This sug-
gests that the FGFR2b recycling route and not merely the presence of
FGFR2b in the cytoplasm regulates specific branches of FGFR2b sig-
nalling in epithelial cells. We used our recently developed visualisation
tool[38] to analyse the subcellular localisation of proteins in the mTOR
signalling pathway belonging to the recycling response cluster and
found enrichment for vesicles, the recycling endosomes, and the late
endosomes in both cell lines (Fig. 2g). Altogether, this data confirms
that recycling is crucial for FGFR2b signalling and identifies the mTOR
pathway as a key downstream signalling effector. Overall, we con-
cluded that FGFR2b likely recruited and specifically phosphorylated
signalling partners in the proximity of the recycling endosomes during
receptor recycling and that this causes changes in downstream
FGFR2b global signalling.

## Phosphorylated proteins are enriched at the recycling endosomes
To further investigate recycling-dependent FGFR2b signalling in a
spatially resolved (at the recycling endosomes) and temporally sen-
sitive (40 min simulation with FGF10) manner, we used the APEX2-
based proximity labelling method[34] (Fig. 3a). This method involves
the fusion of a 27 kDa peroxidase enzyme to a bait protein (FGFR2b-
HA, eGFP-wtRAB11 and eGFP in this study) that will rapidly biotin-
label proteins within 20 nm of the bait protein in less than 1 min
following addition of biotin-phenol (BP) and hydrogen peroxidase
(H$_2$O$_2$) as an oxidant[33]. Biotinylated proteins can then be pulled-down
using streptavidin beads and analysed using MS-based proteomics[34].
We designed an APEX2-based experiment to identify the signalling
partners associated with FGF10-dependent FGFR2b that is localised
at the RAB11-positive recycling endosomes. We stably transfected
HeLa or T47D cells with FGFR2b-HA-APEX (HeLa_FGFR2b-APEX2$^{ST}$

and T47D_FGFR2$^{KO}$_FGFR2b-APEX2$^{ST}$) and verified that FGFR2b sig-
nalling was not altered by the presence of APEX2 upon
FGF10 stimulation over time (Fig. 3b and Supplementary Fig. 4a). To
verify whether FGFR2b trafficking was affected by APEX2, we used
two well-established confocal-based methods, which allowed us to
monitor receptor internalisation and recycling (see Methods) and to
quantify FGFR2b-APEX2 co-localisation with known markers of
trafficking[13,22,37]. FGF7 induced FGFR2b internalisation followed by
receptor degradation, as shown by the lack of staining at the plasma
membrane or in the cytoplasm of cells stimulated for 120 min (Sup-
plementary Fig 4b, c), as previously reported[13,22]. FGF10 induced
FGFR2b to gradually disappear from the cell surface, accumulate in
the cytoplasm, and recycle back to the plasma membrane in all the
tested cell lines (Supplementary Fig 4b, c). Furthermore, FGFR2b co-
localised with Rab11 or with Rab11 and EEA1 in HeLa_FGFR2b-APEX2$^{ST}$
expressing Rab11 or DnRAB11, respectively, and remained at the
plasma membrane in HeLa_FGFR2b-APEX2$^{ST}$ expressing DnDNM2
(Fig. 3c, d), as previously shown in HeLa_FGFR2b$^{ST}$ (Fig. 1a, b). Alto-
gether this data indicates that APEX2 did not alter FGFR2b trafficking.
To exclude events occurring at the recycling endosomes and in the
cytoplasm independent of FGFR2b activation, we expressed either
eGFP-RAB11-APEX2 and eGFP-APEX2 in HeLa-FGFR2b$^{ST}$ (HeLa-
FGFR2b$^{ST}$_RAB11-APEX2 and HeLa-FGFR2b$^{ST}$_GFP-APEX2), respec-
tively (Fig. 3a). For all three APEX2-tagged bait proteins, biotin-
phenol treatment for 40 min and H$_2$O$_2$ incubation for 1 min followed
by streptavidin beads pulldown (hereby pulldown) enriched for the
bait protein, biotinylated proteins and phosphorylated proteins
(Fig. 3e and Supplementary Fig. 4d, e). Indeed, following 1 and 8 min
FGF10 treatment, known interactors of FGFR2b, such as phos-
phorylated PLCγ and SHC, but not histone (H3), were identified in the
pulldown without changes in their basal-level activation (Fig. 3e and
Supplementary Fig. 4e). Next, to confirm that RAB11-APEX2 suc-
cessfully enriched biotinylated proteins in the proximity of the
recycling endosomes during FGFR2b recycling, we immunoblotted
the RAB11-APEX2 pulldown for RAB25 and HA-FGFR2b (Fig. 3f, g).
RAB11-APEX2 is also associated with other known markers of recy-
cling, including RCP[12] (Supplementary Fig. 4f), confirming that our
approach allows the detection of proteins in the proximity of RAB11-
positive recycling endosomes. Taken together, this data supports the
use of APEX2 to reveal phosphorylated signalling partners recruited
to FGF10-stimulated FGFR2b at RAB11-positive recycling endosome.

## Spatially resolved phosphoproteomics shows unknown FGFR2b partners
To uncover FGFR2b signalling partners at the recycling endosomes in
an unbiased manner, we designed a phosphoproteomics approach
based on the detection of phosphorylated proteins in the pulldowns
from cells expressing FGFR2b-APEX2 or RAB11-APEX2 (Fig. 3), hereby
referred to as the Spatially Resolved Phosphoproteomics (SRP)
approach (Fig. 4a). HeLa_FGFR2b-APEX2$^{ST}$ expressing either RAB11,
RAB11-APEX2 or GFP-APEX2 were treated with FGF10 for 40 min
alongside HeLa_FGFR2b-APEX2$^{ST}$ expressing RAB11 and HeLa-FGFR2b$^{ST}$
GFP-APEX2 treated with vehicle, as controls. We collected both the

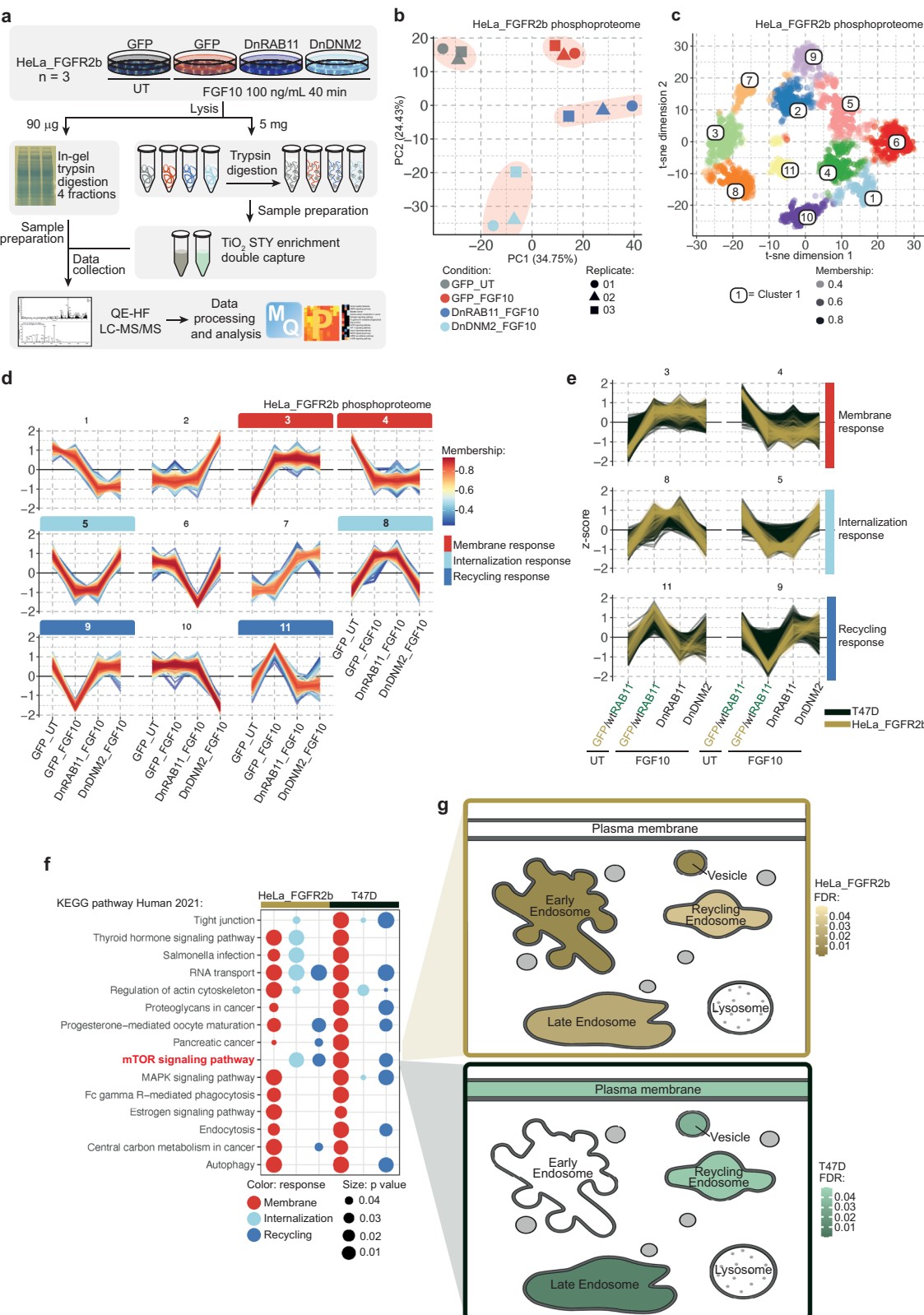

global proteome and phosphoproteome (obtained after TiO$_2$-based chromatography enrichment of phosphorylated peptides) and the proximal proteome and phosphoproteome obtained after enrichment of biotinylated proteins proximal to APEX2-tagged protein baits with streptavidin beads (proteome) followed by protein digestions and TiO$_2$-based chromatography enrichment of phosphorylated peptides (phosphoproteome) (Fig. 4a, Supplementary Fig. 5 and Supplementary

Data 5, 6). We first checked the quality of the global and the proximal proteome and phosphoproteome quantitative datasets, which both showed a strong correlation between replicates and a clear distinction between the global and the proximal samples as assessed by Pearson correlation and PCA plots (Supplementary Fig. 5a–d). Interestingly, the number of phosphorylated sites quantified in the proximal samples (2447) was substantially lower than the number quantified in the global

**Fig. 2 | Phosphoproteomics analysis identifies FGFR2b internalisation- and recycling-dependent signalling pathways. a** Workflow of the phosphoproteomics experiment in HeLa cells transiently expressing FGFR2b (HeLa_FGFR2b) and either GFP, DnRAB11 or DnDNM2 and treated with FGF10 for 40 min. UT, treatment with vehicle as control. **b** Principal component analysis (PCA) of the HeLa phosphoproteome from 2a showed small variation between technical replicates and separated samples based on experimental conditions. **c** t-distributed stochastic neighbour embedding (t-sne) analysis identified 11 clusters corresponding to phosphorylated peptides differentially regulated among four conditions. Colour corresponds to the cluster with the highest membership score, determined using fuzzy c-means clustering based on the median z-score of the four conditions. Each cluster is identified by a unique colour and corresponding number. Opacity corresponds to the membership score assigned to each phosphorylated site within its most likely cluster. Membership scores and t-sne coordinates are available in Source Data File. **d** Plots of the median z-scored intensities of phosphorylated sites

based on the 11 clusters from Fig. 2c identified membrane response (red; clusters 3 and 4), internalisation response (light blue; clusters 5 and 8), and recycling response (dark blue; clusters 9 and 11). Colour key indicates membership value assigned by Fuzzy c-means clustering. **e** Plots of the median z-scored intensities of phosphorylated sites based on the three main clusters identified in Fig. 2d. HeLa_FGFR2b (yellow) and T47D (dark green) were treated as in Fig. 2a and Supplementary Fig. 2h, respectively. **f** KEGG pathway enrichment (calculated with Fisher's exact test and FDR adjustment) between HeLa_FGFR2b and T47D phosphorylated proteins within the membrane (red), internalisation (light blue) and recycling (dark blue) responses (Fig. 2d) identified mTOR signalling as associated with FGFR2b recycling. The size of dot indicates statistical significance based on p value. **g** Visualisation of the subcellular localisation of the phosphorylated proteins belonging to the mTOR signalling pathway KEGG term in HeLa_FGFR2b (yellow) and T47D (green) using SubCellularVis[38].

samples (8545) (Supplementary Fig. 5e, f and Supplementary Data 6). This reduction was not seen in the number of quantified protein groups, indicating that the enrichment for phosphorylated peptides that follows the enrichment for biotinylated proteins had a substantial effect on the identification and quantification (Supplementary Fig. 5e, f). To assess how the double enrichment for biotinylated proteins followed by the enrichment for phosphorylated peptides affected the raw data analysis, we checked the confidence of identification and the distribution of intensities (Supplementary Fig. 5g, h). We found that there was a left-shift in the distribution of the intensities of the double-enriched proximal samples compared to the global samples, and therefore we normalised the global and the proximal phosphoproteome samples separately (Supplementary Fig. 5h). This analysis did not affect the overall quality of the proximal phosphoproteome data, as we found that >89% (6224 and 11726 for global and proximal, respectively) of the phosphorylated sites identified were Class I (≥0.75 localisation probability[39]) and had the expected proportions of single or multiple phosphorylated sites on serine, threonine, or tyrosine residues (Supplementary Fig. 5i)[22]. Finally, we statistically confirmed that the APEX2 tag did not affect the quantification of the global phosphoproteome (Supplementary Fig. 5j), as expected based on immunoblot analysis with the APEX2-tagged proteins (Fig. 3b). We concluded that the double enrichment of biotinylated proteins and phosphorylated peptides did not impact data quality. Given these results, in subsequent analyses the global and proximal phosphoproteome quantitative data were analysed separately (Fig. 4b and Supplementary Fig. 5k).

To reveal the phosphorylated interactome of FGFR2b when localised at the recycling endosomes we normalised the log2 transformed data from the control and from the FGF10-treated FGFR2b-APEX2 and RAB11-APEX2 samples against the corresponding time points of the GFP-APEX2 samples (Fig. 4b and Supplementary Fig. 5k). Hierarchical clustering of the normalised data revealed a cluster of phosphorylated sites enriched in both the FGFR2b-APEX2 and the RAB11-APEX2 samples treated with FGF10, hereby the FGFR2b recycling proximal signalling cluster (Fig. 4c). We noticed an overlap of 588 proteins between the phosphorylated proteins identified in the proximal phosphoproteome and the proteins identified in the proximal proteome which would most likely represent phosphorylated FGFR2b partners at the recycling endosomes (Fig. 4d). The relatively small overlap (588 over 1099 proteins with phosphorylated sites) may indicate the importance of performing the double enrichment step to reveal spatially resolved, phosphorylated signalling partners of the bait of interest. Interestingly, of the 961 phosphorylated sites on the 588 overlap proteins, 77.4% (743) was also found in the FGFR2b recycling proximal signalling cluster (Fig. 4e). Furthermore, when we compared the FGFR2b recycling proximal signalling cluster with the FGF10-regulated phosphorylated sites from the global phosphoproteome, we found only a small overlap of 107 phosphorylated sites (Fig. 4f). FGFR2

and EGFR were found phosphorylated in this overlap (Fig. 4g). One of the catalytic sites of FGFR2 (Y656)[24] was also identified as part of the internalisation response cluster (Fig. 2), corroborating the role of this site for FGFR2 trafficking[13]. Interestingly, T693 on EGFR was found phosphorylated only in the proximal phosphoproteome (Fig. 4g), consistent with its role in regulating FGFR2b recycling at the recycling endosomes[22]. These findings, altogether, indicate that the SRP approach capably distinguished the FGFR2b proximal phosphoproteome enriched at RAB11-positive endosomes from the FGFR2b global phosphoproteome.

KEGG pathway enrichment analysis of the FGF10 global phosphoproteome, FGFR2b Recycling Proximal Signalling Cluster, and the subset of the latter overlapping with the proximal proteome (orange in Fig. 4h) revealed six terms specifically enriched in the FGFR2b proximal datasets, among which autophagy. Interestingly, mTOR signalling pathways, which suppress autophagy[19], was enriched in both the global and proximal FGFR2b phosphoproteome (Fig. 4h). We, therefore, hypothesised that mTOR signalling may be integrating at the global and the proximal level downstream of FGFR2b activation, before converging to regulate autophagy in the proximity of the recycling endosomes during FGFR2b trafficking.

## FGFR2b recycling suppresses mTOR/ULK1-dependent autophagy

To investigate the link between FGFR2b proximal signalling partners and autophagy regulation downstream of mTOR signalling, we created a subnetwork by extracting those proteins annotated to either autophagy or mTOR signalling pathway in KEGG (Fig. 4h). We found several components upstream of mTOR, including RAF1, MAP2K2, RPS6, as well as the mTOR subunits RPTOR and RICTOR, and several proteins known to regulate autophagy via mTOR signalling, among which SGK1, SQSTM1 (also known as p62), TSC1, and the kinase ULK1[19] (Fig. 5a). A subset of candidates within this network, spanning the proximal and global phosphoproteome were confirmed by immunoblot analysis in T47D to match the patterns identified by the SRP approach (Fig. 5b, Supplementary Fig 6a and Supplementary Data 6). Interestingly, RPTOR phosphorylated at S863 was spatially restricted at the recycling endosomes (Fig. 5b), confirming the link between recycling endosomes and autophagy regulation downstream of FGF10/FGFR2b signalling.

To test whether FGF10-mediated FGFR2b recycling regulates autophagy, we assessed autophagy by four established methods: Fluorescence-activated cell sorting (FACS) analysis of cells with fluorescent staining of pre-autophagosome, autophagosomes and autolysosomes using a commercially available kit; acridine orange staining, widely used to stain lysosomes downstream of autophagy as a proxy for autophagy; immunoblotting of known markers for autophagy; and traditional immunofluorescence staining of autophagosomes, lysosomes and mature autolysosomes[40]. Both FACS analysis and acridine

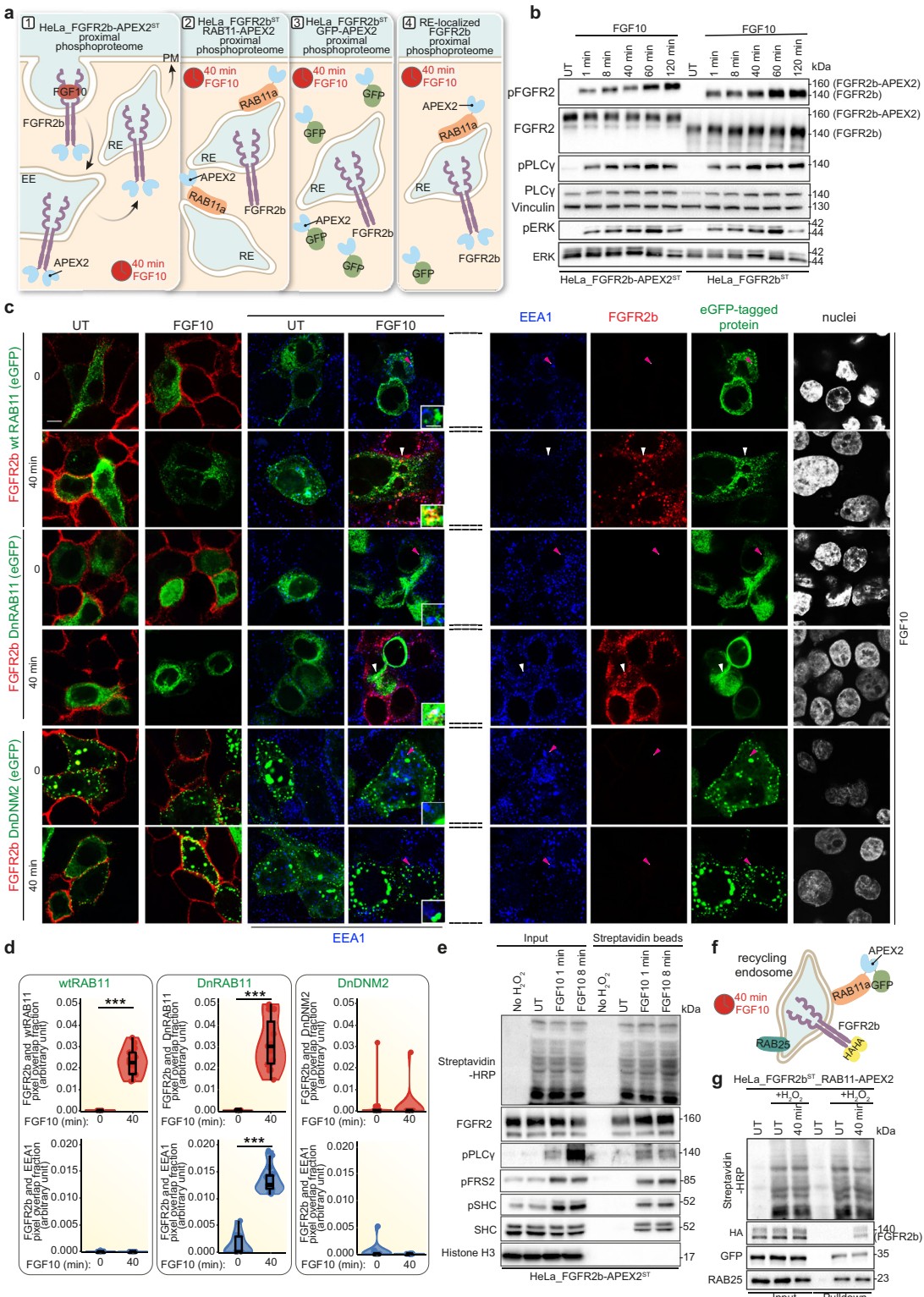

orange staining in HeLa-FGFR2b[ST], T47D and BT20 treated for 2 h with FGF10 and with FGF7 (as a negative control for FGFR2b recycling[13]) showed that FGF10 impaired autophagy compared to control in all cell lines, whereas FGF7 did not (Fig. 5c, d, Supplementary Fig. 6b, c). Based on these results, we decided to use acridine orange staining to evaluate autophagy in our experimental conditions. To assess the dependency of the autophagy response on FGF10-FGFR2b, rather than FGF10-FGFR1b we compared HeLa (with endogenous FGFR1b expression)[37] and HeLa_FGFR2b[ST], showing that only in the presence of FGF10 and

FGFR2b was autophagy reduced (Supplementary Fig. 6d). We also confirmed the effect of FGF7 and of another ligand for FGFR2b, FGF1, on regulating autophagy[41] (Supplementary Fig. 6d). As we starved cells before stimulation with FGFs and starvation is known to increase autophagy[42], we checked the levels of known autophagy markers in starved cells followed or not by stimulation with either serum (as control), FGF7 and FGF10 by immunoblotting. The lipidated form (2) of the autophagosome-formation-associated microtubule-associated proteins 1 A/1B light chain 3B (LC3B)[43] was suppressed to levels seen in

**Fig. 3 | APEX2 tagged-FGFR2b and RAB11a identifies compartment-specific signalling partners upon FGF10 stimulation. a** Schematic underlying the spatially resolved phosphoproteomics (SRP) approach. **b** Immunoblot analysis ($N \geq 3$ independent biological replicates) with the indicated antibodies of HeLa_FGFR2b[ST] (right) or HeLa_FGFR2b-APEX2[ST] (left) stimulated with FGF10 for 1, 8, 40, 60 or 120 min. UT, treatment with vehicle as control. **c** Representative confocal images of FGFR2b (red) internalisation in the cytoplasm and FGFR2b recycling to the plasma membrane in HeLa_FGFR2b-APEX2[ST], expressing eGFP-RAB11a (wtRAB11), dominant negative eGFP-RAB11a_S25N (DnRAB11) or dominant negative dynamin-2_K44A-eGFP (DnDNM2) (green), and treated with FGF10 for 40 min. UT, treatment with vehicle as control. Early endosome antigen 1 (EEA), blue. Scale bar, 5 µm. Zoomed images of the region indicated by the arrowheads (scale bar, 50 µm) and single channels for FGF10-stimulated cells for 0 and 40 min. are shown in the inset and on the right after the broken lines, respectively. White arrowheads indicate co-localisation and pink arrowheads indicate a lack of co-localisation. **d** Quantification of the co-localisation of stimulated FGFR2b (red pixels) with GFP-tagged proteins (green pixels) indicated by red-green pixel overlap fraction (top panel). Quantification of the co-localisation of FGFR2b (red pixels) with EEA1 (blue pixels) indicated by red-blue pixel overlap fraction (bottom panel). Representative images are shown in 3c. Values represent median ± SD from $N = 3$ independent biological replicates where we analysed between 2 and 5 cells for each $N$; ***$p$ value <0.0005 (one-sided students $t$-test). **e** Immunoblot analysis ($N \geq 3$ independent biological replicates) with the indicated antibodies of input or biotinylated proteins enriched with Streptavidin beads from HeLa_FGFR2b-APEX[ST] treated with vehicle (UT), with $H_2O_2$, or with FGF10 for 1 and 8 min. **f** Schematic of FGFR2b localised at RAB11-positive recycling endosomes upon 40 min stimulation with FGF10. **g** Immunoblot analysis ($N \geq 3$ independent biological replicates) with the indicated antibodies of input or biotinylated proteins enriched with Streptavidin beads from HeLa_FGFR2b[ST]_RAB11-APEX2 stimulated with either $H_2O_2$ or with FGF10 for 40 min. UT, treatment with vehicle as control. Source data are provided as a Source Data file.

serum-treated cells by FGF10 treatment alone (Fig. 5e). This FGF10-, but not FGF7-dependent decrease in the levels of lipidated LC3B (2) was seen in HeLa-FGFR2b[ST] and BT20 cells as well, alongside a decrease in active BECLIN1 phosphorylated on S93 (Fig. 5f), another mediator of autophagosome formation and maturation[40,44]. Similarly, we found that p62 (also called SQSTM1, found in Fig. 5a) was stabilised under conditions of increased autophagy[40] (Fig. 5e, f). Finally, we observed an increase in LC3-positive autophagosomes in starved conditions and in FGF7, but not FGF10, stimulated HeLa-FGFR2b[ST] and T47D cells (Supplementary Fig. 6e, f). The number of LAMP1-positive lysosomes did not change in any condition, whereas the number of mature autolysosomes (LC3/LAMP1-positive vesicles) was higher in untreated compared to FGF7 stimulated cells and equal to zero upon FGF10 stimulation (Supplementary Fig. 6e, f), suggesting that starvation and FGF7 or FGF10 treatment differentially regulate the autophagy flux. the results from the four methods used to evaluate autophagy altogether suggest that autophagy regulation is FGFR2b-dependent and requires FGFR2b recycling downstream of FGF10.

To further confirm the importance of FGFR2b in autophagy regulation, we compared autophagy in parental T47D, T47D depleted of *FGFR2*, and T47D depleted of *FGFR2* and overexpressing FGFR2b (T47D_FGFR2b[KO]_FGFR2b[ST]) and found upregulation of autophagy in the absence of FGFR2b and less autophagy in T47D expressing high levels of FGFR2b compared to parental T47D (Fig. 6a, b). Using the same cell model, we also investigated whether autophagy downstream of FGFR2b required mTOR signalling. To test this, we compared autophagy in cells subjected to either starvation, glucose removal and glucose-6-phosphate treatment (which are known to induce mTOR-dependent autophagy) or sodium valproate and fluspiriline treatment (which are known to induce mTOR-independent autophagy)[45,46]. We found that mTOR-dependent but not mTOR-independent autophagy was affected by FGFR2b levels, as cells depleted of *FGFR2* showed the highest levels of mTOR-dependent autophagy, whilst high levels of FGFR2b expression (T47D_FGFR2b[KO]_FGFR2b[ST]) induced the lowest levels of mTOR-dependent autophagy (Fig. 6a). As high expression of RTKs may be associated with higher levels of internalisation in basal conditions and potential difference in signalling regulation[47] (Supplementary Fig. 4b, c), we concluded that internalised FGFR2b was required for the regulation of mTOR-dependent autophagy. Moreover, mTOR-dependent, but not mTOR-independent, autophagy and the autophagy markers lipidated LC3B (2) and phosphorylated BECLIN1 were regulated by FGF10 treatment in parental T47D (Fig. 6c, d). Finally, both mTOR signalling and the known autophagy regulator ULK1 kinase were required for FGF10-dependent regulation of autophagy and of the autophagy marker LC3B 2 in T47D and HeLa cells (Fig. 6e, f). These findings indicate that suppression of autophagy downstream of FGFR2b recycling is mTOR- and ULK1-dependent in FGF10-stimulated epithelial cells.

As we identified ULK1 and RPTOR phosphorylation in the proximal phosphoproteome (Fig. 5a), we next investigated whether ULK1 phosphorylated downstream of mTOR—for instance, on S638 which is known to suppress autophagy[48]—localised at the recycling endosomes during FGFR2b recycling. In both HeLa-FGFR2b[ST] and T47D cells, ULK1 was recruited and phosphorylated on S638 in the proximity of both FGFR2b and RAB11 as shown upon streptavidin beads enrichment of biotinylated proteins followed by immunoblotting (Fig. 7a, b and Supplementary Fig. 7a). We confirmed that FGFR2b and phosphorylated ULK are in close proximity using the Proximity Ligation Assay (Fig. 7c and Supplementary Fig. 7b). Furthermore, confocal analysis of T47D_FGFR2[KO]_FGFR2b-APEX2[ST] cells expressing wtRAB11 and stimulated with FGF10 for 40 min showed a significant co-localisation between phosphorylated ULK1 on S638 and FGFR2b at the recycling endosomes (Fig. 7d, e). These findings confirm that ULK1 is associated to recycling endosomes[49] and suggest that the presence of stimulated FGFR2b at the recycling endosomes is necessary for the recruitment of phosphorylated ULK1 on S638. Indeed, we did not visualise any ULK1 phosphorylated on S638 when FGFR2b recycling was impaired by expressing DnRAB11 (Fig. 7d, e), or when FGFR2b recycling was inhibited through siRNA-mediated knockdown of the FGFR2b-specific recycling adaptors TTP or RCP[22] (Supplementary Fig. 7c). The phosphorylation of ULK1 on S638 downstream of FGFR2b recycling is a specific event, as other FGFR2b downstream pathways, including phosphorylated FRS2 and ERK, were only marginally affected in cells expressing either DnRAB11 or DnDNM2, treated with the primaquine and dynasore compounds, all conditions that impaired FGFR2b trafficking[13,22,50,51] or stimulated with FGF7 which does not regulate FGFR2b recycling (Figs. 1, 7g and Supplementary Fig. 7d, e). Intriguingly, inhibiting FGFR2b localisation at the recycling endosomes by expressing DnRAB11 also misplaced the FGFR2b recycling regulator TTP[13] from recycling endosomes to LAMP1-positive lysosomes (Fig. 7h, I and Supplementary Fig. 7f, g, where it has previously been shown to negatively regulate mTOR signalling[52]. Therefore, we checked mTOR localisation and activation in our experimental conditions. mTOR was localised on lysosomes in both wtRAB11- and DnRAB11-expressing cells (Fig. 7j and Supplementary Fig. 7h). However, mTOR activation decreased in cells with impaired FGFR2b trafficking as shown by the analysis of the level of known genes regulated downstream of mTOR[19] (Supplementary Fig. 7i). We also checked whether inhibiting FGFR2b recycling by expressing DnRAB11 affected other mTOR signalling partners, including RPTOR and AMPK[19]. Inhibiting FGFR2b recycling prevented RPTOR phosphorylation on S863 and AMPK dephosphorylation on T172, two events associated with increased mTORC1 activity (Fig. 7k), Phosphorylation of S638 on ULK1 was also decreased up to 2 h after FGF10 stimulation when FGFR2b recycling was inhibited (Fig. 7k and Supplementary Fig. 7d). These results clearly demonstrate a link between FGFR2b recycling, mTOR signalling, and ULK1 phosphorylation on S638 (Figs. 5a, b, 6, 7 and Supplementary Figs. 6a, 7).

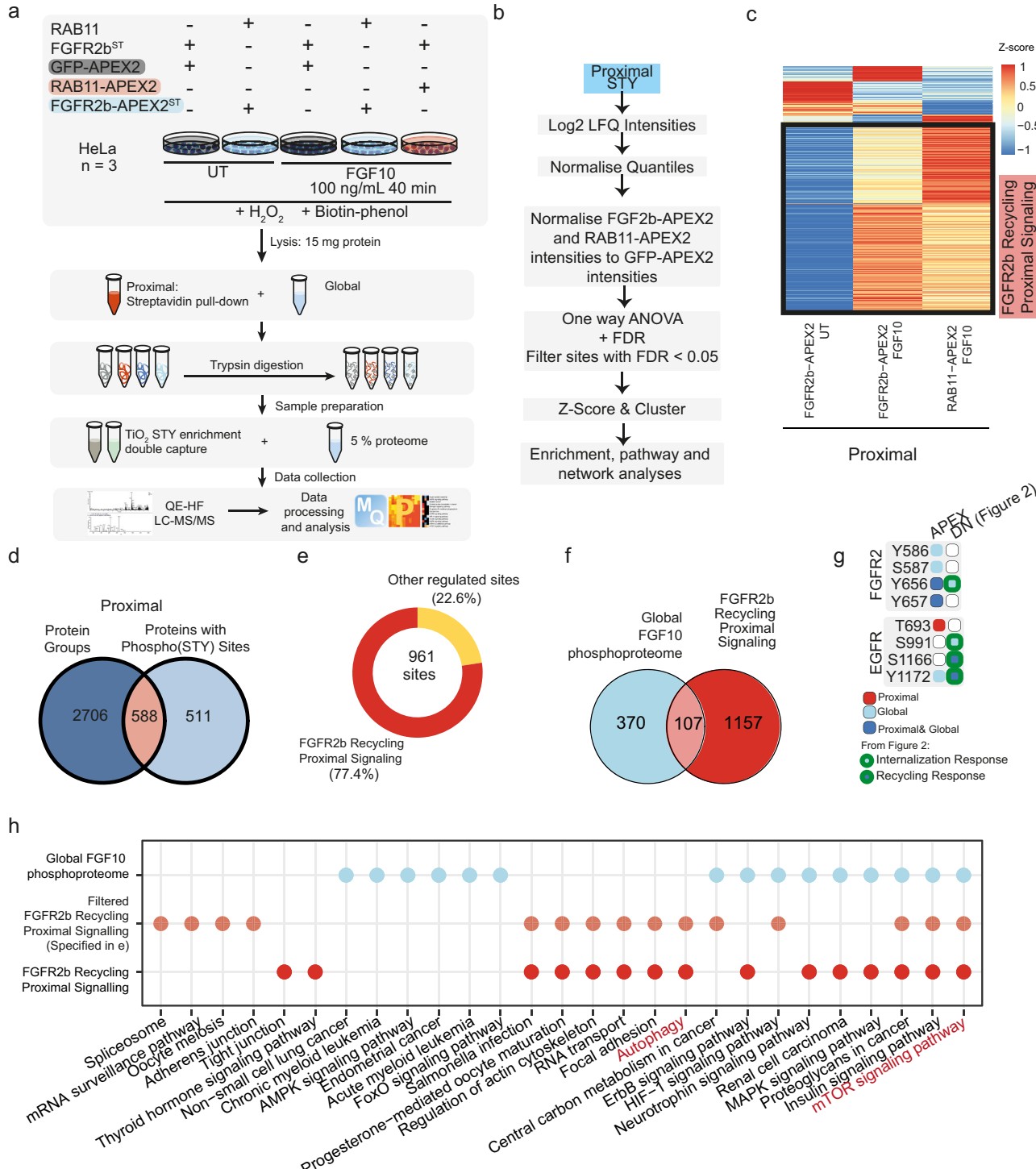

**Fig. 4 | Spatially resolved proteomics and phosphoproteomics reveal FGFR2b-dependent regulation of mTOR signalling and autophagy. a** Workflow of the spatially resolved proteomics and phosphoproteomics experiments in HeLa cells expressing the indicated constructs. **b** Summarised data analysis pipeline of proximal phosphoproteome data. **c** Cluster analysis of the proximal phosphoproteome from the indicated conditions normalised to the proximal phosphoproteome of HeLa_FGFR2b$^{ST}$ GFP-APEX2 for each time point. Phosphorylated sites upregulated at 40 min stimulation with FGF10 in both HeLa_FGFR2b$^{ST}$ RAB11-APEX2 and HeLa_FGFR2b-APEX2$^{ST}$ RAB11 are marked as the FGFR2b Recycling Proximal Signalling Cluster. **d** Overlap of proteins and phosphorylated proteins detected in the proximal proteome and phosphoproteome samples, respectively. **e** Distribution of the phosphorylated sites, 77,4% of which were found in the FGFR2b Recycling Proximal Signalling Cluster. **f** Overlap between the phosphorylated sites upregulated in the global phosphoproteome upon FGF10 stimulation

and the phosphorylated sites upregulated in the FGFR2b Recycling Proximal Signalling cluster from the proximal phosphoproteome (Fig. 4c). **g** Phosphorylated sites identified on FGFR2 and EGFR in the global (blue light) or in the proximal phosphoproteome (red) or in both (blue), and in the phosphoproteome from HeLa_FGFR2b$^{ST}$ cells expressing GFP, GFP-DnRAB11 or GFP-DnDNM2 (Fig. 2a). Light blue with green border indicates phosphorylated sites found in internalisation response clusters and dark blue with green border indicates sites found in recycling response clusters (Fig. 2d). **h** KEGG pathway enrichment (calculated with Fishers Exact Test and FDR adjustment with FDR <0.005) of the phosphorylated sites found in the FGF10 global phosphoproteome (blue light), FGFR2b Recycling Proximal Signalling Cluster (red) and among the phosphorylated sites on proteins quantified at the proteome level from 4e (orange). Source data are provided as a Source Data file.

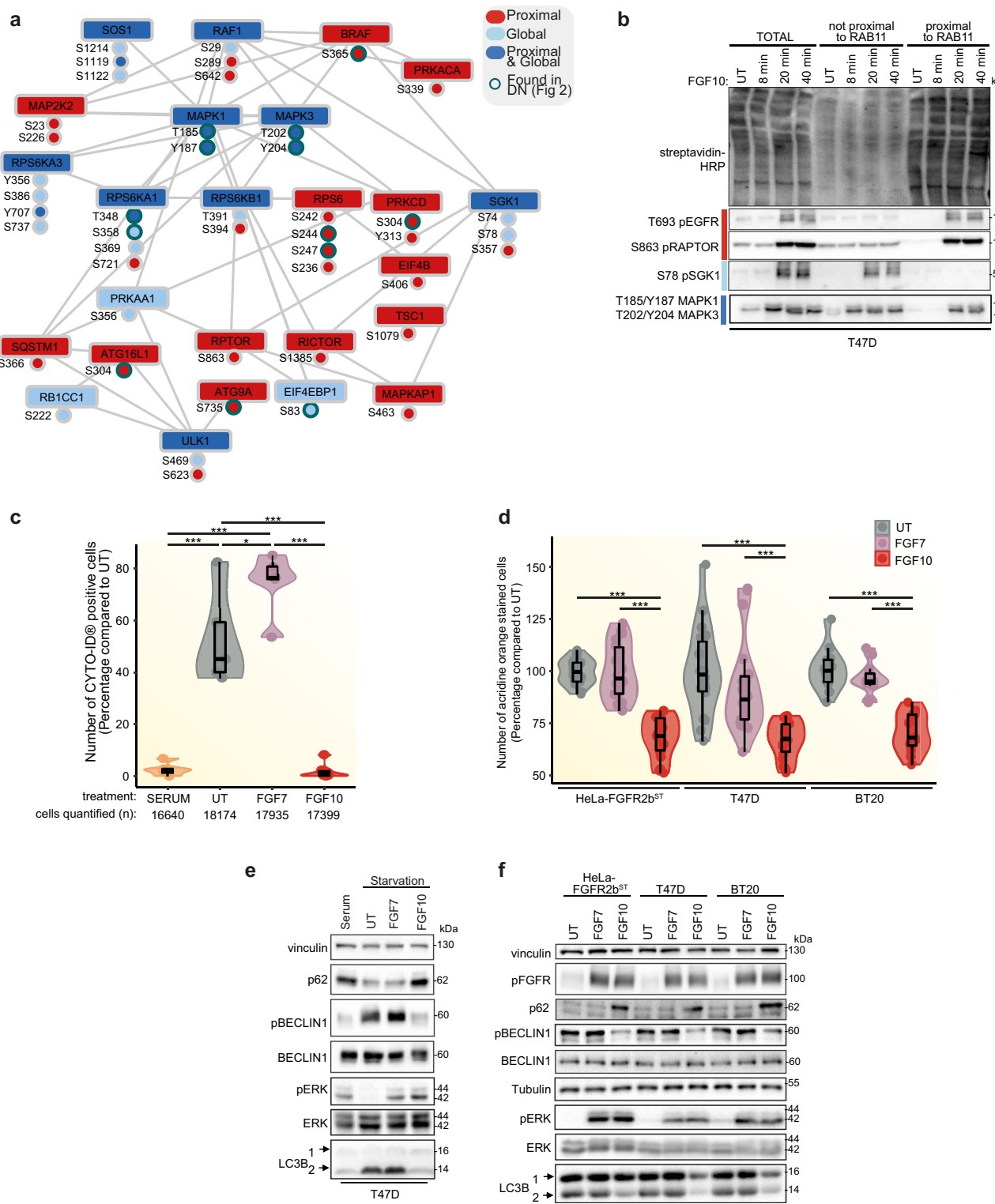

In conclusion, FGFR2b recycling regulates mTOR signalling and the localisation of phosphorylated ULK1 at the recycling endosomes, with these signalling events being crucial for autophagy suppression downstream of FGF10.

## FGFR2b recycling regulates mTOR- and ULK1-dependent cell survival

To investigate whether FGFR2b signalling partners at the recycling endosomes, including phosphorylated RPTOR and ULK1 (Fig. 5a), affected long-term FGFR2b responses during recycling, we tested the impact of impaired FGFR2b trafficking on FGF10-regulated responses. Firstly, we found that autophagy did not change or was slightly increased in FGF10-stimulated cells expressing DnRAB11, as shown by an increase in acridine orange staining in both HeLa-FGFR2b^ST and T47D (Fig. 8a). Furthermore, the number of LC3-positive autophagosomes and the number of LC3/LAMP1-positive mature autolysosomes, but not that of LAMP1-positive lysosomes, increased in cells expressing DnRAB11 compared to cells expressing wtRAB11 upon FGF10 stimulation (Supplementary Fig. 8a, b). The FGFR2b recycling-dependent regulation of autophagy downstream of FGF10 was

**Fig. 5 | FGFR2b regulates mTOR signalling and autophagy from the recycling endosomes. a** Subnetwork of proteins annotated to mTOR pathway or autophagy based on KEGG analysis from Fig. 4h. Node colouring indicates whether the phosphorylated protein or the phosphorylated sites from the KEGG term 'autophagy' were found in global, proximal phosphoproteome or both. Sites from Fig. 2d have a green border. **b** Immunoblot analysis ($N = 3$ independent biological replicates) with indicated antibodies of candidate phosphorylated proteins from the subnetwork (Fig. 5a). T47D were transfected with RAB11-APEX2 (T47D_RAB11-APEX2) stimulated with FGF10 for the indicated time points. UT, treatment with vehicle as control. Non proximal and proximal samples represent the supernatant and the pulldown following enrichment of biotinylated samples with streptavidin beads, respectively, and run against total lysates (total). **c** Autophagy measured using fluorescence-activated cell sorting (FACS) of T47D cells in serum, treated with vehicle (UT), FGF7 or FGF10 for 2 h. Representative images and gating are shown in

Supplementary Fig. 6b–c. The number of cells counted is indicated below the graph. $N = 3$ independent biological replicates. $p$ value <0.05 *, $p$ value <0.0005 *** (one-way ANOVA with Tukey test). **d** Autophagy (measured by staining of autolysosomes with acridine orange) of HeLa_FGFR2b$^{ST}$, T47D and BT20 treated with vehicle (UT), FGF7 or FGF10. $N = 3$ independent biological replicates where at least six treated wells of cells were counted for each $N$. $p$ value <0.001*** (one-way ANOVA with Tukey test). **e** Immunoblot analysis ($N \geq 3$ independent biological replicates) with the indicated antibodies of the effect of serum starvation and FGF treatment on autophagic markers in T47D. LC3B 2 is the lipidated form of LC3B. Cells were treated with vehicle (UT), FGF7 or FGF10. **f** Immunoblot analysis ($N \geq 3$ independent biological replicates) with the indicated antibodies of HeLa_FGFR2b$^{ST}$, T47D and BT20 treated or not with FGF7 or FGF10 for 2 h. Source data are provided as a Source Data file.

confirmed in cells treated with the trafficking inhibitors monensin[37] and dynasore[50] using acridine orange staining, as we observed an increase of autophagy when recycling was inhibited in FGF10-stimulated cells and no changes in cells stimulated with FGF7 for 2 h (Supplementary Fig. 8c). Immunoblot analysis showed reduced levels of lipidated LC3B 2 and RPTOR phosphorylation on S863 and increased phosphorylation of ULK1 on S638 two hours following stimulation with FGF10 in wild-type cells but not in cells expressing DnRAB11 or DnDNM2 (Fig. 8b and Supplementary Fig. 8d). Consistent with previous data, FGF7 did not affect any of these autophagy markers (Figs. 8b, 5e, f and Supplementary Figs. 7e, 8d). Therefore, autophagy is specifically regulated downstream of FGFR2b recycling in response to FGF10. Next, we treated cells with inhibitors of ULK1 (ULK101 and SBI0206965) and of FGFR (PD173074) or mTOR (rapamycin) as controls, and investigated autophagy, apoptosis and proliferation following 24 h stimulation with FGF10 (Fig. 8c–e). ULK1 inhibition reduced autophagy compared to untreated cells, but the level of autophagy was comparable to that of FGF10-treated cells, consistent with our data after 2 h stimulation (Fig. 8c and Supplementary Fig. 8e. As expected, both mTOR and FGFR inhibitors increased autophagy and its markers in response to FGF10 (Fig. 8c and Supplementary Fig. 9e). In addition, ULK and mTOR, but not FGFR, inhibition induced higher levels of apoptosis measured by caspase 3 cleavage and of the apoptotic marker caspase 3 than in FGF10-treated cells (Fig. 8d and Supplementary Fig. 8e). All the inhibitors however induced a decrease in overall cell proliferation (Fig. 8e). Altogether, this data suggests a functional link between FGFR2b recycling, the activation of mTOR and of ULK1 in proximity of the recycling endosomes and the regulation of long-term responses.

In conclusion, we showed the importance of recycling endosomes as signalling platforms to co-ordinate cellular fate by revealing that the inability of active FGFR2b to reach the recycling endosomes disengages the link between mTOR/ULK1 signalling, autophagy and overall cell survival.

## Discussion

The importance of endocytosis in regulating selected RTK signalling cascades to drive cell fate in different contexts, including development or cancer, is now recognised[8,10,14,53,54]. Here, we developed spatially resolved phosphoproteomics (SRP) to uncover FGFR2b signalling partners localised at the recycling endosomes during receptor recycling and we found mTOR-regulated players among them (Fig. 8f). We showed that the autophagy regulator ULK1 was recruited to FGFR2b- and RAB11-positive recycling endosomes to prevent autophagy in FGF10-stimulated cells. The recruitment of phosphorylated ULK1 was prevented in the absence of FGFR2b recycling, resulting in impaired autophagy. Chemical inhibition of ULK1 and of mTOR, one of the known regulators of ULK1[48], not only released FGF10/FGFR2b-dependent suppression of autophagy but also perturbed the longer-term

effects on cell behaviour downstream of FGFR2b activation, including the balance between apoptosis and proliferation (Fig. 8c–e).

Inhibiting FGFR2b trafficking by genetic means alters the global phosphorylation programme in response to FGF10 (Fig. 2), confirming the crucial role of receptor internalisation and recycling in driving signalling dynamics and long-term responses[5,6,13,22]. Indeed, 24.56 and 13.6% of FGF10-dependent regulated phosphorylated sites depended on receptor internalisation and recycling, respectively, in epithelial cells (Supplementary Data 2, 4). This data highlights that certain signalling cascades are activated only when the receptor is at the right place at the right time[55]. Indeed, dysregulation of RTK trafficking leads to alteration in signalling activation such that endocytosis is now considered one of the hallmarks of health and disease, including viral infections, neurodegeneration and cancer[8,14,15,56]. For instance, the FGFR2b internalisation-dependent phosphorylated sites discovered here could inform us on how the protein Dynamin regulates FGFR2b and, more broadly, RTK functions in breast cancer[57]. However, our genetic approach did not distinguish signalling partners specifically recruited to and phosphorylated at the recycling endosomes during FGFR2b recycling. To reveal this, we developed a biotinylation-driven approach that we named spatially resolved phosphoproteomics (SRP). This approach enabled us to generate a snapshot of spatially and temporally resolved signalling partners downstream of FGFR2b (Figs. 4, 5), expanding the analysis from partners in the proximity of a protein bait as shown for GPCRs[29] to phosphorylated partners in the proximity of FGFR2b at the recycling endosomes. We first enriched for biotinylated proteins, and then for phosphorylated peptides (following protein digestion), a method already proven efficient with a BioID-based protocol[58], but that differs from most of the published work in which enrichment for phosphorylated and APEX2-biotinylated proteins was performed prior to protein digestion[59]. Our method therefore resolves phosphorylated sites and not proteins, adding a layer of complexity in the analysis of cellular signalling. We envision that SRP could be easily adapted to study the localised dynamics of other post-translation modifications, thus enlarging the recently published BioID organelle interactome libraries[35]. The advantage of using APEX2-driven biotinylation over BioID or TurboID for defining signalling events in subcellular compartments is the tighter time frame that can be defined (e.g. seconds), which is essential for defining discrete signalling events[60]. The field of spatial proteomics is indeed growing, and technologies are in development to study spatially regulated cellular signalling[34,61–63]. In contrast to other recently developed spatial phosphoproteomics methods[64,65] that enrich for organelles and then analyse global phosphoproteomes, the SRP approach allows the enrichment of phosphorylated proteins at one (or more) organelle under acute stimulation, thus revealing unique phosphoproteome signatures in a spatio-temporal defined manner. In addition, our SRP method provides the exciting opportunity to investigate endosome-proximal phosphorylation events in a high-throughput manner as opposed to

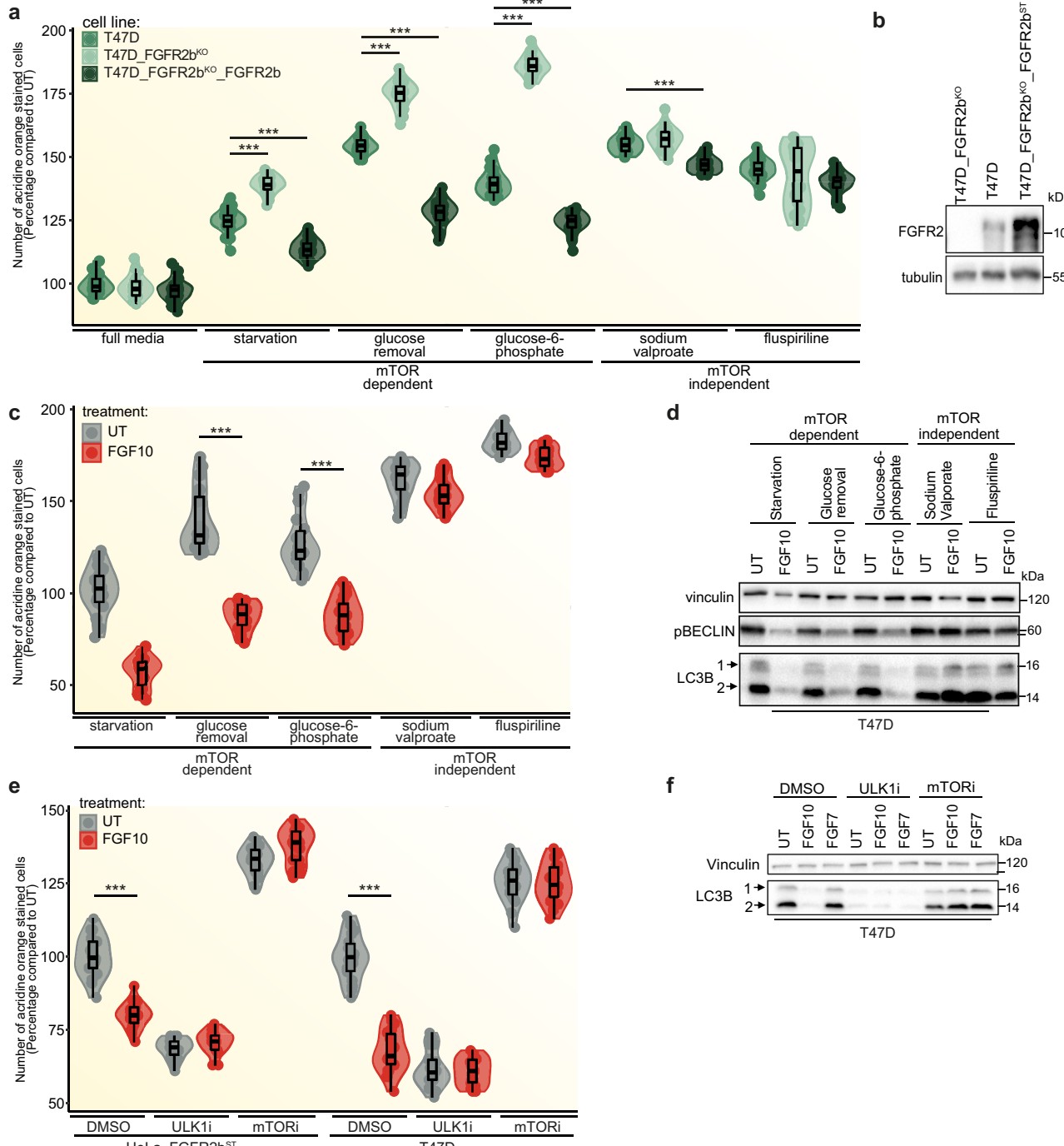

**Fig. 6 | FGF10 regulates autophagy via FGFR2b, mTOR and ULK1. a** Autophagy measured by acridine orange staining after 2 h treatment in the indicated conditions. $N = 3$ independent biological replicates where at least six treated wells of cells were counted. $P$ value <0.001*** (one-way ANOVA with Tukey test). **b** Immunoblot analysis ($N \geq 3$ independent biological replicates) of FGFR2 expression in T47D, T47D_FGFR2b$^{KO}$ and T47D_FGFR2KO_FGFR2b$^{ST}$. **c** Autophagy measured by acridine orange staining after 2 h treatment of T47D cells in the indicated conditions. $N = 3$ independent biological replicates where at least six treated wells of cells were counted. $p$ value <0.001*** (one-way ANOVA with Tukey test). **d** Immunoblot analysis ($N \geq 3$ independent biological replicates) with the indicated antibodies of T47D treated for 2 h in the indicated conditions. **e** Autophagy measured by acridine orange staining after 2 h treatment of HeLa_FGFR2b$^{ST}$ and T47D cells in the indicated conditions. $N = 3$ independent biological replicates where at least six treated wells of cells were counted. $p$ value <0.001*** (one-way ANOVA with Tukey test). **f** Immunoblot analysis ($N \geq 3$ independent biological replicates) with the indicated antibodies of T47D cells treated for 2 h, as indicated. Source data are provided as a Source Data file.

signalling partners identified at early endosomes using biochemical and low-throughput methods. For example, populations of phosphatidylinositol 3-kinase (PI3K) have been shown to be activated at the early endosomes downstream of EGF stimulation[66]. Signalling from the early endosomes has also been described, via MAPK and

JNK[67]. However, signalling partners localised at RAB11-positive recycling endosomes are less known. Here, SRP identifies several phosphorylated sites on core players in mTOR signalling and regulators of autophagy, including ULK1 in proximity of the FGFR2b- and RAB11-positive recycling endosomes.

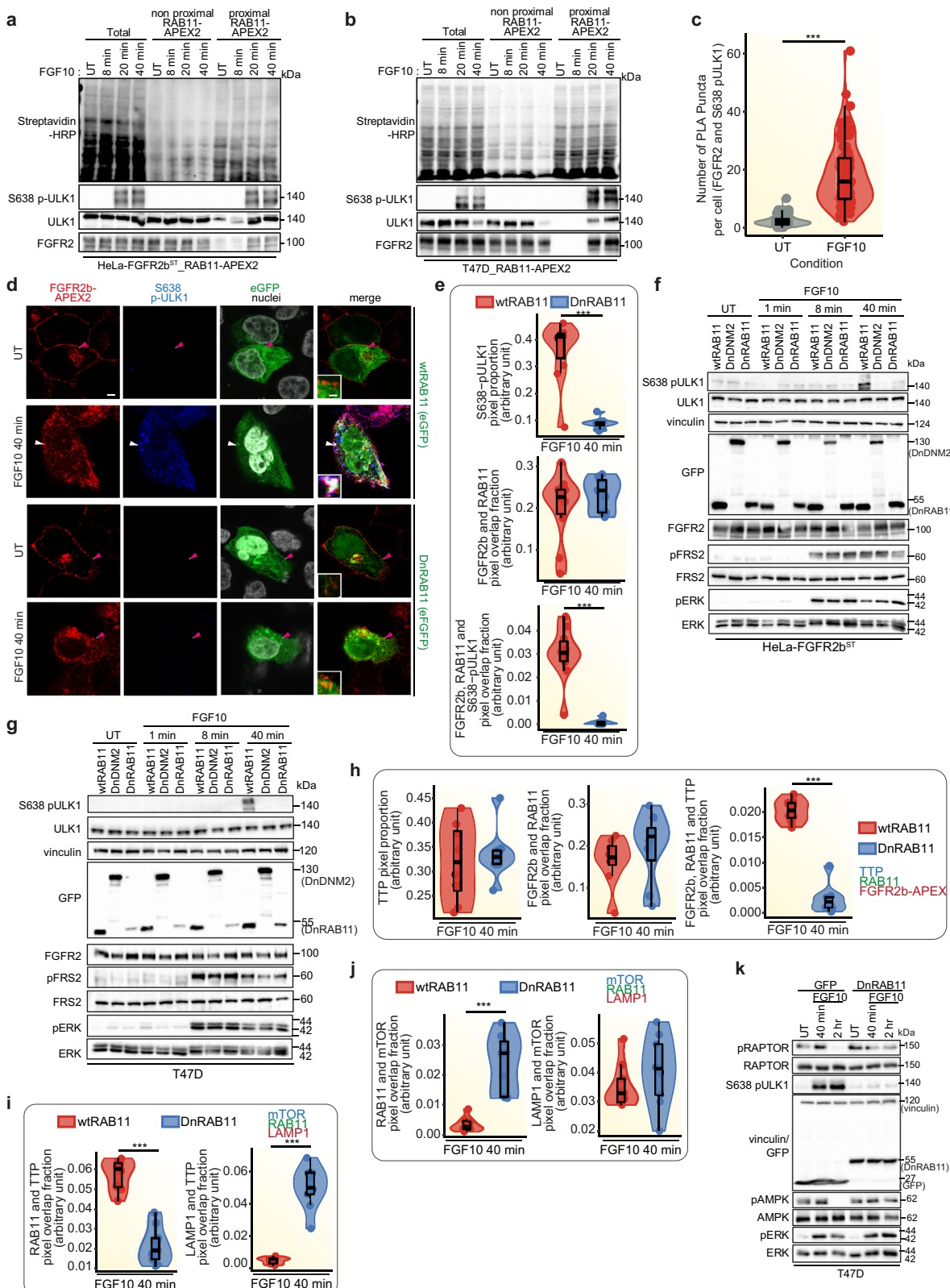

Phosphorylated ULK1 at S638 (its inhibitory state) is primarily seen at the recycling endosomes when stimulated FGFR2b is also localised to recycling endosomes. ULK1 has been previously shown to localise to RAB11-positive recycling endosomes, but its phosphorylation state and the intersection with RTK trafficking have not been previously investigated[48,49]. We also showed that upstream regulators of ULK1, such as mTOR or AMPK, are in the proximity of the recycling endosomes, suggesting that the recycling endosomes may be a site of ULK1 regulation downstream of FGFR2b. The link between mTOR and endocytic trafficking processes such as lysosomal transport has been previously reported[68–70]. A potential dynamic interface between the recycling endosomes and lysosomes, which are localised in close proximity one to each other[71] would favour the interaction of mTOR complexes and ULK1 within the recycling endosomes to drive

**Fig. 7 | Phosphorylated ULK1 is recruited at the recycling endosomes. a**, **b**, **f**, **g**, **k** Immunoblot analysis with the indicated antibodies of HeLa_FGFR2b[ST]_RAB11-APEX2 (**a**), T47D transfected with RAB11-APEX2 (T47D_RAB11-APEX2) (**b**), HeLa_FGFR2b[ST] (**f**) or T47D (**g**) transfected either with wtRAB11, DnRAB11 or DnDNM2 (**f**, **g**), T47D transfected with GFP or DnRAB11 (**k**) stimulated with FGF10 for the indicated time points. Non proximal and proximal samples represent the supernatant and the pulldown following enrichment of biotinylated samples with streptavidin beads, respectively, and run against total lysates (total) (**a**, **b**). N ≥ 3 independent biological replicates. UT, treatment with vehicle as control.
**c** Quantification of proximity ligation assay (PLA) puncta between FGFR2b and S638 pULK1 in HeLa_FGFR2b[ST] cells treated with vehicle (UT) or FGF10 for 40 min.; $p$ value <0.0005 *** (one-sided Students $t$-test). $N$ = 9 independent biological

replicates. **d** Representative confocal images of co-localisation between FGFR2b-APEX2 (red) and phosphorylated ULK1 on S638 (blue) in T47D_FGFR2[KO]_FGFR2b-APEX[ST] transfected with RAB11 or GFP-DnRAB11 (green) and stimulated with FGF10 for 40 min. Scale bar, 5 μm. Inset, zoomed images of the region indicated by the arrowheads (scale bar, 50 μm). The white arrowhead indicates co-localisation, and the pink arrowheads indicate a lack of co-localisation. **e**, **h**, **j**, **i** Quantification of the presence (pixel proportion) and of the co-localisation (pixel overlap fraction) of the indicated proteins. Values represent median ± SD from $N$ = 3 independent biological replicates where we analysed between 2 and 5 cells for each $N$. ***$p$ value <0.0005 (one-sided Student $t$-test). Representative images are shown in Fig. 7d (**e**), Supplementary Fig. 7f (**h**), Supplementary Fig. 7g (**i**) and Supplementary Fig. 7h (**j**). Source data are provided as a Source Data file.

downstream cellular responses. This intriguing possibility is worthy of further investigation—including the analysis of the role of AMPK, another ULK1 regulator[48]. This would have important implications for the understanding of how RTK trafficking and in general the endosomal system, regulates signalling specificity.

RTK signalling and endocytosis have previously been linked to the regulation of autophagy[72] and EGFR recycling has been shown to decrease in cells lacking autophagy regulators[73]. Signals from growth factors are known to converge on the mTORC1 complexes at the lysosomal membrane to inhibit autophagy and catabolic processes[19]. Focusing on the FGFR family, the FGFR2b selective ligand FGF7 has been shown to induce autophagy in keratinocytes after 24 h stimulation[74] and FGF signalling regulates bone growth through autophagy[41]. However, within the 2 h time frame used in our experiments, FGF7 fails to alter ULK1/mTOR signalling or the downstream autophagy response, in contrast to the responses achieved in FGF10-stimulated cells. Indeed, prior to our SRP approach, we had not associated recycling-dependent FGF10-FGFR2b signalling with enhanced mTOR activity. ERK is known to regulate mTOR activity, either indirectly through negative regulation of the TSC complex or by direct phosphorylation of RPTOR[75], while the regulatory relationship between ERK activity reduces AMPK activity in a context-dependent manner[76]. is interesting that ERK activity is comparable between FGF7 and FGF10. However, ERK activation does not lead to mTOR/ULK1-mediated suppression of autophagy downstream of FGF7. This would suggest that a role for ERK in regulating autophagy downstream of FGF7 or FGF10 would be independent of the level of ERK activation. Instead, the recycling endosomes could be required for co-ordinating ERK signalling downstream of membrane activation. The stark difference between FGF7 and FGF10 highlights the role of FGFR2b recycling as the regulator of the FGF10/ULK1/autophagy interplay. How this is orchestrated from the recycling endosomes remains, however, unclear. One possibility is the involvement of EGFR signalling, as we have recently shown that EGFR is phosphorylated downstream of FGF10/FGFR2b recycling at the recycling endosomes[22] and EGFR signalling regulates autophagy[77] with EGFR trafficking requiring autophagy regulators[73]. Alternatively, recycling endosomes and autophagosomes share signalling regulatory components that would require further investigations[49,78]. Thus, a picture of recycling endosomes as a point of convergence for several signalling pathways and for the coordination of long-term responses is clearly emerging. This information can be used to exploit recycling endosomes for nanomedicine, for instance, for better delivery of siRNA against specific signalling players[79].

Recycling is known to control cellular responses, including proliferation, migration, invasion and, as shown in this study, the rate of autophagy[11–13]. It is therefore not surprising that impeding the recycling of FGFR2b leads to dysregulated cellular proliferation and cell death with broader implications for the spatio-temporal regulation of FGFR signalling[80]. We have started dissecting how cell proliferation is tightly regulated by multiple converging mechanisms downstream of FGFR2b, including receptor recycling and its duration[13,22], EGFR, CDK1 and ULK1 phosphorylation occurring at the recycling endosomes[22]

(Fig. 6), and suppression of autophagy (Fig. 7). The G1/S checkpoint is known to be controlled by the homoeostatic balances of nutrients, such as amino acids and sugars, all regulating mTOR signalling[81]. Thus, FGF10 may specifically alter the balance at this checkpoint by suppressing the negative regulation of autophagy on cell cycle progression via mTOR/ULK1 during receptor recycling[81–83]. Alternatively, FGF signalling could regulate the link between cell cycle, number, and size by controlling the activation of CDKs, mTOR, and MAPKs, respectively[84]. The combination of pharmacological inhibition of signalling, autophagy, and mTOR signalling inhibitors has shown greater cytotoxic effects in several diseases[85]. It is, therefore, time to speculate that such a combination may prove efficient in FGFR2b-driven genetic diseases or cancer, including breast cancer[86,87].

In conclusion, we discovered a role for internalised FGFR2b in regulating autophagy from the recycling endosomes. The approach described here, and the datasets collected provide a resource to the cell signalling research community and can be used to further study the role of internalised, activated RTKs in modulating signalling cascades.

## Methods
### Plasmids
eGFP-RAB11 (Addgene #12674); eGFP-RAB11_S52N (mutagenesis of eGFP-Rab11[13]); eGFP (Addgene #34680); dynamin-2_K44A-eGFP (mutagenesis of Dynamin-eGFP[13]); HA-FGFR2b[13]; APEX2 (Addgene #49386); pCDH-EF1-HA-FGFR2b-T2A-mApple and pCDH-EF1-HA-FGFR2b-APEX2-T2A-mApple were generated for this study using the pCDH-mApple backbone[88]. Inserts were PCR amplified, digested and subcloned into the multiple cloning site using MfeI/EcoRI and NotI adaptors. For HA-FGFR2b-APEX2, APEX2 (SalI adaptors) was amplified, digested and subcloned into pDisplay HA-FGFR2b[13]. eGFP-RAB11-APEX2 and eGFP-APEX2 were PCR amplified, digested and subcloned into pcDNA4 (gift form Jennet Gummadova) using EcoRI and XbaI acceptors.

### Antibodies
Antibodies were diluted 1:1000 for immunoblotting and 1:100 for immunofluorescence and validated for the species and the application on the manufacturer's websites. Antibodies purchased from Cell Signaling Technology: p44/42 MAPK (Erk1/2) (137F5) Rabbit mAb (#4695); FGF Receptor 2 (D4L2V) Rabbit mAb (#23328); GFP (D5.1) Rabbit mAb (#2956); Phospho-FGF Receptor (Tyr653/654) (55H2) Mouse mAb (#3476); Phospho-PLCγ1 (Tyr783) Antibody (rabbit polyclonal) (#2821); PLCγ1 Antibody (rabbit polyclonal) (#2822); Phospho-SHC (Tyr239/240) Anitbody (#2434 S); Shc Antibody (rabbit polyclonal) (#2432); Phospho-FRS2-α (Tyr196) Antibody (rabbit polyclonal) (#3864); EEA1 (mouse) (#3288); LAMP1 (rabbit polyclonal) (#15665); LC3B Antibody (rabbit polyclonal) (#2775); Phospho-Beclin-1 (Ser93) (D9A5G) Rabbit mAb (#14717); Beclin-1 (D40C5) Rabbit mAb (#3495); Phospho-ULK1 (Ser638) (D8K9O) Rabbit mAb (#14205); ULK1 (D8H5) Rabbit mAb (#8054); SQSTM1/p62 Antibody (rabbit polyclonal) (#5114); Raptor (24C12) Rabbit mAb (#2280); Phospho-AMPKα

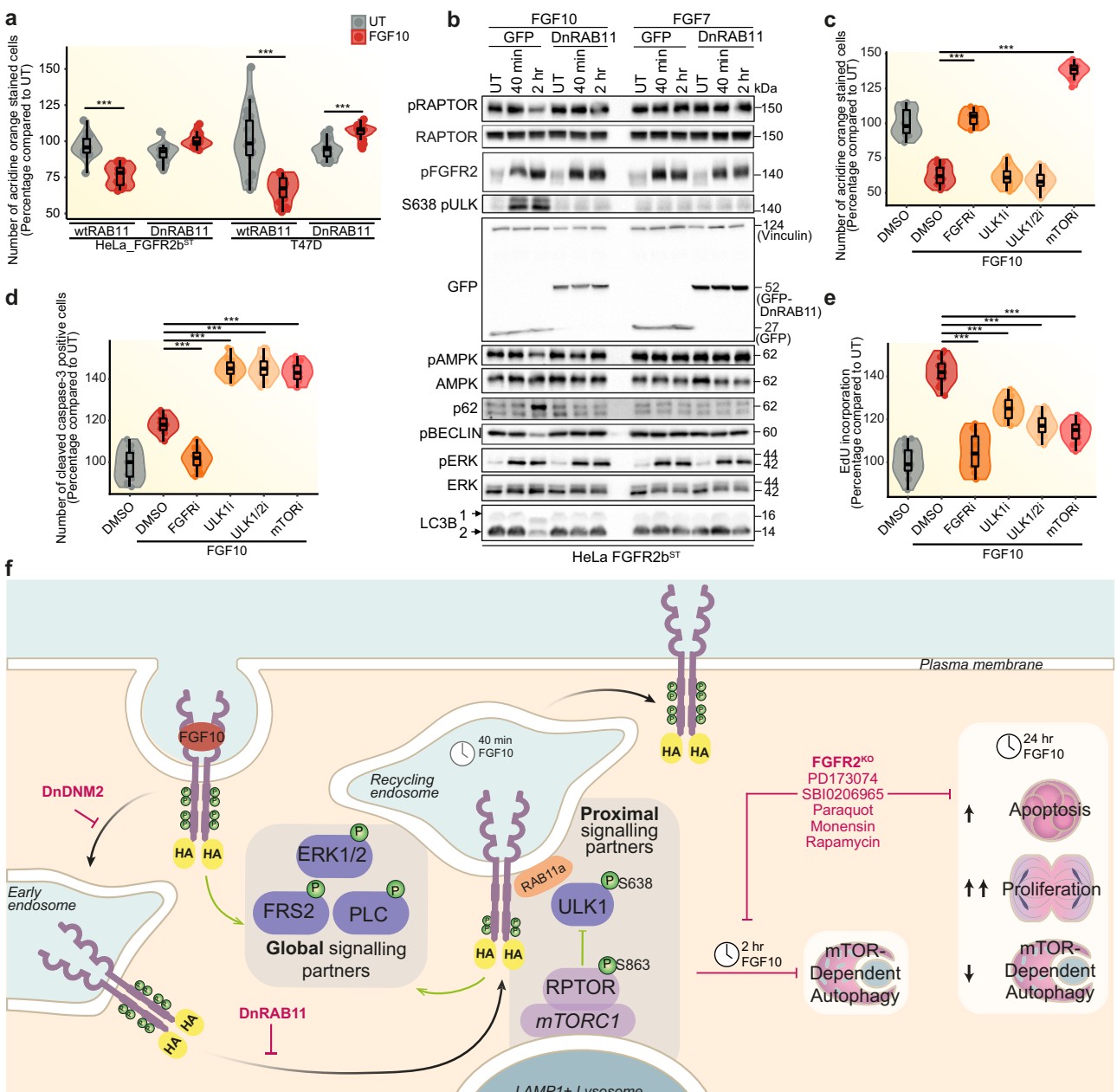

**Fig. 8 | FGFR2b recycling regulates autophagy and the balance between proliferation and cell death. a** Autophagy measured by acridine orange staining of HeLa_FGFR2b$^{ST}$ (left) or T47D (right) transfected either with wtRAB11, DnRAB11 or DnDNM2 and incubated or not with FGF10 for 2 h. $N = 3$ independent biological replicates where at least six treated wells of cells were counted. $p$ value ≤ 0.001*** (one-way ANOVA with Tukey test). **b** Immunoblot analysis ($N = 3$ independent biological replicates) with the indicated antibodies of HeLa_FGFR2b$^{ST}$ transfected either with GFP or DnRAB11 and treated with vehicle (UT) or with FGF7 and FGF10 for different time points. Measurement of autophagy by acridine orange staining (**c**), cell proliferation by EdU incorporation (**d**) and apoptosis by cleaved caspase 3

activated dye (**e**), in T47D treated with the FGFR inhibitor (FGFRi: PD173074), ULK1 inhibitor (ULKi: ULK101), ULK1/2 inhibitor (ULK1/2i: SBI0206965) or mTOR inhibitor (mTORi: Rapamycin), stimulated or not with FGF10 for 2 h. Data were presented as percentage compared to untreated cells. $N = 3$ independent biological replicates where at least six treated wells of cells were counted. $p$ value ≤ 0.001*** (one-way ANOVA with Tukey test). **f** Model of FGFR2b global and proximal signalling partners during recycling to the plasma membrane. Long-term responses are indicated based on the data of this study. The black arrow indicates FGFR2b trafficking. The green arrow indicates events activated by FGFR2b regardless of its subcellular localisation. Source data are provided as a Source Data file.

(Thr172) (40H9) Rabbit mAb (#2535); AMPKα (D5A2) Rabbit mAb (#5831); mTOR (7C10) Rabbit mAb (#2983); Rab7 Anitbody (#2094 S); Cleaved Caspase 3 (Asp175) Antibody (rabbit polyclonal) (#9661 S); purchased from Sigma-Aldrich: Monoclonal Anti-γ-Tubulin antibody produced in mouse (#T5326); Anti-Vinculin antibody, mouse monocolonal (#V9264); Anti-HA (12CA5) (mouse monoclonal); purchased from Abcam: Anti-Histone H3 antibody−Nuclear Marker and ChIP Grade (rabbit polyclonal) (#ab-1791); Anti-Rab25 antibody (rabbit polyclonal) (#ab45855); Anti-LAMP1−Lyososome Marker (rabbit

polyclonal); purchased from Invitrogen: Goat anti-Rabbit IgG (H + L) Secondary Antibody, Alexa Fluor® 488 (#A11034); Goat anti-Mouse IgG (H + L) Secondary Antibody, Alexa Fluor® 488 (#A11001); Goat anti-Rabbit IgG (H + L) Secondary Antibody, Alexa Fluor® 568 (#A11011); Donkey Anti-Mouse IgG (H + L) Secondary Antibody, Alexa Fluor® 647 (#A31571); Donkey Anti-Rabbit IgG (H + L) Secondary Antibody, Alexa Fluor® 647 (#A31573); purchased from other suppliers: Anti-ERK1/2 Antibody (MK1) (mouse monoclonal) (Santa Cruz Biotechnology, #sc-135900); Anti-FRS2 Antibody (mouse monoclonal) (Santa Cruz

Biotechnology, #sc-17841); Purified; Mouse LAMP1/CD107a Lumenal Domain Antibody (polyclonal) (R&D systems, #AF4320); CHMP1b (rabbit polyclonal) (Proteintech, #14639-1-AP); SH3BP4 Polyclonal antibody (rabbit) (Proteintech, 17691-1-AP); FIP1/RCP Antibody (rabbit polyclonal) (Novus Biologicals, #NBP2-20033); Peroxidase-AffiniPure F(a'')2 Fragment Goat Anti-Mouse IgG (H + L) (Stratech, #115-036-045); Peroxidase-AffiniPure F(a'')2 Fragment Goat Anti-Rabbit IgG (H + L) (Stratech, #115-036-062): mouse anti-LC3 antibody (MBL MI52-3).

## Cell culture

Human epithelial cell lines were purchased from ATCC: HeLa (CCL-2), T47D (HTB-133) and BT20 (HTB-19). Cells were authenticated in 2019 through short tandem repeat (STA) analysis of 21 markers by Eurofins Genomics with positive results, checked monthly for mycoplasma via a PCR-based detection assay (Venor®GeM – Cambio), and grown in the indicated media supplemented with 2 mM L-glutamine and 100 U/ml penicillin, 100 µg/ml streptomycin and 10% foetal bovine serum. HeLa, BT20 and Lenti-XL cells were grown in StableCell™ DMEM−- high glucose (Sigma-Aldrich). T47D were grown in RPMI 1640 Medium, GlutaMAX™ Supplement (Gibco).

## Transfection

All transfections were carried out in Gibco opti-MEM glutamax reduced serum media (Thermo Fisher Scientific). HeLa cells were transfected using Lipofectamine 2000 (Thermo Fisher Scientific) according to the manufacturer's instructions, 24 h after RNA interference transfection where indicated. T47D and BT20 cells were transfected using Escort IV according to manufacturer instructions, same as above. Assays were performed 36 h after transfection. A mock control was used if unspecified.

HeLa cells stably expressing HA-FGFR2b or HA-FGFR2b-APEX2 are referred to as follows: HeLa_FGFR2b[ST], HeLa_FGFR2b-APEX2[ST]. Cells were transiently transfected with the following constructs: eGFP (GFP in text), dynamin_K44A-eGFP (DnDNM2 in text), eGFP-RAB11_S25N (DnRAB11 in text), eGFP-RAB11 (wtRAB11 in text), eGFP-RAB11-APEX2 (RAB11-APEX2 in text) and eGFP-APEX2 (GFP-APEX2 in text).

## T47D cells depleted of *FGFR2*

Guide RNAs (crRNA) (IDT) specific to *FGFR2* (FGFR2 crRNA 1: GCCCTACCTCAAGGTTCTCA; FGFR2 crRNA 2: ACCTTGAGAACC TTGAGGTA) were combined with a common trans-activating crRNA (Alt-R® CRISPR-Cas9 tracrRNA) (IDT) to create a functional ribonucleoprotein (RNP) duplex. These were then pre-complexed with Cas9 nuclease (Alt-R® S.p. Cas9 Nuclease V3) (IDT) and transiently transfected into parental T47D using Viromer® CRISPR transfection reagent (Cambridge Bioscience) according to manufacturer's instructions. Colonies were selected and screened for high frequencies of genomic editing using the free Inference of CRISPR Edits (ICE) analysis tool by Synthego. Loss of protein expression was then confirmed by Western blot (see Supplementary Fig. 2). T47D depleted of FGFR2 are referred to as T47D_FGFR2b[KO].

## Lentiviral transduction

Lenti-X cells were transfected in Gibco opti-MEM Glutamax reduced serum media (Thermo Fisher Scientific) with the pCDH-EF1-HA-FGFR2b-T2A-mApple or pCDH-EF1-HA-FGFR2b-APEX2-T2A-mApple viral vectors, alongside VSV-G envelope expressing plasmid pMD2.G and lentiviral packaging plasmids pMDLg/pRRE and pRSV-Rev (all generously gifted from Dr. Hurlstone), using FuGENE® HD Transfection Reagent (Promega), following manufacturer instructions. After 48 h, the lentivirus-containing media was sterile filtered using a 0.22 µm syringe filter and stored at −80 °C. The lentiviral media was added to HeLa or T47D_FGFR2b[KO] in Gibco opti-MEM Glutamax reduced serum media (Thermo Fisher Scientific) containing 10 ng/mL Polybrene Infection/Transfection Reagent (gifted from Dr. Hurlstone).

Colonies were selected and protein expression was then confirmed by Western blot. These cell lines are referred to as HeLa_FGFR2b[ST], HeLa_FGFR2b-APEX2[ST] and T47D_FGFR2b[KO]_FGFR2b-APEX2[ST].

## Sample preparation for quantitative proteomics

HeLa samples for phosphoproteomics were prepared as follows: Cells were washed with PBS and lysed in ice-cold 1% triton lysis buffer supplemented with Pierce protease inhibitor tablet (Life Technologies) and phosphatase inhibitors: 5 nM Na3VO4, 5 mM NaF and 5 mM b-glycerophosphate. 5 mg of protein was obtained for each experimental condition. Proteins were precipitated overnight at −20 °C in four-fold excess of ice-cold acetone. The acetone-precipitated proteins were solubilized in a denaturation buffer (6 M urea, 2 M thiourea in 10 mM HEPES pH 8). Cysteines were reduced with 1 mM dithiothreitol (DTT) and alkylated with 5.5 mM chloroacetamide (CAA). Proteins were digested with endoproteinase Lys-C (Wako, Osaka, Japan) and sequencing grade modified trypsin (modified sequencing grade, Sigma), followed by quenching with 1% trifluoroacetic acid (TFA). Peptides were purified using reversed-phase Sep-Pak C18 cartridges (Waters, Milford, MA) and eluted with 50% ACN and enriched for phosphoserine-, phosphothreonine- and phosphotyrosine-containing peptides, with Titansphere chromatography. 6 mL of 12% TFA was added to the eluted peptides and subsequently enriched with TiO2 beads (5 µm, GL Sciences Inc., Tokyo, Japan). The beads were suspended in 20 mg/mL 2,5-dihydroxybenzoic acid (DHB), 80% ACN, and 6% TFA and the samples were incubated in a sample-to-bead ratio of 1:2 (w/w) in batch mode for 15 min with rotation. After 5 min centrifugation, the beads were washed in 0% ACN, 6% TFA followed by 40% ACN, 6% TFA and collected on C8 STAGE-tips. Elution of phosphorylated peptides was done with 20 ul 5% NH3 followed by 20 µl 10% NH3 in 25% ACN, which were evaporated to a final volume of 5 µL in a sped vacuum. The concentrated phosphorylated peptides were acidified with the addition of 20 µl 0.1% TFA, 5% ACN and loaded on C18 STAGE-tips. Peptides were eluted from STAGE-tips in 20 µL of 40% ACN followed by 10 µL 60% ACN and 100% ACN and reduced to 5 µL by Speed Vac and 5 µL 0.1% FA, 5% ACN was added[22].

HeLa samples for proteomics were prepared as follows: 30 ug of protein was collected as a proteome sample for in-gel digestion[12]. Samples were prepared in lysis buffer as above containing 10 mM DTT, alkylated with 5.5 mM CAA and run on 1.00 mm Invitrogen NuPAGE 4–12% Bis-Tris Gel (Invitrogen) in NuPAGE MOPs running buffer (Invitrogen). Gels were washed in ddH2O, and proteins were subsequently fixed and stained with Colloidal Coomassie Stain (Invitrogen). Each sample was equally separated into four fractions, and de-stained using 50% 20 mM ammonium bicarconate (NH4HCO3) + 50% ethanol (EtOH). Peptides were digested by trypsin in 50 mM NH4HCO3, neutralised and extracted using increasing concentrations of ACN, starting with 50% ACN, 3% TFA and ending with 100% ACN. Digested peptides were evaporated to a final volume of 100 µL in a speed vacuum and loaded on C18 STAGE-tips. Peptides were eluted from STAGE-tips with 40% ACN followed by 60% ACN.

T47D samples for proteomics and phosphoproteomics were prepared as follows: cells were washed with PBS and prepared as described above up to elution off Sep-pak C18 cartridges (Waters, Milford, MA) with 50% ACN. Prior to enrichment for phosphoserine-, phosphothreonine- and phosphotyrosine-containing peptides, with Titansphere chromatography, a small amount of the eluted peptides (1%) was taken for proteome analysis: after evaporation in a speed vacuum, peptides were resuspended in 40 µl of 0.1% TFA, 5% ACN and loaded on C18 STAGE-tips. 6 mL of 12% TFA in ACN was added to the eluted peptides and subsequently enriched with TiO2 beads (5 µm, GL Sciences Inc., Tokyo, Japan). The beads were resuspended in 20 mg/mL 2,5-dihydroxybenzoic acid (DHB), 80% ACN, and 6% TFA and the samples were incubated in a sample-to-bead ratio of 1:2 (w/w) in batch mode for 15 min with rotation. After 5 min centrifugation, the

supernatant were collected and incubated a second time with a two-fold dilution of the previous bead suspension. Sample preparation continued as described above.

HeLa samples for global and proximal proteomics and phospho-proteomics were prepared as follows: Cells were pre-incubated for 40 min with Biotin-Phenol (Iris Biotech) and either left untreated or treated with FGF10 (100 ng/mL) for 40 min. Hydrogen peroxide (Sigma-Aldrich) was added for 1 min before quenching with Trolox (Sigma-Aldrich) and sodium ascorbate (VWR) during ice-cold lysis. Cells were lysed using APEX-RIPA buffer (50 mM Tris-HCl pH 7.5, 150 mM NaCl, 0.1% SDS, 1% Triton, 0.5% sodium deoxycholate, 1x protease inhibitor cocktail, 1 mM PMSF, 10 mM sodium ascorbate, 10 mM sodium azide and 5 mM Trolox) and protein extracted as described above. For each sample, 15 mg of protein was collected. Of this amount,120 µg was taken and run on a gradient gel to acquire the global proteome; 5 mg was precipitated in acetone and processed as described above to obtain the global phosphoproteome; the rest of the lysate was enriched for biotinylation using a 2 h room temperature pull-down with streptavidin beads, to generate the proximal pro-teome. A fifth of the bead slurry was stripped using boiling 4x sample buffer enrich with biotin, the supernatant was run on a gradient gel to acquire the proximal proteome. The remaining streptavidin bead slurry was stripped using boiling 8 M guanidine pH 1.5 supplemented in 5 mM TCEP and 10 mM CAA. Reduced samples were then digested with Lys-C for 60 min RT, diluted to 1 M guanidine using Tris 25 mM pH 8.5, before digestion with trypsin overnight and enrichment for phosphorylated peptides as described above.

## Mass spectrometry

Purified peptides were analysed by LC-MS/MS using an UltiMate® 3000 Rapid Separation LC (RSLC, Dionex Corporation, Sunnyvale, CA) coupled to a QE-HF (Thermo Fisher Scientific, Waltham, MA) mass spectrometer[22]. Mobile phase A was 0.1% FA in water and mobile phase B was 0.1% FA in ACN and the column was a 75 mm × 250 µm inner diameter, 1.7 mM CSH C18, analytical column (Waters). A 1 µl aliquot of the sample (for proteome analysis) or a 3 µl aliquot was transferred to a 5 µl loop and loaded onto the column at a flow of 300 nl/min at 5% B for 5 and 13 min, respectively. The loop was then taken out of line and the flow was reduced from 300 to 200 nl/min in 1 min, and to 7% B. Pep-tides were separated using a gradient that went from 7 to 18% B in 64 min, then from 18 to 27% B in 8 min and finally from 27 B to 60% B in 1 min. The column was washed at 60% B for 3 min and then re-equilibrated for a further 6.5 min. At 85 min the flow was increased to 300 nl/min until the end of the run at 90 min. Mass spectrometry data were acquired in a data-directed manner for 90 min in positive mode. Peptides were selected for fragmentation automatically by data-dependent analysis on the basis of the top 8 (phosphoproteome ana-lysis) or top 12 (proteome analysis) with m/z between 300 to 1750 Th and a charge state of 2, 3 or 4 with a dynamic exclusion set at 15 s. The MS Resolution was set at 120,000 with an AGC target of 3e6 and a maximum fill time set at 20 ms. The MS2 Resolution was set to 60,000, with an AGC target of 2e5, and a maximum fill time of 110 ms for Top12 methods, and 30,000, with an AGC target of 2e5, and a maximum fill time of 45 ms for Top8 analysis. The isolation window was 1.3 Th and the collision energy was 28.

## Raw files analysis

Raw data were analysed by the MaxQuant software suite (https://www.maxquant.org; version 1.6.2.6 and 1.5.6.5) using the integrated Andromeda search engine[89]. Proteins were identified by searching the HCD-MS/MS peak lists against a target/decoy version of the human Uniprot Knowledgebase database that consisted of the complete proteome sets and isoforms (v.2019; https://uniprot.org/proteomes/UP000005640_9606) supplemented with commonly observed con-taminants such as porcine trypsin and bovine serum proteins. Tandem mass spectra were initially matched with a mass tolerance of 7 ppm on precursor masses and 0.02 Da or 20 ppm for fragment ions. Cysteine carbamidomethylation was searched as a fixed modification. Protein $N$-acetylation, $N$-pyro-glutamine, oxidised methionine, and phosphor-ylation of serine, threonine, and tyrosine were searched as variable modifications for the phosphoproteomes. Protein $N$-acetylation, oxi-dised methionine and deamidation of asparagine and glutamine were searched as variable modifications for the proteome experiments. Biotinylation by BP (Y; (C18H23N3O3S)) was included as a variable and fixed modification for global and proximal raw files. Label-free para-meters were used for all the analysis as described[90]. The false discovery rate was set to 0.01 for peptides, proteins and modification sites. The minimal peptide length was six amino acids. Site localisation prob-abilities were calculated by MaxQuant using the PTM scoring algorithm[39]. The dataset was filtered by posterior error probability to achieve a false discovery rate below 1% for peptides, proteins and modification sites. Only peptides with an Andromeda score >40 were included.

## Omics data analysis

All statistical and bioinformatics analyses were done using the freely available software Perseus, version 1.6.5.0 or 1.6.2.1.[91], R framework (version 4.2.0) and Bioconductor (version 3.15)[92], Python framework (version 3.7) (available at http://www.python.org), SubcellulaRVis[38], STRING v11.5[93], Cytoscape (version 3.7.2)[94]. Over-representation ana-lysis (ORA) of KEGG terms was performed using Enrichr and the EnrichR R interface (version 3.15)[95].

All measured peptide intensities were normalised using the nor-malizeQuantiles() function from the R-package limma (version 3.52.3). Potential contaminant proteins or phosphorylated peptides and pep-tides or phosphorylated peptides matching the reverse sequence database were removed[22]. For all datasets, phosphorylated peptides with a localisation score greater than 0.75 were included in the downstream bioinformatics analysis. Pearson correlation was calculated in R.

For both the HeLa and the T47D phosphoproteomics datasets, samples were normalised separately and were grouped based on treatment and only phosphorylated peptides with values in all three replicates of at least one treatment were included in further analysis. Missing values were subsequently imputed from a normal distribution using Perseus default settings. The median $z$-score of intensities were used for further analysis. The HeLa-FGFR2b dataset was separated into eleven clusters by fuzzy c-means clustering using the fanny() function from the R-package cluster (version 2.1.4) performed after a multi-sample ANOVA test with FDR >0.0001 in Perseus. The clustering results of the HeLa-FGFR2b dataset were then used as a training dataset for the classification of phosphorylated sites of the T47D cell line. Kernelized Parzen window (i.e. kernel density estimation) classi-fier scripted via Python library statsmodels (version: 0.11.1) was used as a supervised learning method for generating the classification results. Over-representation analysis (ORA) of KEGG terms was performed using Enrichr and the EnrichR R inferface[95] and significantly over-represented terms within the data were represented in dot plots. The SubCellulaRVis tool was used to visualise the subcellular localisation of proteins[38].

For the SRP datasets, phosphorylated site intensities acquired from global and proximal MS runs were normalised separately. Repli-cates were summarised by calculating the median of normalised intensities. Missing values were imputed using random draws from a truncated distribution with the impute.QRLIC() function from the R CRAN package imputeLCMD. PCA, Students' $t$-test and one-way ANOVA were calculated using the prcomp(), t.test() and aov() func-tions in R, respectively. Hierarchical clustering was performed using the hclust() function in R. ORA and the use of SubCellulaRVis tool were performed as described above[38]. Networks were constructed using

STRING; only interactions with an experimental confidence >0.4 were included. KEGG pathways were extracted using the *EnrichmentBrowser* package from Bioconductor.

## Cell lysis and immunoblotting
Cells were serum-starved overnight in a serum-free medium and treated with PBS or stimulated for the indicated time points with 100 ng/mL of FGF1, FGF7 or FGF10[13]. Where indicated, cells were pre-incubated for 2 h with DMSO or 100 nM PD173074 (Selleckchem, #S1264), 0.5 µM Rapamycin (Sigma-Aldrich, #37094), 2 µM ULK101 (Selleckchem, #S8793), 10 µM SBI0206965 (Selleckchem, #S7885), 10 µM Dynasore (Abcam, #ab120192), Monensin sodium salt, Na⁺ ionophore (Abcam, #ab120499) and 200 µM Primaquine bisphosphate (Sigma-Aldrich, #160393). Control cells were pre-incubated with DMSO alone. Cells treated with glucose-6-phosphate (Sigma-Aldrich, #10127647001), Sodium valproate (Sigma-Aldrich, #BP452) and Fluspiriline (Sigma-Aldrich, #F100) were treated for 24 h prior to FGF10 stimulation. After stimulation, cell extraction and immunoblotting were performed. Proteins were resolved by SDS-PAGE and transferred to nitrocellulose membranes (Protran, Biosciences). Proteins of interest were visualised using specific antibodies, followed by peroxidase-conjugated secondary antibodies and by an enhanced chemiluminescence kit (Amersham Biosciences). Blots were visualised using the Universal Hood II Gel Molecular Imaging System (Bio-Rad). Each experiment was repeated at least three times and produced similar results.

## Biotinylation assays
Biotinylation pull downs experiments were performed as described previously[22]. Briefly, cells were pre-incubated (40 min) with Biotin-Phenol (Iris Biotech) after stimulation with ligands. Hydrogen peroxide (Sigma-Aldrich) was added for 1 min before quenching with Trolox (Sigma-Aldrich) and Sodium ascorbate (VWR) during ice-cold lysis. A 2 h RT pull-down with streptavidin beads was then performed and the supernatant was run against the bound proteins and the total lysates.

## RNA isolation and real-time qPCR analysis
RNA from cell lines was isolated with TRIZOL® (Invitrogen). After chloroform extraction and centrifugation, 5 µg RNA was DNase treated using RNase-Free DNase Set (Qiagen) and 1 µg of DNase treated RNA was then taken for cDNA synthesis using the Protoscript I first strand cDNA synthesis kit (New England Biolabs). Selected genes (Supplementary Table 1) were amplified by quantitative real-time PCR (RT-qPCR) using Sygreen (PCR Biosystems). Relative expression was calculated using the delta-delta CT methodology and beta-actin was used as a reference housekeeping gene. qPCR machine used was Applied Biosystems MX300P.

## Proximity ligation assay
Proximity ligation assay (PLA) was performed using Duolink® In Situ starter kit Mouse/Rabbit (Sigma-Aldrich, DUO92101) following manufacturer's instructions, to assess co-localisation between HA-FGFR2b and pULK1_S638. Briefly, T47D cells were transfected with FGFR2b_HA and 10,000 cells were seeded on the IBIDI slide. Cells were serum-starved for 24 h and placed on ice for 30 min. anti-HA was incubated for 40 min on ice, removed and replaced with serum-free media containing 100 ng/mL FGF10 for 40 min at 37 C. Cells were then fixed with 4% formaldehyde and permeabilised in 0.02% saponin. pULK1 S638 antibody was incubated in Duolink® antibody diluent for 1 h at RT, washed and incubated with Duolink® PLUS/MINUS probes for 1 h at 37 °C. Cells were then washed and incubated with Duolink® ligase solution for 30 min at 37 °C, washed and incubated with polymerase for 100 min at 37 °C. Cells were washed a final time, then mounted using DuoLink® In Situ mounting media containing dapi. Cells were

imaged using a Zeiss Axio Imager upright fluorescence microscope and images analysed using ImageJ.

## EdU incorporation assay
Indicated cells were labelled with 20 µM 5-ethynyl-2′-deoxyuridine (EdU) for 4 h and processed following the manufacturer's protocol (Click-iT® EdU Alexa Fluor® 488 Imaging Kit, Thermo Fisher). EdU is incorporated into newly synthesised DNA, which can be visualised using a fluorescent azide. Prior to imaging cells were then stained with 5 ng/ml Hoechst 3342 for 15 min. Stained cells were analysed using a Leica microscope system. Statistical analysis was performed at the endpoint across repeats, as indicated in the Figure legends.

## Cleaved caspase assay
Apoptosis was measured in cells receiving 24 h treatment with FGF10. Appropriately treated cells were incubated with 20 mM CellEvent™ Caspase 3/7 Green Detection Reagent (Invitrogen) made to 100X in PBS for 4 h in darkness, then washed thoroughly in 1X PBS. This used a fluorogenic substrate for activated caspase 3/7, which is only cleaved by cleaved caspase, enabling DNA binding and fluorescence. Fluorescence was measured at 502 nm excitation and 530 nm emission. Statistical analysis was performed at the endpoint across repeats, as indicated in the Figure legends.

## Acridine orange staining
Populations of cells were assayed for autophagy using 5 mM Acridine Orange (Sigma) for 30 min, after which excess was removed by thorough washing with 1X PBS. This fluorophore appears green when diffuse but is shifted to the red end of the spectrum when accumulated in vesicles[96]. As such, excitation/emission wavelengths of 500/526 nm were used to measure the intensity of diffuse acridine orange (non-specific) and 460/650 nm to assess autophagic staining. The ratio of these values represents stained autophagosomes. Statistical analysis was performed at the endpoint across repeats, as indicated in the Figure legends.

## FACS
Autophagy was assessed by fluorescence-activated cell sorting (FACS) analysis using the CYTO-ID® Autophagy detection kit (Enzo, ENZ-51031-0050), which labels pre-autophagosomes, autophagosomes and autolysosomes with a fluorescent dye, with minimal lysosomal staining. Samples were prepared following the manufacturer's instructions. Briefly, T47D cells with seeded in 6-well plates and serum-starved overnight or kept in full media. Cells in PBS were left untreated (UT) or treated with FGF7 or FGF10 for 2 h. Cells were then trypsinised, pelleted, washed in PBS and resuspended and incubated in CYTO-ID® green stain solution for 30 min, in the dark at RT. Cells were pelleted, fixed in 4% formaldehyde for 20 min in dark and washed three times in PBS before being transferred to 96-well plate for imaging on Amnis® ImageStream®ˣ Imaging Flow Cytometer and analysis using IDEAS® 6.0 software. Plots were gated to eliminate any aggregated cells. A second gate was added to count the single-cell population. A third gate was then added to stratify the cell population with the highest GFP staining. The same gating was applied to each sample allowing a percentage of 'high GFP' cells to be calculated from each single-cell population.

## Immunofluorescence and quantification
To detect HA-FGFR2b or endogenous FGFR2, we incubated cells with 10 µg/ml of anti-HA (Covance) or anti-FGFR2 antibody (Cell Signalling) for 45 min with gentle agitation at 4 °C. The binding of the antibody did not activate receptor signalling in untreated cells nor induced receptor internalisation (see control cells in Figs. 1, 3, 6), as previously reported[12,13,22,37]. After stimulation, cells were incubated at 37 °C for different time points. At each time point, non-permeabilized cells were

either fixed to visualise the receptor on the cell surface (plasma membrane) or acid-washed in ice-cold buffer (50 mM glycine, pH 2.5) to remove surface-bound antibodies. Acid-washed cells were then fixed and permeabilized to visualise the internalised receptor (cytoplasm). Finally, to detect the receptor cells were stained with AlexaFluor488-conjugated donkey anti-mouse or anti-rabbit (Invitrogen). Nuclei were stained with DAPI. Coverslips were then mounted in a mounting medium (Vectashield; Vector Laboratories).

For co-localisation experiments, cells were fixed with 4% paraformaldehyde/2% sucrose for 10 min at room temperature, permeabilized with 0.02% saponin (Sigma) (except for the experiment looking at LAMP1-mTOR co-localisation), blocked in 0.5% BSA and 0.5% Triton X-100 in PBS for 120 min at room temperature. treated with the indicated primary antibody for 60 min at 37 °C (overnight at 4 C in case of mTOR/LAMP1 staining)[97], and stained with AlexaFluor488 (or 568 or 647)-conjugated donkey anti-mouse or anti-rabbit. Samples expressing GFP-tagged proteins were kept in the dark. Nuclei were stained with DAPI. Coverslips were then mounted in a mounting medium (Vectashield; Vector Laboratories).

All the images were acquired at room temperature on a Leica TCS SP8 AOBS inverted confocal using a 100x oil immersion objective and 3x confocal zoom (except for the LC3/LAMP1 staining, where we used 2x zoom). The confocal settings were as follows: pinhole, 1 airy unit, format, 1024 × 1024. Images were collected using the following detection mirror settings: FITC 494–530 nm; Texas red 602–665 nm; Cy5 640–690 nm. The images were collected sequentially. Raw images were exported as.lsm files, and adjustments in image contrast and brightness were applied identically for all images in a given experiment using the freely available software ImageJ v. 1.52p[98].

Quantification of FGFR2b recycling, co-localisation (pixel overlap fraction), and Expression Fraction (pixel proportion) was performed as recently described in detail[22]. Briefly, quantification of internalisation and recycling was performed as follows. For each time point and each treatment, the presence (total) and the localisation (cell surface versus internalised) of HA-FGFR2b or endogenous FGFR2b were assessed in at least seven randomly chosen fields. Approximately 100 cells per condition (both acid-washed and not) were analysed from three independent experiments. The results are expressed as the percentage of receptor-positive cells (green) over total cells (corresponding to DAPI-stained nuclei) and referred to the values obtained at time zero. Statistical analysis was performed across repeats, as indicated in the figure legends.

Quantification of expression fraction, overlap fraction and co-localisation was performed as follows. Images were pre-processed using an À trous wavelet bandpass filter to reduce the contribution of high-frequency speckled noise to the co-localisation calculations. Pixel intensities were then normalised from the original 8-bit range [0,255] to [0,1]. To ensure that co-localisation was only computed in the regions of interest (ROI), we used the Fiji/ImageJ built-in ROI manager to create and record these regions (minimum two cells and up to five per biological replicates with $N = 3$). To measure differences in expression over time or between conditions, we computed the fractions of expressed red marker R, green marker G. or far-red marker F. pixels over a region of interest. To quantify the overlap fraction between two (R and G) or three (R, F and G) markers, we first multiplied the (normalised) channel intensities together to compute a new image whose intensity increases to 1 where the markers strongly overlap and decreases or become null for non-overlapping pixels. Our overlap fraction coefficient (OF) becomes the fraction of strictly positive pixels in the combined image over the number of pixels in the region of interest. Finally, to quantify the actual level of colocalization between two markers (e.g. R and G), we used the Manders Colocalization Coefficients (MCC) M1 and M2. M1 measures the fraction of the R marker in compartments that also contain the G marker, and M2, the fraction of the G marker in compartments that also contain the R marker. Lower-bound thresholds for pixel intensities were automatically determined using the Costes method. To measure the simultaneous overlap of our three, red, far-red and green markers (R, F and G), we first used the overlap image between marker R and marker F, as defined above. We then measured the MCC colocalization parameter of this combined image against a green marker using the MCC formulae above, together with the Costes method to determine the thresholds. The scripts for the quantification of co-localisation were written in the Python language and the code for Costes-adjusted MCC was taken verbatim from the CellProfiler code base. We analysed three independent experiments and between 2 and 5 cells for the experiment. The Student's $t$-test was subsequently used to determine the difference in pixel overlap fraction between different experimental conditions in Figs. 1, 6, as indicated in the figure legends.

Quantification of LC3-, LAMP1- and LC3/LAMP1-positive vesicles was performed manually using ImageJ. For $N = 3$ independent experiment, we analysed between 15 and 25 cells per image by adjusting the threshold of each channel to a value equal to 50, followed by particle counting. The number of LC3- or LAMP1-positive vesicles was divided by the number of DAPI-stained nuclei to obtain the ratio of autophagosomes and lysosomes, respectively. We manually counted the number of LC3/LAMP1-positive vesicles using the merged image to determine the ratio of mature autophagosomes.

### Statistics and reproducibility
All experiments were repeated at least three times as independent biological replicates with similar results. Representative Western blots or IF images are shown. The number of independent experiments, treatments and relative controls (e.g. DMSO against inhibitor or PBS against growth factors) and statistical analysis are indicated in figure legends or in the appropriate method section with the exception of box plots whose parameters were as follows: minima and maxima = 25 and 75 % quantile, centre = median (50% quantile), box bounds = median + 1.58 × IQR/sqrt(n), whiskers = smallest/largest observation greater/less than or equal to lower/upper box bounds − 1.5 × IQR. Control is indicated as UT in the figures.

### Reporting summary
Further information on research design is available in the Nature Research Reporting Summary linked to this article.

### Data availability
A Source Data file is provided with this paper. The mass spectrometry proteomics data in Thermo Scientific's *.raw format have been deposited to the ProteomeXchange Consortium via the PRIDE[99] partner repository with the following accession codes: PXD028370 (Signalling and recycling endosome_dataset 01), PXD028330 (Signalling and recycling endosome_dataset 02) and PXD028371 (Signalling and recycling endosome_dataset 03). The Uniprot sequence data used in this study are available in the Uniprot database (release 2020-04) under UP000005640 [https://ftp.uniprot.org/pub/databases/uniprot/previous_releases/release-2020_04/knowledgebase/]. The processed mass spectrometry proteomics data generated in this study are provided as Supplementary Data files. Source data are provided with this paper.

### Code availability
Code has been uploaded to GitHub (https://github.com/JoWatson2011/APEX2_Analysis_Watson_Ferguson_2022 and https://doi.org/10.5281/zenodo.7197969).

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

## Acknowledgements

We thank Professor Martin Lowe and Professor Alan Whitmarsh, for reading the manuscript and the Bioimaging and the Bio-MS Facilities (University of Manchester). We thank Dr. A. Badrock, Hurlstone and Wilcock for the reagents and all members of Francavilla's team for discussion. Research in CF lab is supported by the Wellcome Trust (107636/Z/15/Z and 107636/Z/15/A to C.F.), the Biotechnology and Biological Sciences Research Council (BB/R015864/1 to C.F. and J.-M.S.) and Medical Research Council (MR/T016043/1 to C.F. and J.-M.S.). PhD students are supported by BBSRC Doctoral Training Programme (H.R.F. and J.W.: BB/M011208/1). For the purpose of open access, the author has applied a CC BY public copyright licence to any Author Accepted Manuscript version arising from this submission.

## Author contributions

M.P.S. performed experiments, supervised H.R.F., R.M.B. and K.H.B. and wrote the manuscript. H.R.F. and J.W. performed experiments and data analysis. R.M.B., K.H.B. and J.F. performed experiments. H.M. helped with data analysis. P.F. contributed to sample preparation for MS analysis. G.H. performed the FACS analysis. D.K. provided technical advice for the MS experiments. J.-M.S. supervised J.W. and H.M. C.F. conceptualised the study, acquired funding, supervised the work and wrote the manuscript. All the authors contributed to writing and approving the manuscript.

## Competing interests

The authors declare no competing interests.
