## [Peer Review File · Nature Communications]

REVIEWER COMMENTS

Reviewer #1 (Remarks to the Author):

In the submitted manuscript entitled “Spatially Resolved Phosphoproteomics Reveals Fibroblast Growth Factor Receptor Recycling-driven Regulation of Autophagy and Survival”, the authors revealed the functional link between FGFR2b recycling and mTOR/ULK1 signalling-regulated apoptosis through APEX2-based proximity proteome and phosphoproteome profiling. This study established a well-designed system to study FGFR2b recycling by setting up a variety of cell lines: first, the HeLa cell line with exogenous stable FGFR2B expression and the T47D cell line with endogenous FGFR2B expression; second, DnDNM2 and DnRAB11 systems were established by taking advantage of the role of DNM2 and RAB11 in the specific nodes of RTKs recycle process, by which recycling endosomes (REs) are distinguished. The application of these two systems is ingenious. The systematic functional study for validating the autophagy and mTOR/ULK1 signaling is impressive. For the proteomics part, the development and application of spatially resolved phosphoproteomics have drawn great attention recently. For example, TurboID-based approach (Ref. 58) and sequential cell fractionation-based approach (Nat. Commun. 2021, PMID: 34876567) have been reported. Based on similar proximity labeling strategy by selecting APEX2 as the enzyme, the authors claimed the development of APEX2-based spatially resolved phosphoproteomics. However, according to current data analysis and presentation, it's hard to conclude such an effort. The cherry-picking of autophagy, mTOR and phosphorylated ULK1 is quite random. In general, this work is full of high-quality biological validation. However, the success of their spatially resolved phosphoproteomics is questionable.

Major comments:

1. As the authors stated (line 181), “this approach did not reveal which FGFR2b signaling partners were recruited to and specifically phosphorylated in the proximity of the Res during receptor recycling”. Since the main aim of this study was to develop and apply spatially resolved phosphoproteomics, the first part of global phosphoproteome profiling is to support such an effort and should therefore be moved to the latter part or even the supplementary part.
2. In Fig. 1a-b, the authors presented solid imaging data for confirming the successful development of the cell line systems for studying REs. However, in the APEX2-based proximity labeling part in Fig. 2c, the authors didn't adopt the same assay for validating the APEX2-based system. Since this is the most critical part for successful proximity labeling of REs, the authors should present similar data with proper REs controls, such as EEA1.
3. As properly presented in Fig. 2e and Supplementary Fig. 3d, the authors validated the proper APEX2-based proximity labeling and FGF10 stimulation by western blot. In Fig. 3a, the authors presented nice experimental design for proximity proteomics accordingly. However, the authors failed to present pair-wise quantitative comparison between the five experimental setting. Since

APEX2-based proximity labeling could be very noisy, the selectivity of the labeling in comparison to the proper controls is very critical for proximity proteomics and phosphoproteomics. In another word, it is not convincing to simply claim the identification of certain proximal proteins or phosphorylated proteins without properly excluding the background noise in the control panels. The authors have to reprocess their data in a quantitative manner against proper controls as presented in Fig. 3a. Accordingly, the cherry-picking for mTOR signaling and Autophagy should be better supported by the quantitative proximity proteome and phosphoproteome data.

4. In addition, the phosphorylation site data presented in Fig. 4f should also be extended with related quantification data. The authors should also present and explain how the quantitation data could support current conclusion, especially with the nice validation for the S638 p-ULK1 as presented in Fig. 5a-b.

Minor comments:

1. It is interesting to know whether the RAB11 or other signaling partners recruited into the RE locate inside or outside of the recycling endosome, and how to ensure the labeling efficiency considering the existence of endosome membrane.
2. The author mentioned that dominant negative Dynamin (DnDNM2) inhibited FGFR2b internalization in line 123. The author only detect the localization of FGFR2b at 40 min, the recycling time point, but not detect the appropriate time point of internalization. Moreover, from the Fig. 1b, the result showed that dominant negative Dynamin inhibited the colocalization of FGFR2B with RAB11, but has no significant effect on colocalization of FGFR2B with EEA1. Therefore, how the author concluded that negative Dynamin (DnDNM2) inhibited FGFR2b internalization, which will influence the clustering of internalization response in Fig. 2d-e.
3. In Fig. 1c, the results showed that after the FGF10 stimulation, the level of total ERK was also increased, how to explain this alteration?
4. In Fig. 1c-e, the author blotted the pFGFR and pERK1/2 to indicate that impeding FGFR2b trafficking did not alter FGFR2b activation or the phosphorylation of ERK1/2 downstream of FGF10. What is the logic of setting FGF10 stimulation time points? what is the reason for the different time setting with DnDNM2 cells (omitted 1 min time point) compared to another two? And the reason for stimulating time in HeLa cell line is different from another two cell lines is also not described.
5. It is not clear if the author repeats the experimental results in triplicates?
6. In Fig. 3e: Please describe the reason for apparent low level of streptavidin-HRP blot in lane "FGF10 1 min, input" but with equal level in the corresponding pulldown lane.
7. In Fig. 5g: "percentage" in the Y-axis need to be corrected.

Reviewer #2 (Remarks to the Author):

The manuscript by Watson et al. focuses on the spatial specificity of FGFR signaling. The authors used sophisticated proteomics approaches in a combination with molecular cell biology techniques to reveal FGFR2b signaling partners proximal to recycling endosomes. The authors demonstrate that upon FGF10 binding the ligand complexes with FGFR2b are sorted to recycling endosomes, where FGFR2b displays specific signaling activity, affecting mTOR/ULK1 signaling and by this suppressing autophagy and facilitating cell survival. These findings are novel and highly significant for the cell signaling and trafficking field, and in my opinion fully deserve publication in Nature Communications. Data presented are mostly of high quality and support conclusions of the authors. However, there are several important points that require some additional experimental work before recommendation of the manuscript for publication:

Major points:

1. In most of studies focused on autophagy modulation by FGFR2b at RE authors used DnRAB11, where recycling endosomes formation is hampered. Can authors exclude the possibility that autophagy modulation by FGFR2b occurs not at RE stage but already after FGFR2b recycling, when recycled FGFR2b is present at the plasma membrane again (and maybe differs in some way from "fresh" receptor)?
2. The authors should expand time-dependent signaling studies for FGFR2b-regulated proteins (ERK1/2, PLCgamma, FRS2 and identified autophagy proteins) for longer time points (e.g. several time points from few min to few hours) with DnRab11 and DnDnm2 as well as with several siRNA against specific endocytic proteins of CME and CIE to depict precisely FGFR2b signaling events during receptor endocytosis and intracellular trafficking.
3. It would be worthwhile to demonstrate interaction of FGFR2b upon trafficking with identified proteins (e.g. related to autophagy) using proximity-ligation assay (PLA).
4. How unique is autophagy modulation at RE by FGF and FGFR members? Can e.g. FGFR1-FGF1 complex (directed via CME mainly to lysosomes) contribute to autophagy modulation at any trafficking stage? Can FGF10 affect autophagy by acting on FGFR1b?

5. The authors should comment on why FGF10/FGFR2b complexes are recycled and FGF7/FGFR2B not. What is the molecular basis of this phenomenon and possibly physiological implications?

6. The display of Fig. 1 requires revision. Are double columns in Fig.1A (plasma membrane, cytoplasm) specimens +/-FGF10? If yes, the labelling is missing (it is also not clear from the figure legend). Furthermore, separate fluorescent channels should be shown, not only merge, as it is very hard to judge on colocalization based on presented images.

7. Fig. 1B what is N=3. Is it an average colocalization from a three set of cells analyzed in three independent experiments (how big was the individual group of cells?) or just three individual cells from a single experiment? This should be clarified as it is not clear neither from the Fig. legend nor from Methods section.

8. It should be explained in the results section how FGFR2 recycling was measured with confocal microscopy as it is not clear now in the manuscript. The samples after internalization and acidic wash should be shown to demonstrate that after this treatment no cell surface staining is left (indicating non-internalized FGFR2 pool) and cell surface staining results solely from FGFR2 recycling.

9. Fig.1B and lines 121-132: the authors state that expression of DnDNM2 inhibited FGFR2b internalization. However, in Fig. 2B it looks like upon DN2 blockade there is no significantly less (statistics are missing...) colocalization of FGFR2b with EEA1 (light blue bars vs red ones). Is really DN2 blockade inhibiting FGFR2b/FGF10 uptake? Maybe other, CIE independent pathways are activated at these conditions as well, contributing to the internalization of FGFR2b? This point should be clarified. Furthermore, display of top graph in Fig. 1b is misleading – is it colocalization of FGFR2b with Rab11.GFP or DN2.GFP (I guess both). I would suggest making separate graphs for clarity and altering Y axis labelling.

10. Line 134: authors state: “....impeding FGFR2b trafficking did not alter FGFR2b activation or the phosphorylation of ERK1/2 downstream of FGF10 (Fig. 1 c-e)...”. However, especially on Fig. 1d one can see a clear-cut difference in ERK1/2 signaling kinetics, which is reversed upon Rab11 and Dnm2 blockade. This should be explained/commented by the authors? Furthermore, the signaling time (Fig. 1c-e) points should correspond with time points shown for trafficking (Fig. 1 and b). And general point: what is N number (fully independent experiments) for experiments shown in Fig. 1c-e? It is not marked at any point of the manuscript.

11. P7/Line 196:..."and verified that FGFR2b signaling and trafficking were not altered by the presence of APEX2 upon FGF10 stimulation over time (Fig. 3b-d.)". Fig. 3b – a clear differences can be seen in PLCgamma activation for FGFR2b-APEX fusion (largely enhanced signal, fluctuating signal as well) vs FGFR2b. What is a real N (independent experiments) for Fig. 3b? The signaling results should be quantified as it is possible that APEX affected specificity of FGFR2b signaling, thus having an impact on subsequent MS experiments.

12. Fig. 3c – scale bars are missing.

13. Fig. 3d – how exactly quantifications were performed? what is N? how many cells per N were counted?

14. Fig. 3e – what is N for this experiment? Why there is less protein in FGF10 1 min sample? Can authors provide longer exposure for Histone H3 blots (more comparable to SHC signal strength) to firmly judge on no H3 binding?

15. Fig. 5g – how exactly quantification was done? what is N=12 (12 cells? 12 experiments with how many cells per experiment?)

16. Fig. 5h and i – how many times experiment was done and showed the same results? Detection for total beclin is missing (Fig. 5i).

17. Why levels of ULK1 are so dramatically reduced in proximal RAB11-APEX2 (Fig. 6a)? What is N in here?

18. Separate channels in Fig. 6c should be shown and preferably with better resolution/quality as magnified merge images are completely filled with green/blue signals of low resolution, therefore some colocalization (seen in yellow) is not surprising. The images quality should be improved. Scale bar is missing.

19. How quantifications in Fig. 6d have been done? how many cells per N have been measured? The labeling of Y axis should be changed as it is difficult to find out what is exactly measured.

20. Fig 7a and c – how quantifications have been made in detail? what is N?

21. Fig 7c – autophagy/apoptosis/proliferation should be shown on separate graphs as these are different tests and cannot be compared between each other on a single graph.

Minor points:

1. The Figures depicting MS analyses are in my opinion much too much detailed with limited significant contribution to the readability of the whole work for non MS experts. The authors could consider moving parts of MS data to supplement, leaving the most significant ones.

2. Typing errors:

P5/line 120 – vesicles for vescicles

P6/lines 174/175 – there is two times “presence”

P6 line 194 – localized for localize

P11 line 331 – lipidated instead of lapidated (present few times more throughout the text)

Congratulations on your impressive work,

With best regards,

Łukasz Opalinski

Reviewer #3 (Remarks to the Author):

Watson et al. describe an interesting tool to demonstrate spatio-temporal phosphorylation that occurs specifically at the RE. The authors put this into the context of FGFR2b recycling and the downstream effects of perturbed trafficking. The paper provides an interesting tool that could be valuable for distinguishing post-translational modifications dependent on specific localization.

The paper is well written and figures are well presented, however I have a few comments on the trafficking assays and analysis of downstream effects that are used to demonstrate the phenotype and consequences of perturbed trafficking, specifically autophagy.

Minor comments:

There are a number of typos throughout the manuscript, including a range of spellings of “lipidated”

For IF: In figure legends, please specify number of cells analysed from number of independent experiments, and for blots the number of repeats that were quantified (this should also be done), and the representative nature of the blots chosen. Statistical analysis also needs to be performed, and described in the figure legend and indicated on the graphs.

Terminology should be consistent throughout, especially in regard to internalization, cytoplasm, plasma membrane, binding, recycling. Recommend simplifying as one or other in both legends and figures.

Conditions in experiments should be referred to in respect to the experimental controls, and the control should be specified. Where “untreated” there should be a vehicle control, such as PBS or DMSO. Or mock transfected etc.

Labelling of Western blots and IF should be made as efficient and clear as possible. There are also a few errors left over from assembling the figures, and typos in the labelling (Supp Fig 6c).

Graph presentation, especially in respect to x/y axes, should be changed for clarity. It is most informative for the reader to compare CT to treatments in each readout, ie Fig 7c. Present grouped or as separate graphs showing effects of CT vs treatments on autophagy, apoptosis, proliferation. Don't show inset table, simplify labelling to change terms used to quote the target of inhibition, and refer to original inhibitor names in materials and methods or figure legend.

ERK and mTOR activation is tested separately in Figs 1,3, 6 and 7. These pathways are interlinked, and this could be commented on in the discussion.

Major comments:

Short introductory phrases should be included to describe experiments in the results sections, and a short description of what the results show. Some of the blots are large and complex, and the reader would benefit from being guided through what they should focus on. Ie. Immunofluorescence analysis of LC3 puncta in the presence or absence of xyz compared to CT, showed an increase in LC3 puncta, indicating an increase in basal autophagy in Y cells (Fig Xa). This is often absent.

Materials and methods are not clear at times and lacking important information. I see that there are numerous references to a recent publication from the same group, but details for antibodies, and key relevant experimental details must be included.

One of my main concerns for this manuscript is the differentiation between antibody binding, vs recycling. The assay must be explained in more detail in the materials and methods, ie is binding performed at 4C? Is there an acid wash after binding and before fixation for analysis of recycled FGFR2b? In IF figures and description in the results, it is not clear that binding and recycling have been separated, however this is crucial for the experiment. Often, this may be performed by monitoring a modification that the protein undergoes while it is internalized. However in this case, this could also be done by stripping the membrane of non-internalised protein (acid wash) before then monitoring the amount of protein returned to the cell surface. Please provide an explanation of how these two phenotypes are distinguished from one another.

The inhibition of trafficking is not mentioned in the manuscript. Does the FGFR2b become blocked in EEs? Is this what is demonstrated by showing coloc with EEA1 in Fig 1a? Does it become degraded? It would be interesting to see the fate of this FGFR2b in regards to endosome/lysosome localization.

My other main concern is the quality of the IF images shown. Firstly, the figures are not clearly labelled, I believe that their may be CT v FGF10 labelling absent (ie Fig 1a). Images could be clarified by showing individual channels plus overlay, especially in the zoomed regions, which should be included for ALL conditions, including control. Zoomed panels must also be treated in the same way as the main image, not manipulated differently. Throughout the paper, zoomed images are manipulated too much, and the colocalisation is not clear. This is especially evident in Fig6, where it seems that gain/exposure of the images has been pushed too much during post-acquisition processing.

The Lamp/mTOR imaging in Fig 6c/h does not look convincing. What is the IF protocol used? When performing IF for endo/lysosomes it can be tricky, especially when balancing conditions to visualize mTOR and additional markers. The protocol may need to be revisited further to optimize this staining, and show robust mTOR/Lamp1. Steps to optimize should include fixation, permeabilisation, blocking buffer. If optimisation has already been performed, please provide evidence of validation of IF. IF protocols should also be clearly explained in Materials and methods. As in the previous point, this will likely also be clarified by better image representation.

Fig 1a: Please confirm that zooms are consistent in all images. If not, put scale bars on separate images. Images should aim to represent the result shown in the quantification in the clearest way possible through cell choice and the representation. It seems that some of the cells are much smaller in certain conditions. Why is this only in the bottom panel? If this is a true representation, it should be commented on, and this must be taken into consideration in quantification.

Re image quantification. Should be represented as in Fig7, showing individual cell values. Method of quantification must be carefully selected. Please ensure that this method takes into consideration the changes in presence/intensity in the FGFR2b channel, as there is a big difference across conditions.

It is not clear what quantification refers to in Fig 3d. It would be helpful to clarify which cells are shown in Fig 3c and which are quantified, standardize labelling throughout. I presume that Fig 3c upper panel is HeLa_FGFR2BST, while Fig 3c Lower panel is HeLa_FGFR2Bst-APEX2ST. If so, where is T47D...? If not included in main figs, please include imaging in supplementary. It is also unclear how this quantification has been done. Is quant of total FGFR2b done on cells that are not shown? Or is it calculated from PM + internalized? Why is quant here done differently to Fig 1? The quantification does not seem to reflect what is shown in the images. Staining is almost absent in FGF10 40', however quantification shows that this is still about 90% of CT, this doesn't look like a convincing representation.

The measurement of autophagy needs to be performed more precisely. The authors have looked at the acidic compartments of the cells, relating them to lysosomes. Firstly, the acridine orange images should be included in the manuscript. Secondly, to interpret autophagy data properly, autophagic flux must be measured. This should be performed through Western blotting of LC3 I-II conversion, measuring the flux by comparing treated conditions of perturbation (ie FGF10) +/- Bafilomycin A1

which blocks lysosomal degradation. If possible, an autophagy cargo should also be used to provide complementary data to LC3I-II, generic (but not always relevant) p62, or even FGFR2b (which would also contribute to the manuscript and show if FGFR2b is degraded). This should also be performed by IF in parallel, for autophagosomes and lysosomes. Commonly, LC3/Lamp1 co-staining or tandem-LC3 is performed in parallel with the IF conditions. From quantification of LC3 puncta, vs Lamp1 vs LAMP1+LC3 puncta, the number of autophagosomes, mature autolysosomes, and mature autolysosomes in blocked autophagic conditions, can be determined. Together, this will allow the author to demonstrate robustly the effect that the modulations have on autophagic flux.

Point-by-point responses to reviewer's comments.

Our responses to reviewer comments are provided below in **blue font** after each comment and changes to the manuscript are visualized by *italic blue font*.

REVIEWER COMMENTS

Reviewer #1 (Remarks to the Author):

In the submitted manuscript entitled “Spatially Resolved Phosphoproteomics Reveals Fibroblast Growth Factor Receptor Recycling-driven Regulation of Autophagy and Survival”, the authors revealed the functional link between FGFR2b recycling and mTOR/ULK1 signalling-regulated apoptosis through APEX2-based proximity proteome and phosphoproteome profiling. This study established a well-designed system to study FGFR2b recycling by setting up a variety of cell lines: first, the HeLa cell line with exogenous stable FGFR2B expression and the T47D cell line with endogenous FGFR2B expression; second, DnDNM2 and DnRAB11 systems were established by taking advantage of the role of DN M2 and RAB11 in the specific nodes of RTKs recycle process, by which recycling endosomes (REs) are distinguished. The application of these two systems is ingenious. The systematic functional study for validating the autophagy and mTOR/ULK1 signaling is impressive. For the proteomics part, the development and application of spatially resolved phosphoproteomics have drawn great attention recently. For example, TurboID-based approach (Ref. 58) and sequential cell fractionation-based approach (Nat. Commun. 2021, PMID: 34876567) have been reported. Based on similar proximity labeling strategy by selecting APEX2 as the enzyme, the authors claimed the development of APEX2-based spatially resolved phosphoproteomics. However, according to current data analysis and presentation, it's hard to conclude such an effort. The cherry-picking of autophagy, mTOR and phosphorylated ULK1 is quite random. In general, this work is full of high-quality biological validation. However, the success of their spatially resolved phosphoproteomics is questionable.

We thank the reviewer for his/her detailed assessment of our work and we appreciate the reviewers' positive comments on our manuscript. We have addressed the reviewer's concerns about the analysis and presentation of the spatially resolved phosphoproteomics below and changed the manuscript accordingly.

Major comments:

1. As the authors stated (line 181), “this approach did not reveal which FGFR2b signaling partners were recruited to and specifically phosphorylated in the proximity of the Res during receptor recycling”. Since the main aim of this study was to develop and apply spatially resolved phosphoproteomics, the first part of global phosphoproteome profiling is to support such an effort and should therefore be moved to the latter part or even the supplementary part.

We appreciate this observation, but we think that the first part of our study is crucial to define the importance of developing the spatially resolved phosphoproteomics approach. Therefore, instead of moving this part to supplementary we have clarified the message by rewording the text to avoid confusion for the readership.

The text (from line 329) now reads:

Overall, we concluded that FGFR2b likely recruited and specifically phosphorylated signalling partners in the proximity of the recycling endosomes during receptor recycling and that this causes changes in downstream FGFR2b global signalling.

2. In Fig. 1a-b, the authors presented solid imaging data for confirming the successful development of the cell line systems for studying REs. However, in the APEX2-based proximity labeling part in Fig. 2c, the authors didn't adopt the same assay for validating the APEX2-based system. Since this is the most critical part for successful proximity labeling of REs, the authors should present similar data with proper REs controls, such as EEA1.

We thank the reviewer for the suggestion, and we have performed the requested experiment as shown in the updated Fig 3c-d and legend:

d

c FGFR2b (red) internalization (cytoplasm) and FGFR2b recycling (plasma membrane) in HeLa_{FGFR2b-APEX2ST}, expressing eGFP-RAB11a (wtRAB11), dominant negative eGFP-RAB11a_{S25N} (DnRAB11), or dominant negative dynamin-2_{K44A}-eGFP (DnDNM2) (green), and treated with FGF10 for 0 and 40 min or left untreated (control). Early endosome antigen 1 (EEA1) (blue) is a marker for early endosomes²². Scale bar, 5 μ m. Single channels are shown on the right for FGF10-stimulated cells for 0 and 40 min.. White arrowheads indicate co-localization or lack thereof. **d** Quantification of the co-localization of stimulated FGFR2b (red pixels) with GFP-tagged proteins (green pixels) indicated by red-green pixel overlap fraction (top panel). Quantification of the co-localization of FGFR2b (red pixels) with EEA1 (blue pixels) indicated by red-blue pixel overlap fraction (bottom panel). Representative images are shown in c. Values represent median \pm SD from N=3 where we analysed between 2 and 5 cells for each N; * p-value < 0.0005 (students t-test)²²

APEX2 does not alter FGFR2b trafficking as shown by comparing the results of the trafficking and co-localization assay shown in Fig 1a-b and updated Fig 3c-d. We have changed the result section (from line 393) as follows:

To verify whether FGFR2b trafficking was affected by APEX2, we used two well-established confocal-based methods, which allowed us to monitor receptor internalisation and recycling (see Methods) and to quantify FGFR2b-APEX2 co-localization with known markers of trafficking^{13,22,36}. FGF7 induced FGFR2b internalisation followed by receptor degradation, as shown by the lack of staining at the plasma membrane or in the cytoplasm of cells stimulated for 120 min (Supplementary Fig 4b-c), as previously reported^{13,22}. FGF10 induced FGFR2b to gradually disappear from the cell surface, accumulate in the cytoplasm, and recycle back to the plasma membrane in all the tested cell lines (Supplementary Fig 4b-c). Furthermore, FGFR2b co-localized with Rab11 or with Rab11 and EEA1 in HeLa_{FGFR2b-APEX2ST} expressing Rab11 or DnRAB11, respectively, and remained at the plasma membrane in HeLa_{FGFR2b-APEX2ST} expressing DnDNM2 (Fig 3c-d), as previously shown in HeLa_{FGFR2bST} (Fig 1a-b). Altogether this data indicates that APEX2 did not alter FGFR2b trafficking.

3. As properly presented in Fig. 2e and Supplementary Fig. 3d, the authors validated the proper APEX2-based proximity labeling and FGF10 stimulation by western blot. In Fig. 3a, the authors presented nice experimental design for proximity proteomics accordingly. However, the authors failed to presented pair-wise quantitative comparison between the five

experimental setting. Since APEX2-based proximity labeling could be very noisy, the selectivity of the labeling in comparison to the proper controls is very critical for proximity proteomics and phosphoproteomics. In another word, it is not convincing to simply claim the identification of certain proximal proteins or phosphorylated proteins without properly excluding the background noise in the control panels. The authors have to reprocess their data in a quantitative manner against proper controls as presented in Fig. 3a. Accordingly, the cherry-picking for mTOR signaling and Autophagy should be better supported by the quantitative proximity proteome and phosphoproteome data.

We apologise for having failed to clearly explain how we analysed the quantitative MS values of our datasets. Indeed, we took into consideration the background and we performed pairwise comparisons as described in Figure 3a and 4a. We used GFP-APEX2 as a control for the background noise expected in both the proximal proteome and phosphoproteome. When pre-processing our data, we normalised the intensity values of each phosphorylated site identified upon treatment against the intensity value identified in the GFP-APEX2 control of the corresponding timepoint. Furthermore, we used non-stimulated conditions as a control for FGF10-specific effects and cells expressing RAB11-APEX2 as a spatial control as previously described for the GPCRs (Lobingier et al., Cell, 2017). To clarify this crucial point of the spatially resolved phosphoproteomics approach we have added a schematic of the workflow used to analyse the proximal phosphoproteome data in Figure 4e, and details of how all data were analysed in Supplementary Fig. 5k:

Supplementary Fig. 5k.

We have provided details of the MS quantitation of the endosomal/signalling markers, either used for western blot in Figs. 3e or Supplementary Fig 4d or highlighted in the text, in the figure for reviewers below. These values are also included in Supplementary Tables 5 and 6. The heatmaps below clearly demonstrate that the western blot analysis of FGFR2b and RAB11 proximal proteins reproduces the MS quantitation.

Furthermore, we have provided below the visualisation of the MS quantitation of mTOR signalling partners in the proximal phosphoproteome data. These values can also be found in Supplementary Tables 5 and 6. We have previously used similar methods to narrow down candidates from complex MS datasets to be validated by complementary approaches (Francavilla, Mol Cell, 2013; Francavilla, Nat Struct Mol Biol, 2016; Francavilla, Cell reports, 2017; Smith, Ferguson, EMBO J., 2021). In this manuscript, following normalisation against the GFP-APEX2 controls as described above, we identified a cluster of phosphorylated sites on proteins biotinylated by FGFR2b-APEX22 and RAB11-APEX2 that are upregulated following FGF10 stimulation for 40 minutes (Fig. 4f); this cluster, which we name FGFR2b Recycling Proximal Signalling, is enriched for mTOR signalling and autophagy. Furthermore, it can be seen in the heatmap below (figure for reviewer) that of the mTOR pathway participants found in our proximal dataset the majority are upregulated in both FGF2b-APEX2 and RAB11-APEX2 FGF10-treated samples. We aimed to highlight that spatially resolved phosphoproteomics was uncovering signalling events that were diluted in the global phosphoproteome data. Given space requirements, we could not focus on all of the pathways that were specifically enriched in the FGFR2b Recycling Proximal Signalling signature, including Spliceosome, Insulin Signalling and Proteoglycans in Cancer (Fig. 5c). Instead, we focussed on mTOR and autophagy. A key reason for this was that the degradation-inducing ligand of FGFR2b, FGF7, has previously been associated to the promotion of autophagy 24hr after stimulation in keratinocytes. Our data seemed to indicate a contrasting effect following FGF10 stimulation (e.g. through the inhibitory phosphorylation of ULK1). Previous work from our lab and others has indicated that the trafficking route of FGFR2b and other RTKs is important for their regulation of cellular outputs, and we were intrigued by the possibility this may also be true for the regulation of autophagy, a cellular output that is typically associated with signalling from the cell surface and the lysosome. ULK1 was a sensible candidate to investigate this cellular behaviour further due to it being a well-known driver of autophagy and terminal target in the mTOR pathway, with behaviours that could be probed using our experimental models.

To further validate our method, we have validated by western blot other candidates from the network shown in Figure 5f, which further shows our spatially resolved phosphoproteomics approach is able to distinguish between phosphorylated proteins either proximal to the recycling endosome, both proximal to the recycling endosome and global, or not proximal to the recycling endosome (global only). The western blot (now in updated Figure 5g), is presented below:

g immunoblot analysis with indicated antibodies of candidate phosphorylated proteins from subnetwork (Fig. 5f). T47D were transfected with RAB11-APEX2 (T47D_RAB11-APEX2) stimulated with FGF10 for the indicated timepoints. Non proximal and proximal samples represent the supernatant and the pull-down following enrichment of biotinylated samples with streptavidin beads, respectively, and run against total lysates (total).

The results section (from line 675) has been updated:

A subset of candidates within this network, spanning the proximal, global and both proximal and global were confirmed by immunoblot analysis in T47D to match the patterns identified by the SRP approach (Fig. 5g).

4. In addition, the phosphorylation site data presented in Fig. 4f should also be extended with related quantification data. The authors should also present and explain how the quantitation data could support current conclusion, especially with the nice validation for the S638 p-ULK1 as presented in Fig. 5a-b.

We have provided below the visualisation of the normalised and transformed MS quantitation of mTOR signalling partners in the proximal phosphoproteome data. Of the mTOR pathway participants found in our proximal proteome and phosphoproteome data, the majority are upregulated on both the proteome and phosphoproteome level in both FGF2b-APEX2 and RAB11-APEX2 FGF10-treated samples. As these values are found in supplementary tables 5 and 6 (and are a subset of the data visualised in Fig. 4f) we have not included the visualisation in the manuscript.

A full workflow for how our quantitative data was handled can be found in Supplementary Fig 5k and for the proximal phosphoproteome specifically in Fig. 4e, which show how different aspects of our quantitative MS data were handled independently. Crucially, it can be noted that our spatially resolved phosphoproteomics approach results in a shift in MS intensities between proximal and global phosphoproteome samples (Supplementary Fig. 5h). We therefore chose to normalise these two aspects of the data separately, making them not comparable on the quantitative level (i.e. not statistically comparable). This is why we chose not to present the proximal and global MS quantitation together and instead present them separately, in Figs. 4f and 5a respectively. In order to find a point of qualitative comparison, we use data analysis techniques such as clustering to capture patterns in different aspects of our complex dataset. This generated the two signatures we analysed further, which we refer to as FGFR2b Recycling Proximal Signalling and Global signalling and go on to compare qualitatively using functional tests such as KEGG pathway enrichment (Fig. 5c) and protein-protein interaction network analysis (Fig. 5f). The latter is particularly valuable for reducing the complexity of the data, by labelling different phosphorylated sites that were found in one or both of the two signatures. Focussing on the mTOR effector ULK1, we could see that the inhibitory phosphorylation site was part of the proximal signature but not the global, leading us to the hypothesis that ULK1 is locally inhibited in the proximity of FGFR2b at the recycling endosome, leading to autophagy repression.

Taken together, this demonstrates that the results of our functional analysis, which ensue from robust handling of quantitative data, fully support the novelty of our SRP approach and the

conclusion that FGF10:FGFR2b induces mTOR-dependent repression of autophagy at the recycling endosome.

Minor comments:

1. It is interesting to know whether the RAB11 or other signaling partners recruited into the RE locate inside or outside of the recycling endosome, and how to ensure the labeling efficiency considering the existence of endosome membrane.

RAB11 is a small GTPase localized in the cytosol which is recruited to the membrane of recycling endosomes when bound to GTP (Ullrich, *Journal of Biological Chemistry*, 1993; Ullrich, *Journal of Cell Biology*, 1996; Sonnichsen, *Journal of Cell Biology*, 2000). Therefore, RAB11 localizes “outside” of the recycling endosomes and towards the cytoplasm. The construct eGFP-RAB11-APEX2 used in this manuscript allows efficient labelling of RAB11 partners in the cytoplasm. Indeed, we found phosphorylated ULK1 and BAD among RAB11 partners which are cytosolic proteins (Saxton, *Cell*, 2017; Sakamaki, *PNAS*, 2011). FGFR and EGFR are receptor tyrosine kinases expressed on the plasma membrane which are internalized into vesicles through the formation of “membrane invagination”, therefore their cytoplasmic C-terminus is exposed towards the cytoplasm when FGFR and EGFR localize to recycling endosomes (Goh, *Cold Spring Harb Perspect Bio*, 2013). Therefore, FGFR and EGFR can be RAB11 partners at the recycling endosomes and the endosome membrane does not prevent labelling efficiency.

2. The author mentioned that dominant negative Dynamin (DnDNM2) inhibited FGFR2b internalization in line 123. The authors only detect the localization of FGFR2b at 40 min, the recycling time point, but not detect the appropriate time point of internalization. Moreover, from the Fig. 1b, the result showed that dominant negative Dynamin inhibited the colocalization of FGFR2B with RAB11 but has no significant effect on colocalization of FGFR2B with EEA1. Therefore, how the author concluded that negative Dynamin (DnDNM2) inhibited FGFR2b internalization, which will influence the clustering of internalization response in Fig. 2d-e.

The K44A mutant form of the GTPase dynamin2 (Dominant negative dynamin, DnDNM2 in our manuscript) is known to prevent the clathrin-mediated internalization of Receptor Tyrosine Kinases like Epidermal Growth Factor Receptor (EGFR) and FGFR (Vieira, *Science*, 1996; Sigismund, *Dev Cell*, 2008; Smith, *EMBO J*, 2021). We have previously shown that expressing DnDNM2 in the epithelial breast cancer cell line T47D inhibited FGFR2b internalization (Smith, *EMBO J*, 2021). Here, we show the same effect of DnDNM2 in inhibiting FGFR2b internalization in the epithelial cell line HeLa (Fig 1a-b). Fig 1a shows the localization of FGFR2b at 0 and 40 min in unstimulated cells or upon stimulation with FGF10 in three different experimental conditions. When cells express either Rab11-GFP or mutated Rab11-GFP (wtRAB11 and DnRAB11 in our manuscript) FGFR2b (red) localizes at the plasma membrane at time 0 and 40 min in untreated cells and at time 0 in stimulated cells as no signals was detected in the cytoplasm (right panels in Fig 1a), whereas FGFR2b localizes in RAB11- or DnRAB11-positive compartments in cells stimulated for 40 min with FGF10 as the red signal was detected only in the cytoplasm (right panel). In cells expressing RAB11, FGFR2b and RAB11 co-localize (yellow) which indicates that FGFR2b has been internalized and is in the recycling endosomes, as previously reported (Belleudi, *Traffic*, 2007, Francavilla, *Mol Cell*, 2013). However, in cells expressing DnRAB11, FGFR2b co-localizes with DnRAB11 and also with EEA1 (blue), a marker of early endosomes, which indicates receptor internalization (Francavilla, *J Cell Biol*, 2009). The white signal indicates that FGFR2b has been internalized but it is not in the recycling compartment. On the other hand, when cells express DnDNM2 (green in the eight bottom panels of Fig 1a), FGFR2b (red) is detected at the plasma membrane in all conditions and never detected in the cytoplasm, thus indicating lack of internalization. Furthermore, there is no co-localization between FGFR2b and the marker of internalization EEA1 in the cytoplasm. The results of this experiment are quantified in Fig 1b,

which does not show the co-localization of FGFR2b with RAB11 in DnDNM2-expressing cells (green) and shows no co-localization of FGFR2b with either DnDNM2 or EEA1. This indicates that DnDNM2 does prevent FGFR2b internalization. In conclusion, the interpretation of Fig 2d-e is correct.

We have changed the result (from line 158) as follow:

We transiently expressed (more than 80% of positive cells) Dynamin_K44A-eGFP (dominant negative Dynamin, DnDNM2) or eGFP-RAB11_S25N (dominant negative RAB11, DnRAB11), which are known to inhibit FGFR2b internalization and recycling to the plasma membrane respectively, in response to FGF10 stimulation for 40 min²². At this time point FGFR2b was localized in the recycling endosomes in cells expressing wild-type e-GFP-RAB11 (wild-type RAB11, wtRAB11) (Fig. 1a-b)^{13,22}. We also stimulated cells with FGF10 for 120 min to study the fate of FGFR2b at a longer time point. As shown for FGFR1³⁶, FGFR2b co-localized with the marker of early endosomes EEA1, and with DnRAB11 in cells expressing DnRAB11 and was not found at the plasma membrane upon 40 and also 120 min stimulation with FGF10 (Fig. 1a) These findings suggest that FGFR2b is trapped in EEA1/DnRAB11-positive vesicles. When cells express DnDNM2 (green in the eight bottom panels of Fig 1a), FGFR2b (red) was detected at the plasma membrane at all time points and was never detected in the cytoplasm, thus indicating lack of internalization. Furthermore, there was no co-localization between FGFR2b and the marker of early endosomes EEA1 in the cytoplasm. The results of this experiment are quantified in Fig 1b. In conclusion, expressing DnDNM2 and DnRAB11 impair FGFR2b trafficking and will be used here to study trafficking-dependent changes in FGFR2b signalling in response to FGF10.

Fig 1b has been also updated to include 120 min time point as Fig 1a-b (see next page):

Fig. 1. FGFR2b activation is not affected by receptor sub-cellular localization. **a** FGFR2b (red) internalization (cytoplasm) and FGFR2b recycling (plasma membrane) in HeLa cells stably transfected with FGFR2b-HA (HeLa_FGFR2bST), expressing eGFP-RAB11a (wtRAB11), dominant negative eGFP-RAB11a_S25N (DnRAB11), or dominant negative dynamin-2_K44A-eGFP (DnDNM2) (green), and treated with FGF10 for 0, 40 and 120 min or left untreated (control). Early endosome antigen 1 (EEA1) (blue) is a marker for EEs²². Scale bar, 5 μ m. Single channels are shown on the right for FGF10-stimulated cells for 0, 40 and 120 min.. White arrowheads indicate co-localization or lack thereof. **b** Quantification of the co-localization of stimulated FGFR2b (red pixels) with GFP-tagged proteins (green pixels) indicated by red-green pixel overlap fraction (left panel). Quantification of the co-localization of FGFR2b (red pixels) with EEA1 (blue pixels) indicated by red-blue pixel overlap fraction (right panel). Representative images are shown in 1a. Values represent median \pm SD from N=3 where we analysed between 2 and 5 cells for each N; *** p-value < 0.0005 (students t-test)²².

The results (from line 154) have been updated:

We also stimulated cells with FGF10 for 120 min to study the fate of FGFR2b at a longer time point. As shown for FGFR1³⁶, FGFR2b co-localized with the marker of early endosomes EEA1, and with DnRAB11 in cells expressing DnRAB11 and was not found at the plasma membrane upon 40 and also 120 min stimulation with FGF10 (Fig. 1a).

3. In Fig. 1c, the results showed that after the FGF10 stimulation, the level of total ERK was also increased, how to explain this alteration?

We have included a more representative image of this western blot; this was an effect of re-probing for total ERK over phosphorylated ERK. The replacement in Fig. 1c is as follows:

4. In Fig. 1c-e, the author blotted the pFGFR and pERK1/2 to indicate that impeding FGFR2b trafficking did not alter FGFR2b activation or the phosphorylation of ERK1/2 downstream of FGF10. What is the logic of setting FGF10 stimulation time points? what is the reason for the different time setting with DnDNM2 cells (omitted 1 min time point) compared to another two? And the reason for stimulating time in HeLa cell line is different from another two cell lines is also not described.

As previously reported (Francavilla, Mol Cell, 2013; Smith, EMBO J, 2021), we chose early time points (up to 40 min) to assess early signalling activation and potential changes due to overexpression of either trafficking proteins or of FGFR2b. The 40 min time point was chosen to assess whether signalling activation changed upon changes in FGFR2b localization. Results in Fig 1a-c clearly show that the localization of FGFR2b does not affect FGFR2b activation or ERK phosphorylation in HeLa cells. Due to technical reasons we had to omit one time point in Supplementary Fig 1a-b. However, we reported the results of stimulation for 1 min in DnDNM2-expressing cells and for 40 min in RAB11-, DnRAB11- and DnDNM2-expressing cells in Fig 6f-g where no significant differences were observed in any of the cell lines. In conclusion, altering FGFR2b trafficking did not alter FGFR2b activation or the phosphorylation of ERK1/2 downstream of FGF10.

We have clarified this point in the result section (lines 175) as follow:

Immunoblot analysis of cells stimulated for early time points (up to 40 min) with FGF10 to replicate the trafficking assay showed that impeding FGFR2b trafficking did not alter FGFR2b activation or the phosphorylation of ERK1/2 downstream of FGF10 (Fig. 1c, Supplementary Fig 1)37.

5. It is not clear if the author repeats the experimental results in triplicates?

We confirm this in the Methods section (from line 1613):

All experiments were repeated at least three times with similar results. Representative Western blot are shown. Statistical analysis is indicated in figure legends.

6. In Fig. 3e: Please describe the reason for apparent low level of streptavidin-HRP blot in lane “FGF10 1 min, input” but with equal level in the corresponding pulldown lane.

We have included a different western blot which shows a more evenly distributed streptavidin-HRP across all time points. The new western blot can be found in Fig. 3e as follows:

7. In Fig. 5g: “percentage” in the Y-axis need to be corrected.

Now corresponding to Figure 5h, this has been changed.

Reviewer #2 (Remarks to the Author):c

The manuscript by Watson et al. focuses on the spatial specificity of FGFR signaling. The authors used sophisticated proteomics approaches in a combination with molecular cell biology techniques to reveal FGFR2b signaling partners proximal to recycling endosomes. The authors demonstrate that upon FGF10 binding the ligand complexes with FGFR2b are sorted to recycling endosomes, where FGFR2b displays specific signaling activity, affecting mTOR/ULK1 signaling and by this suppressing autophagy and facilitating cell survival. These findings are novel and highly significant for the cell signaling and trafficking field, and in my opinion fully deserve publication in Nature Communications. Data presented are mostly of high quality and support conclusions of the authors. However, there are several important points that require some additional experimental work before recommendation of the manuscript for publication:

We thank the reviewer for the detailed assessment of our work, and we appreciate the reviewers’ positive comments on our manuscript. We have answered to reviewers’ concerns below.

Major points:

1. In most of studies focused on autophagy modulation by FGFR2b at RE authors used DnRAB11, where recycling endosomes formation is hampered. Can authors exclude the possibility that autophagy modulation by FGFR2b occurs not at RE stage but already after

FGFR2b recycling, when recycled FGFR2b is present at the plasma membrane again (and maybe differs in some way from "fresh" receptor)?

We show that phosphorylation of ULK1 at S638, an inhibitory site of ULK1 activation and therefore a negative regulator of autophagy, occurs proximal to the recycling endosomes, indicated by experiments in Figures 6a and 6b at 40 minutes, a time-point at which FGFR2b is internalized following activation, and downstream of FGF10 is accumulated in the recycling endosomes (Fig 1a, 3c, Supplementary Fig 4c). We also show using inhibitor of recycling, dominant-negative RAB11 (Figure 6d-6h; Figure 7b) that when recycling is inhibited, we did not see FGF10-mediated pULK1_S638 at 40 minutes, further supporting autophagy as an FGFR2b recycling endosome proximal signature. Finally, we showed the same lack of ULK phosphorylation in cells treated with trafficking inhibitors (Supplementary Fig 8d) and in cells depleted of two regulators of FGFR2b recycling, RCP and TTP (Supplementary 8c). Based on our recent publication Smith et al, EMBO J, 2021, RCP and TTP/SH3BP4 regulate exit and entry of FGFR2b in the recycling endosomes respectively. Altogether, these data, although they do not exclude the possibility of autophagy modulation also from the plasma membrane after internalization and recycling, strongly suggest that FGF10-FGFR2b-ULK1-mediates autophagy regulation occurs in very close proximity to the recycling endosomes.

The new Supplementary Fig 8c and legend are shown below:

c Immunoblot analysis with the indicated antibodies of T47D cells with siRNA-mediated knockdown of TTP or RCP, compared to siRNA control, treated with FGF10 for indicated time points

2. The authors should expand time-dependent signaling studies for FGFR2b-regulated proteins (ERK1/2, PLCgamma, FRS2 and identified autophagy proteins) for longer time points (e.g. several time points from few min to few hours) with DnRab11 and DnDnm2 as well as with several siRNA against specific endocytic proteins of CME and CIE to depict precisely FGFR2b signaling events during receptor endocytosis and intracellular trafficking.

The timeframe explored by this manuscript has been restricted to these early events (less than 4h) as this represents the window of time where localised phosphorylation events can lead to direct changes due to signalling. After these early events it becomes impossible to directly link the localised phosphorylation changes, restricted to recycling, to phenotypic effects due to the expansion of post signalling events such as altered gene expression post transcription as seen in Supplementary Fig. 8i. Indeed, we have started to explore these longer-term changes where we have seen a switch from suppression of autophagy to promotion of autophagy following FGF10 treatment from 24h-72h (figure below, left panel). The reason for these changes may

be linked to the early expression changes in metabolic enzymes seen in Supplementary Fig. 8i. For example, when looking at ATP as a global read out of metabolic activity we observe a shift after 4h treatment (figure below, right panel). This is likely a result of altered metabolism and gene expression mediated regulation of autophagy, and most likely mTOR, rather than early signaling events investigated in this manuscript. Although we certainly agree with the view that these events are both interesting and need exploring further, this would be out of the remit of this manuscript where we focused on the development of the SRP approach to assess how locally restricted phosphorylation events results in immediate and measurable cell behaviour. We provide one of these unpublished assays for the reviewer below:

FGFR2b is known to be internalized through CME in response to its two ligands, FGF7 and FGF10 (Belleudi, Traffic, 2007), therefore we did not include any inhibition of the CME or CIE to link FGFR2b signaling events during receptor endocytosis and intracellular trafficking. However, we further characterised the recycling dependency of FGF10-FGFR2b regulation of autophagy, and we used siRNA mediated knockdown of proteins TTP and RCP, which were previously shown to inhibit recycling of FGFR2b (Smith et al., EMBO J, 2021). Our data show that when recycling is inhibited through inhibition of recycling adaptors of FGFR2b, we no longer see phosphorylation of ULK1 at S638, as shown now in Supplementary Figure 8c. Western blot and updated figure legend can be found as follows:

c Immunoblot analysis with the indicated antibodies of T47D cells with siRNA-mediated knockdown of TTP or RCP, compared to siRNA control, treated with FGF10 for indicated time points

The results section has been updated as follows (from line 839):

Indeed, we did not visualize any ULK1 phosphorylated on S638 when FGFR2b recycling was impaired by expressing DnRAB11 (Fig. 6d-e), or when FGFR2b recycling was inhibited through siRNA-mediated knockdown of the FGFR2b-specific recycling adaptors TTP or RCP (Supplementary Fig. 8c).

3. It would be worthwhile to demonstrate interaction of FGFR2b upon trafficking with identified proteins (e.g. related to autophagy) using proximity-ligation assay (PLA).

We have performed a PLA to show increased interaction between FGFR2b and S638 phosphorylated ULK1 in T47D cells in response to FGF10 stimulation for 40 minutes when compared to untreated T47D cells.

The results have been updated; quantification of PLA can be found in Figure 6c, representative images can be in Supplementary Figure 8b, as shown below:

c Quantification of proximity ligation assay (PLA) puncta between FGFR2b and S638 pULK1 in HeLa_FGFR2bST cells treated with FGF10 40 mins compared to untreated (UT); p -value < 0.0005 *** (Students t -test)

b Representative images corresponding to quantification in Fig. 6c, from proximity ligation assay between FGFR2b and S638 pULK1 (green) in T47D cells treated with FGF10 compared to untreated (UT).

We have updated the results section (from line 829):

In both HeLa-FGFR2bST and T47D cells ULK1 is recruited and phosphorylated on S638 in proximity of both FGFR2b and RAB11 as shown upon streptavidin beads enrichment of biotinylated proteins followed by western blot (Figure 6a-b, Supplementary Fig 8a). We confirmed that FGFR2b and phosphorylated ULK are in close proximity using the Proximity Ligation Assay (Fig. 6c and Supplementary Fig. 8b).

The methods have been added (from line 1657):

Proximity Ligation Assay

Proximity Ligation Assay (PLA) was performed using Duolink® In Situ starter kit Mouse/Rabbit (Sigma-Aldrich, DUO92101) following manufacturer's instructions, to assess co-localization between HA-FGFR2b and pULK1_S638. Briefly, T47D cells were transfected with FGFR2b_HA and 10,000 cells seeded on IBIDI slide. Cells were serum starved for 24h and placed on ice for 30 min. anti-HA was incubated for 40 min on ice, removed and replaced with serum-free media containing 100 ng/mL FGF10 for 40 min at 37 °C. Cells were then fixed with 4% formaldehyde and permeabilised in 0.02% saponin. pULK1 S638 antibody was incubated in Duolink® antibody diluent for 1h at RT, washed and incubated with Duolink® PLUS/MINUS probes for 1h at 37 °C. Cells were then washed and incubated with Duolink® ligase solution for 30 min at 37 °C, washed and incubated with polymerase for 100 min at 37 °C. Cells were washed a final time, then mounted using DuoLink® In Situ mounting media containing dapi. Cells were imaged using a Zeiss Axio Imager upright fluorescence microscope and images analysed using ImageJ.

4. How unique is autophagy modulation at RE by FGF and FGFR members? Can e.g. FGFR1-FGF1 complex (directed via CME mainly to lysosomes) contribute to autophagy modulation at any trafficking stage? Can FGF10 affect autophagy by acting on FGFR1b?

We thank the reviewer for this suggestion. We have compared autophagy in HeLa wild-type cells (FGFR2b negative; FGFR1b positive based on Francavilla et al, JCB, 2009) to autophagy in HeLa-FGFR2bST. We show that FGF10-FGFR1b signaling does not inhibit autophagy in comparison to untreated cells, and that FGF1-activation of FGFR significantly inhibits autophagy, as previously shown (Cinque et al., Nature, 2015).

The results are included in Supplementary Fig. 6c:

c Quantification of autophagy, assessed by acridine orange, comparing HeLa and HeLa_FGFR2bST cells treated with indicated ligands for 2h; p-value < 0.0005 ***, ANOVA post-hoc Tukey.

The updates results section (from line 706) now reads:

To assess the dependency of the autophagy response on FGF10-FGFR2b, rather than FGF10-FGFR1b we compared HeLa (with endogenous FGFR1b expression)³⁶ and HeLa_FGFR2bST, showing that only in the presence of FGF10 and FGFR2b was autophagy reduced (Supplementary Fig 6c). We also confirmed the effect of FGF7 and of another ligand for FGFR2b, FGF1, on regulating autophagy⁴² (Supplementary Fig 6c).

5. The authors should comment on why FGF10/FGFR2b complexes are recycled and FGF7/FGFR2B not. What is the molecular basis of this phenomenon and possibly physiological implications?

We and others have previously reported that FGF10 induces FGFR2b recycling whereas FGF7 induces FGFR2b degradation (Belleudi, Traffic, 2007; Francavilla, Mol Cell, 2013; Smith, EMBO J, 2021). We have also shown that FGF10 induces a specific phosphorylation pattern on FGFR2b cytoplasmic domain (Y734 phosphorylation) which in turns recruits the recycling adaptor protein TTP/SH3BP4 to FGFR2b, thus regulating FGFR2b entry into the recycling endosomes (Francavilla, Mol Cell, 2013; Smith, EMBO J, 2021). As FGF7 does not induce FGFR2b phosphorylation on Y734, TTP is not recruited and FGFR2b is sorted to late endosomes instead to recycling endosomes. Once in the recycling endosomes the adaptor protein RCP/RAB11FIP1 allows exit of FGFR2b from the recycling endosomes (Smith, EMBO J, 2021). When FGFR2b is in the recycling endosomes cells proliferate and migrate more (Francavilla, Mol Cell, 2013; Smith, EMBO J, 2021). These results have clear physiological implications because they form the foundation of why different ligands binding to and activating the same receptor induce differential signalling outputs and physiological responses.

We have made a comment on this in the introduction (from line 108):

The FGFR family is a useful model for studying the contribution of trafficking to signalling outputs²³. There are four FGFRs, with FGFR1-3 having splice-variants denoted as b and c isoforms, and 21 FGF ligands, with each FGFR/FGF pair regulating signalling specificity in a context-dependent manner during development, in maintaining adult homeostasis, and in several diseases such as cancer^{24,25}. One stark example of such functional selectivity is given by FGFR2b which is expressed on epithelial cells²⁴⁻²⁶. Stimulation of FGFR2b with FGF7 induced receptor degradation in contrast to stimulation with FGF10 which resulted in recycling of FGFR2b via RAB11-positive recycling endosomes^{13,22,27}. These two different trafficking routes of FGFR2b were associated with different phosphorylation dynamics within the signalling cascade and an increase in cell proliferation and proliferation/migration, respectively^{13,22}. Therefore, the duration and location of FGFR signalling must be strictly regulated to modulate the appropriate cellular outputs^{23,25}.

6. The display of Fig. 1 requires revision. Are double columns in Fig.1A (plasma membrane, cytoplasm) specimens -/+FGF10? If yes, the labelling is missing (it is also not clear from the figure legend). Furthermore, separate fluorescent channels should be shown, not only merge, as it is very hard to judge on colocalization based on presented images.

We have updated Fig 1a and Fig 1a legend:

Fig. 1. FGFR2b activation is not affected by receptor sub-cellular localization. *a* FGFR2b (red) internalization (cytoplasm) and FGFR2b recycling (plasma membrane) in HeLa cells stably transfected with FGFR2b-HA (HeLa_FGFR2bST), expressing eGFP-RAB11a (wtRAB11), dominant negative eGFP-RAB11a_S25N (DnRAB11), or dominant negative dynamin-2_K44A-eGFP (DnDNM2) (green), and treated with FGF10 for 0, 40 and 120 min or left untreated (control). Early endosome antigen 1 (EEA1) (blue) is a marker for EEs²². Scale bar, 5mm. Single channels are shown on the right for FGF10-stimulated cells for 0, 40 and 120 min.. White arrowheads indicate co-localization or lack thereof.

7. Fig. 1B what is N=3. Is it an average colocalization from a three set of cells analyzed in three independent experiments (how big was the individual group of cells?) or just three individual cells from a single experiment? This should be clarified as it is not clear neither from the Fig. legend not from Methods section.

We apologise for the missing information. We have quantified three independent experiments and between 2 and 5 cells for experiment as previously reported (Smith, EMBO J, 2021). We

have added details on how we performed the quantification in the Methods section (from line 1758) by summarizing the steps used in our recent publication (Smith, EMBO J, 2021):

Quantification of FGFR2b recycling, co-localization (pixel overlap fraction), and Expression Fraction (pixel proportion) was performed as recently described in detail²². Briefly, quantification of internalization and recycling was performed as follows. For each time point and each treatment, the presence (total) and the localization (cell surface versus internalized) of HA-FGFR2b or endogenous FGFR2b were assessed in at least seven randomly chosen fields. Approximately 100 cells per condition (both acidic-washed and not) were analyzed from three independent experiments. The results are expressed as the percentage of receptor-positive cells (green) over total cells (corresponding to DAPI-stained nuclei) and referred to the values obtained at time zero. Statistical analysis was performed across repeats, as indicated in the figure legends.

We have added this information in the figure legend of Fig.1b:

*Quantification of the co-localization of stimulated FGFR2b (red pixels) with GFP-tagged proteins (green pixels) indicated by red-green pixel overlap fraction (left panel). Quantification of the co-localization of FGFR2b (red pixels) with EEA1 (blue pixels) indicated by red-blue pixel overlap fraction (right panel). Representative images are shown in 1a. Values represent median \pm SD from N=3 where we analysed between 2 and 5 cells for each N; *** p-value < 0.0005 (students t-test)²².*

8. It should be explained in the results section how FGFR2 recycling was measured with confocal microscopy as it is not clear now in the manuscript. The samples after internalization and acidic wash should be shown to demonstrate that after this treatment no cell surface staining is left (indicating non-internalized FGFR2 pool) and cell surface staining results solely from FGFR2 recycling.

The FGFR2b internalization and recycling assay shown in the updated Supplementary Fig. 4b-4c is a well-established assay (e.g. Di Guglielmo, Nat Cell Biol, 2003) which the authors have performed and quantified several times (Francavilla, J Cell Biol, 2009; Francavilla, Mol Cell, 2013; Smith, EMBO J, 2021). In each figure, the right panels labelled as “cytoplasm” show the samples after acidic wash, fixation and permeabilization, whereas panels labelled as “plasma membrane” show samples before acidic wash and not permeabilized. The surface staining in FGF10-stimulated cells for 120 min indicate FGFR2b recycling as there is no staining in the corresponding acidic washed cells on the right. This is reproduced in Fig 1 and 3.

We have clarified this in the result section (from lines 393):

To verify whether FGFR2b trafficking was affected by APEX2, we used two well-established confocal-based methods, which allowed us to monitor receptor internalisation and recycling (see Methods) and to quantify FGFR2b-APEX2 co-localization with known markers of trafficking^{13,22,36}. FGF7 induced FGFR2b internalisation followed by receptor degradation, as shown by the lack of staining at the plasma membrane or in the cytoplasm of cells stimulated for 120 min (Supplementary Fig 4b-c), as previously reported^{13,22}. FGF10 induced FGFR2b to gradually disappear from the cell surface, accumulate in the cytoplasm, and recycle back to the plasma membrane in all the tested cell lines (Supplementary Fig 4b-c). Furthermore, FGFR2b co-localized with Rab11 or with Rab11 and EEA1 in HeLa_FGFR2b-APEX2ST expressing Rab11 or DnRab11, respectively, and remained at the plasma membrane in HeLa_FGFR2b-APEX2ST expressing DnDNM2 (Fig 3b-c), as previously shown in HeLa_FGFR2bST (Fig 1a-b). Altogether this data indicates that APEX2 did not alter FGFR2b trafficking.

9. Fig.1B and lines 121-132: the authors state that expression of DnDNM2 inhibited FGFR2b internalization. However, in Fig. 2B it looks like upon DN2 blockade there is no significantly less (statistics are missing...) colocalization of FGFR2b with EEA1 (light blue bars vs red ones). Is really DN2 blockade inhibiting FGFR2b/FGF10 uptake? Maybe other, CIE independent pathways are activated at these conditions as well, contributing to the internalization of FGFR2b? This point should be clarified. Furthermore, display of top graph in Fig. 1b is misleading – is it colocalization of FGFR2b with Rab11.GFP or DN2.GFP (I guess both). I would suggest making separate graphs for clarity and altering Y axis labelling.

As highlighted in the response to reviewer 1 minor point 2, the K44A mutant form of the GTPase dynamin2 (Dominant negative dynamin, DnDNM2 in our manuscript) is known to prevent the clathrin-mediated internalization of Receptor Tyrosine Kinases like Epidermal Growth Factor Receptor (EGFR) and FGFR (Vieira, Science, 1996; Sigismund, Dev Cell, 2008; Smith, EMBO J, 2021). We have previously shown that expressing DnDNM2 in the epithelial breast cancer cell line T47D inhibited FGFR2b internalization (Smith, EMBO J., 2021). Here, we show the same effect of DnDNM2 in inhibiting FGFR2b internalization in the epithelial cell line HeLa (Fig 1a-b). Fig 1a shows the localization of FGFR2b at 0 and 40 min in unstimulated cells or upon stimulation with FGF10 in three different experimental conditions. When cells express either Rab11-GFP or mutated Rab11-GFP (wtRAB11 and DnRAB11 in our manuscript) FGFR2b (red) localizes at the plasma membrane at time 0 and 40 min in untreated cells and at time 0 in stimulated cells as no signals was detected in the cytoplasm (right panels in Fig 1a), whereas FGFR2b localizes in RAB11- or DnRAB11-positive compartments in cells stimulated for 40 min with FGF10 as the red signal was detected only in the cytoplasm (right panel). In cells expressing RAB11, FGFR2b and RAB11 co-localize (yellow) which indicates that FGFR2b has been internalized and is in the recycling endosomes, as previously reported (Belleudi, Traffic, 2007, Francavilla, Mol Cell, 2013). However, in cells expressing DnRAB11, FGFR2b co-localizes with DnRAB11 and also with EEA1 (blue), a marker of early endosomes, which indicates receptor internalization (Francavilla, J Cell Biol, 2009). The white signal indicates that FGFR2b has been internalized but it is not in the recycling compartment. On the other hand, when cells express DnDNM2 (green in the eight bottom panels of Fig 1a), FGFR2b (red) is detected at the plasma membrane in all conditions and never detected in the cytoplasm, thus indicating lack of internalization. Furthermore, there is no co-localization between FGFR2b and the marker of internalization EEA1 in the cytoplasm. The results of this experiment are quantified in Fig 1b, which does not show the co-localization of FGFR2b with RAB11 in DnDNM2-expressing cells, but it shows no co-localization of FGFR2b with either DnDNM2 (green) or EEA1. This indicates that DnDNM2 does prevent FGFR2b internalization. This finding is also in line with data showing that FGFR2b is internalized through CME in response to its two ligands, FGF7 and FGF10 (Belleudi, Traffic, 2007). We have changed the results as follow (from line 159):

We transiently expressed (more than 80% of positive cells) Dynamin_K44A-eGFP (dominant negative Dynamin, DnDNM2) or eGFP-RAB11_S25N (dominant negative RAB11, DnRAB11), which are known to inhibit FGFR2b internalization and recycling to the plasma membrane respectively, in response to FGF10 stimulation for 40 min²². At this time point FGFR2b was localized in the recycling endosomes in cells expressing wild-type e-GFP-RAB11 (wild-type RAB11, wtRAB11) (Fig. 1a-b)^{13,22}. We also stimulated cells with FGF10 for 120 min to study the fate of FGFR2b at a longer time point. As shown for FGFR1³⁶, FGFR2b co-localized with the marker of early endosomes EEA1, and with DnRAB11 in cells expressing DnRAB11 and was not found at the plasma membrane upon 40 and also 120 min stimulation with FGF10 (Fig. 1a) These findings suggest that FGFR2b is trapped in EEA1/DnRAB11-positive vesicles. When cells express DnDNM2 (green in the eight bottom panels of Fig 1a), FGFR2b (red) was

detected at the plasma membrane at all time points and was never detected in the cytoplasm, thus indicating lack of internalization. Furthermore, there was no co-localization between FGFR2b and the marker of early endosomes EEA1 in the cytoplasm. The results of this experiment are quantified in Fig 1b. In conclusion, expressing DnDNM2 and DnRAB11 impair FGFR2b trafficking and will be used here to study trafficking-dependent changes in FGFR2b signalling in response to FGF10.

In Fig 1b we report only the statistically significant results and we have now updated the figure by taking into accounts novel data (120 min stimulation). Furthermore, we have separated the graphs as suggested (see updated Fig 1a above).

10. Line 134: authors state: “...impeding FGFR2b trafficking did not alter FGFR2b activation or the phosphorylation of ERK1/2 downstream of FGF10 (Fig. 1 c-e)...”. However, especially on Fig. 1d one can see a clear-cut difference in ERK1/2 signaling kinetics, which is reversed upon Rab11 and Dnm2 blockade. This should be explained/commented by the authors?

This has been replaced with a more representative image shown below:

Furthermore, the signaling time (Fig. 1c-e) points should correspond with time points shown for trafficking (Fig. 1 and b).

As highlighted in response to reviewer 1 minor point 4, we chose early time points (up to 40 min) to assess early signalling activation and potential changes due to overexpression of either trafficking proteins or of FGFR2b (this manuscript; Francavilla, Mol Cell, 2013; Smith, EMBO J, 2021). The 40 min time point was chosen to assess whether signalling activation changed upon changes in FGFR2b localization. Results in Fig 1a-c clearly show that the localization of FGFR2b does not affect FGFR2b activation or ERK phosphorylation in HeLa cells. Due to technical reasons we had to omit one time point in Fig 1d-e. However, we reported the results of stimulation for 1 min in DnDNM2-expressing cells and for 40 min in RAB11-, DnRAB11- and DnDNM2-expressing cells in Fig. 6f-6g where no significant differences were observed in any of the cell line. In conclusion, altering FGFR2b trafficking did not alter FGFR2b activation or the phosphorylation of ERK1/2 downstream of FGF10.

We have clarified this point in the result section (from lines 175) as follow:

Immunoblot analysis of cells stimulated for early time points (up to 40 min) with FGF10 to replicate the trafficking assay showed that impeding FGFR2b trafficking did not alter FGFR2b activation or the phosphorylation of ERK1/2 downstream of FGF10 (Fig. 1c, Supplementary Fig 1)37.

And general point: what is N number (fully independent experiments) for experiments shown in Fig. 1c-e? It is not marked at any point of the manuscript.“

We confirm that all experiments have been repeated in biological replicates in the Methods section:

“All experiments were repeated at least three times with similar results. Representative Western blot are shown. Statistical analysis is indicated in figure legends.”

11. P7/Line 196:...”and verified that FGFR2b signaling and trafficking were not altered by the presence of APEX2 upon FGF10 stimulation over time (Fig. 3b-d..). Fig. 3b – a clear differences can be seen in PLCgamma activation for FGFR2b-APEX fusion (largely enhanced signal, fluctuating signal as well) vs FGFR2b. What is a real N (independent experiments) for Fig. 3b? The signaling results should be quantified as it is possible that APEX affected specificity of FGFR2b signaling, thus having an impact on subsequent MS experiments.

The updated Fig 3b-d clearly shows that the presence of APEX2 does not alter FGFR2b signalling and trafficking. All the WB experiments have been repeated at least three times. We have included a more representative western blot in Figure 3b, as shown below:

Furthermore, we have performed the same experiment shown in Fig. 1a also using cells expressing FGFR2b-APEX2 and shown the same pattern of trafficking. The results can be found in Figure 3c and Figure 3d, as shown below.

Figure legends have been updated:

(c) FGFR2b (red) internalization (cytoplasm) and FGFR2b recycling (plasma membrane) in HeLa_FGFR2b-APEX2ST, expressing eGFP-RAB11a (wtRAB11), dominant negative eGFP-RAB11a_S25N (DnRAB11), or dominant negative dynamin-2_K44A-eGFP (DnDNM2) (green), and treated with FGF10 for 0 and 40 min or left untreated (control). Early endosome antigen 1 (EEA1) (blue) is a marker for early endosomes²². Scale bar, 5 μ m. Single channels are shown on the right for FGF10-stimulated cells for 0- and 40-min. White arrowheads indicate co-localization or lack thereof. (d) Quantification of the co-localization of stimulated FGFR2b (red pixels) with GFP-tagged proteins (green pixels) indicated by red-green pixel overlap fraction (top panel). Quantification of the co-localization of FGFR2b (red pixels) with EEA1 (blue pixels) indicated by red-blue pixel overlap fraction (bottom panel). Representative images are shown in c. Values represent median \pm SD from N=3 where we analysed between 2 and 5 cells for each N; * p-value < 0.005 (students t-test)²²

The results section (from line 393) has been updated:

To verify whether FGFR2b trafficking was affected by APEX2, we used two well-established confocal-based methods, which allowed us to monitor receptor internalisation and recycling (see Methods) and to quantify FGFR2b-APEX2 co-localization with known markers of trafficking^{13,22,36}. FGF7 induced FGFR2b internalisation followed by receptor degradation, as shown by the lack of staining at the plasma membrane or in the cytoplasm of cells stimulated for 120 min (Supplementary Fig 4b-c), as previously reported^{13,22}. FGF10 induced FGFR2b to gradually disappear from the cell surface, accumulate in the cytoplasm, and recycle back to the plasma membrane in all the tested cell lines (Supplementary Fig 4b-c). Furthermore, FGFR2b co-localized with Rab11 or with Rab11 and EEA1 in HeLa_{FGFR2b-APEX2ST} expressing Rab11 or DnRab11, respectively, and remained at the plasma membrane in HeLa_{FGFR2b-APEX2ST} expressing DnDNM2 (Fig 3b-c), as previously shown in

HeLa_FGFR2bST (Fig 1a-b). Altogether this data indicates that APEX2 did not alter FGFR2b trafficking.

Our quantitative MS data demonstrated that the cellular phosphorylation program was not altered by the APEX2-fusion of target proteins. We prove this by performing statistical testing (Supplementary Fig. 4j and table below) on those phosphorylation sites detected in the global samples collected from each experimental condition. In UT and FGF10 40` treated cells, very few (1 and 96 respectively) significant changes to phosphorylated sites (FDR < 0.05) were induced by APEX2 fusions to FGFR2, GFP and RAB11. This suggests that regulation at phosphorylated sites was driven by FGF10 treatment rather than the APEX2-tagged proteins.

Comparison of global STY	Test	FDR < 0.05
GFP-APEX2 UT, FGFR2-APEX2 UT	T-test + FDR adjustment	1 / 11799 quantified sites
GFP-APEX2 40`, FGFR2-APEX2 40`, RAB11- APEX2 40`	One-way ANOVA + FDR adjustment	96 / 11799 quantified sites

This has been clarified in the results (from line 572) as follows:

Finally, we statistically confirmed that the APEX2 tag did not affect the quantification of the global phosphoproteome (Supplementary Fig. 5j), as expected based on immunoblot analysis with the APEX2 tagged proteins (Fig. 3b). We concluded that the double enrichment of biotinylated proteins and phosphorylated peptides did not impact data quality. Given these results, in subsequent analyses the global and proximal phosphoproteome quantitative data were analysed separately (see Supplementary Fig. 5k for more details).

12. Fig. 3c – scale bars are missing.

We have corrected this mistake.

13. Fig. 3d – how exactly quantifications were performed? what is N? how many cells per N were counted?

Details of the quantification of the internalization/recycling assay which has been performed as published before (Francavilla, J Cell Biol, 2009; Francavilla, Mol Cell, 2013; Smith, EMBO J., 2021), is now included in the Methods section (from line 1758) and reads as follows:

Quantification of FGFR2b recycling, co-localization (pixel overlap fraction), and Expression Fraction (pixel proportion) was performed as recently described in detail²². Briefly, quantification of internalization and recycling was performed as follows. For each time point and each treatment, the presence (total) and the localization (cell surface versus internalized) of HA-FGFR2b or endogenous FGFR2b were assessed in at least seven randomly chosen fields. Approximately 100 cells per condition (both acidic-washed and not) were analyzed from three independent experiments. The results are expressed as the percentage of receptor-positive cells (green) over total cells (corresponding to DAPI-stained nuclei) and referred to the values obtained at time zero. Statistical analysis was performed across repeats, as indicated in the figure legends.

Quantification of Expression Fraction, Overlap Fraction and Co-localization was performed as follow. Images were pre-processed using an “À trous” wavelet band pass filter to reduce the contribution of high frequency speckled noise to the co-localization calculations. Pixel intensities were then normalized from the original 8-bit range [0,255] to [0,1]. To ensure that co-localization was only computed in well-determined regions of interest (ROI), we used the Fiji/ImageJ built-in ROI manager to create and record these regions (minimum two cells and

up to five per biological replicates with N=3). To measure differences in expression over time or between conditions, we computed the fractions of expressed red marker R, green marker G, or far-red marker F. pixels over a region of interest. To quantify the overlap fraction between two (R and G) or three (R, F and G) markers, we first multiplied the (normalized) channel intensities together to compute a new image whose intensity increases to 1 where the markers strongly overlap and decreases or becomes null for non-overlapping pixels. Our overlap fraction coefficient (OF) becomes the fraction of strictly positive pixels in the combined image over the number of pixels in the region of interest. Finally, to quantify the actual level of colocalization between two markers (e.g. R and G), we used the Manders Colocalization Coefficients (MCC) M1 and M2. M1 measures the fraction of the R marker in compartments that also contain the G marker, and M2, the fraction of the G marker in compartments that also contain the R marker. Lower-bound thresholds for pixel intensities were automatically determined using the Costes method. To measure the simultaneous overlap of our three, red, far-red and green markers (R, F, G), we first used the overlap image between marker R and marker F as defined above. We then measured the MCC colocalization parameter of this combined image against a green marker using the MCC formulae above, together with the Costes method to determine the thresholds. The scripts for the quantification of co-localization were written in the Python language and the code for Costes-adjusted MCC was taken verbatim from the CellProfiler code base. We analysed three independent experiments and between 2 and 5 cells for experiment. The Student's t-test was subsequently used to determine the difference in pixel overlap fraction between different experimental conditions in Fig 1 and 6, as indicated in the figure legends.

14. Fig. 3e – what is N for this experiment? Why there is less protein in FGF10 1 min sample? Can authors provide longer exposure for Histone H3 blots (more comparable to SHC signal strength) to firmly judge on no H3 binding?

This has been replaced with a more representative repeat with suggested improvements, as shown below:

15. Fig. 5g – how exactly quantification was done? what is N=12 (12 cells? 12 experiments with how many cells per experiment?)

The quantification is explained in the methods section, briefly it is a population analysis using three 96-wells per N, with approximately 20,000 cells per well.

The methods section (from line 1695) reads:

Populations of cells were assayed for autophagy using 5 mM Acridine Orange (Sigma) for 30 min after which excess was removed by thorough washing with 1X PBS. This fluorophore appears green when diffuse but is shifted to the red end of the spectrum when accumulated in acidic vesicles⁹². As such, excitation/emission wavelengths of 500/526 nm were used to measure intensity of diffuse acridine orange (non-specific) and 460/650 nm to assess autophagic staining. The ratio of these values represents stained autophagosomes. Statistical analysis was performed at the endpoint across repeats, as indicated in the Figure legends.

16. Fig. 5h and i – how many times experiment was done and showed the same results? Detection for total beclin is missing (Fig. 5i).

Each experiment in the manuscript been repeated three times, and for western blots a representative blot shown. Total beclin1 has been added to Figure 5j-k as suggested.

17. Why levels of ULK1 are so dramatically reduced in proximal RAB11-APEX2 (Fig. 6a)? What is N in here?

Every experiment has a minimum of N = 3. ULK1 levels are reduced proximal to RAB11-APEX2 in untreated and 8-minute stimulated cells as ULK1 is recruited to the recycling endosome in response to FGF10-mediated FGFR2b recycling. To make this clear, we have changed the results text (from line 829) as follows:

In both HeLa-FGFR2bST and T47D cells ULK1 is recruited and phosphorylated on S638 in proximity of both FGFR2b and RAB11 as shown upon streptavidin beads enrichment of biotinylated proteins followed by western blot (Figure 6a-b, Supplementary Fig 8a). We confirmed that FGFR2b and phosphorylated ULK are in close proximity using the Proximity Ligation Assay (Fig. 6c and Supplementary Fig. 8b).

18. Separate channels in Fig. 6c should be shown and preferably with better resolution/quality as magnified merge images are completely filled with green/blue signals of low resolution, therefore some colocalization (seen in yellow) is not surprising. The images quality should be improved. Scale bar is missing.

We have revised Fig 6 and move part of the results in Supplementary Fig 6. Separate channels are shown as suggested by the reviewer and, for each experimental condition, scale bars have been added in one of the panels. Quantification has been performed on cells from three independent experiments, but only one representative image has been selected for the

main/supplementary figures. The updated figures and figure legends can be found as shown below:

Fig. 6. Phosphorylated ULK1 recruitment at the recycling endosomes depends on FGFR2b recycling. *a, b* Immunoblot analysis ($N \geq 3$) with the indicated antibodies of HeLa-FGFR2bST-RAB11-APEX2 (*a*) or T47D transfected with RAB11-APEX2 (T47D_D-RAB11-APEX2) (*b*) stimulated with FGF10 for the indicated timepoints. Non proximal

and proximal samples represent the supernatant and the pulldown following enrichment of biotinylated samples with streptavidin beads, respectively, and run against total lysates (total).

c Quantification of proximity ligation assay (PLA) puncta between FGFR2b and S638 pULK1 in HeLa_FGFR2bST cells treated with FGF10 40 mins compared to untreated (UT); p -value < 0.0005 *** (Students t -test)

d Co-localization of FGFR2b-APEX2 (red) with phosphorylated ULK1 on S638 (blue) in T47D_FGFR2^{KO}_FGFR2b-APEXST transfected with RAB11 or GFP-DnRAB11 (green) and stimulated or not with FGF10 for 40 min as indicated. Scale bar, 5 μ m. The white arrowhead indicates co-localization or lack thereof.

e Quantification of the co-localization of FGFR2b (red pixels) with GFP-tagged proteins (green pixels) indicated by red-green pixel overlap fraction (top panel), of the co-localization of FGFR2b (red pixels) with GFP-tagged proteins (green) and with phosphorylated ULK1 (blue pixels) indicated by red-green-blue pixel overlap fraction (middle panel). The presence of phosphorylated ULK1 was determined by pixel proportion (see Methods) (bottom panel). Representative images are shown in 6c. Values represent median \pm SD from $N=3$ where we analysed between 2 and 5 cells for each N ;²² p -value < 0.005 **; p -value < 0.0005 *** (Students t -test).

f, g. Immunoblot analysis ($N \geq 3$) with the indicated antibodies of HeLa_FGFR2bST (e) or T47D (f) transfected either with wtRAB11, DnRAB11, or DnDNM2 and left either untreated (UT) or treated with FGF10 for the indicated time points.

h Quantification of the co-localization of FGFR2b (red pixels) with GFP-tagged proteins (green pixels) indicated by red-green pixel overlap fraction (first panel), of the co-localization of FGFR2b (red pixels) with GFP-tagged proteins (green) and with TTP (blue pixels) indicated by red-green-blue pixel overlap fraction (second panel), of the presence of TTP determined by pixel proportion (see Methods) (third panel). Representative images are shown in Supplementary Fig 8f.

i Quantification of the co-localization of LAMP1 (red pixels) with TTP (blue pixels) indicated by red-blue pixel overlap fraction (first panel) and of the co-localization of GFP-tagged proteins (green) with TTP (blue pixels) indicated by green-blue pixel overlap fraction (second panel). Representative images are shown in Supplementary Fig 8g. Values represent median \pm SD from $N=3$ where we analysed between 2 and 5 cells for each N ; *** p -value < 0.0005 (Student t -test).

j Quantification of the co-localization of LAMP1 (red) with mTOR (blue) Indicated by red-blue pixel overlap fraction (right panel) and of the co-localization of GFP-tagged proteins (green) with mTOR (blue pixels) indicated by green-blue pixel overlap fraction (left panel). Representative images are shown in Supplementary Fig 8h. Values represent median \pm SD from $N=3$ where we analysed between 2 and 5 cells for each N ; *** p -value < 0.0005 (Student t -test). *** p -value < 0.0005 (Students t -test).

k Immunoblot analysis with indicated antibodies of T47D transfected with GFP or DnRAB11 and left either untreated (UT) or treated with FGF10 for the indicated time points.

Supplementary Fig. 8 *FGFR2b* regulates *mTOR* and *ULK1* signalling from the REs. **a** Immunoblot analysis ($N \geq 3$) with the indicated antibodies of HeLa *FGFR2b-APEX2ST* (left) and *T47D_FGFR2b^{KO}-FGFR2b-APEX2ST* (right). Non proximal and proximal samples

represent the supernatant and the pull-down following enrichment of biotinylated samples with streptavidin beads, respectively, and run against total lysates (total). **b** Immunoblot analysis with the indicated antibodies of T47D cells with siRNA-mediated knockdown of TTP or RCP, compared to siRNA control, treated with FGF10 for indicated time points **c** Immunoblot analysis with the indicated antibodies of HeLa_FGFR2bST cells pre-treated with primaquine or Dynasore for 2 h followed by stimulation with FGF10 for the indicated time points. **d** Immunoblot analysis with the indicated antibodies of T47D cells transfected with either GFP or DnRAB11 and stimulated with FGF7 for the indicated time points. **e** Representative images corresponding to quantification in Fig. 6c, from proximity ligation assay between FGFR2b and S638 pULK1 (green) in T47D cells treated with FGF10 compared to untreated (UT). **f** Co-localization of FGFR2b or LAMP1 (red) with TTP (blue) in T47D_FGFR2^{KO}_FGFR2b-APEXST transfected with wtRAB11 or DnRAB11 (green) and stimulated or not with FGF10 for 40 min as indicated. Scale bar, 5 μ m. The white arrowhead indicates co-localization or lack thereof. **g, h** Co-localization of LAMP1 (red) with TTP (blue) (**g**) mTOR (blue) (**h**) in T47D_FGFR2^{KO}_FGFR2b-APEXST transfected with wtRAB11 or DnRAB11 (green) and stimulated or not with FGF10 for 40 min as indicated. Scale bar, 5 μ m. The white arrowhead indicates co-localization or lack thereof. **i** Expression of indicated genes in HeLa FGFR2b or T47D transfected with wtRAB11, DnRAB11 or DnDNM2 or pre-incubated with rapamycin for 2 h followed by stimulation with FGF10 for 4 h. qPCR data are presented as heat map from N= 3.

We have changed the result section (from line 826) as follows:

As we identified ULK1 phosphorylation on S638 in the proximal phosphoproteome (Fig. 5f) and this phosphorylation event is known to suppress autophagy⁴⁹, we investigated whether phosphorylated ULK1 on S638 localized at the recycling endosomes during FGFR2b recycling. In both HeLa-FGFR2bST and T47D cells ULK1 is recruited and phosphorylated on S638 in proximity of both FGFR2b and RAB11 as shown upon streptavidin beads enrichment of biotinylated proteins followed by western blot (Figure 6a-b, Supplementary Fig 8a). We confirmed that FGFR2b and phosphorylated ULK are in close proximity using the Proximity Ligation Assay (Fig. 6c and Supplementary Fig. 8b). Furthermore, confocal analysis of T47D_FGFR2^{KO}_FGFR2b-APEXST cells expressing wtRAB11 and stimulated with FGF10 for 40 min showed a significant co-localization between phosphorylated ULK1 on S638 and FGFR2b at the recycling endosomes (Fig. 6d-e). These findings confirm that ULK1 is associated to recycling endosomes⁵⁰ and suggest that the presence of stimulated FGFR2b at the recycling endosomes is necessary for the recruitment of phosphorylated ULK1 on S638. Indeed, we did not visualize any ULK1 phosphorylated on S638 when FGFR2b recycling was impaired by expressing DnRAB11 (Fig. 6d-e), or when FGFR2b recycling was inhibited through siRNA-mediated knockdown of the FGFR2b-specific recycling adaptors TTP or RCP (Supplementary Fig. 8c). The phosphorylation of ULK1 downstream of FGFR2b recycling is a specific event, as other FGFR2b downstream pathways, including phosphorylated FRS2 and ERK, were only marginally affected in cells expressing either DnRAB11 or DnDNM2, treated with the primaquine and dynasore compounds, all conditions that impaired FGFR2b trafficking^{13,22,51,52} or stimulated with FGF7 which does not regulate FGFR2b recycling (Fig. 1, Fig. 6f-g and Supplementary Fig. 8d-e). Intriguingly, inhibiting FGFR2b localization at the recycling endosomes by expressing DnRAB11 also misplaced the FGFR2b recycling regulator TTP¹³ from recycling endosomes to LAMP1-positive lysosomes (Supplementary Fig 8f-g, Fig. 6h-i), where it has previously been shown to negatively regulate mTOR signalling⁵³. Therefore, we checked mTOR localization and activation in our experimental conditions. mTOR was localized on lysosomes in both wtRAB11- and DnRAB11-expressing cells (Supplementary Fig. 8h, Fig. 6j). However, mTOR activation decreased in cells with impaired FGFR2b trafficking as shown by the analysis of the level of known genes regulated downstream of mTOR¹⁹ (Supplementary Fig. 8i). We also checked whether inhibiting FGFR2b recycling by expressing DnRAB11 affected other mTOR signalling partners, including RAPTOR and AMPK¹⁹. Inhibiting FGFR2b recycling prevented RAPTOR phosphorylation and AMPK

dephosphorylation on S863 and T172, respectively, events associated with increased mTORC1 activity (Fig. 6k), Phosphorylation of S638 on ULK1 was also decreased up to 2 h after FGF10 stimulation when FGFR2b recycling was inhibited (Fig. 6k, Supplementary Fig. 8d). These results clearly demonstrate a link between FGFR2b recycling and mTOR signalling.

In conclusion, FGFR2b recycling regulates mTOR signalling and the localization of phosphorylated ULK1 at the recycling endosomes, with these signalling events being crucial for autophagy suppression downstream of FGF10.

19. How quantifications in Fig. 6d have been done? how many cells per N have been measured? The labeling of Y axis should be changed as it is difficult to find out what is exactly measured.

We have described how we performed the quantification in the Methods section by summarizing the method recently published by us (Smith, EMBO J, 2021). The Method section reads as follows:

Quantification of FGFR2b recycling, co-localization (pixel overlap fraction), and Expression Fraction (pixel proportion) was performed as recently described in detail²². Briefly, quantification of internalization and recycling was performed as follows. For each time point and each treatment, the presence (total) and the localization (cell surface versus internalized) of HA-FGFR2b or endogenous FGFR2b were assessed in at least seven randomly chosen fields. Approximately 100 cells per condition (both acidic-washed and not) were analyzed from three independent experiments. The results are expressed as the percentage of receptor-positive cells (green) over total cells (corresponding to DAPI-stained nuclei) and referred to the values obtained at time zero. Statistical analysis was performed across repeats, as indicated in the figure legends.

Quantification of Expression Fraction, Overlap Fraction and Co-localization was performed as follow. Images were pre-processed using an “À trous” wavelet band pass filter to reduce the contribution of high frequency speckled noise to the co-localization calculations. Pixel intensities were then normalized from the original 8-bit range [0,255] to [0,1]. To ensure that co-localization was only computed in well-determined regions of interest (ROI), we used the Fiji/ImageJ built-in ROI manager to create and record these regions (minimum two cells and up to five per biological replicates with N=3). To measure differences in expression over time or between conditions, we computed the fractions of expressed red marker R, green marker G, or far-red marker F, pixels over a region of interest. To quantify the overlap fraction between two (R and G) or three (R, F and G) markers, we first multiplied the (normalized) channel intensities together to compute a new image whose intensity increases to 1 where the markers strongly overlap and decreases or becomes null for non-overlapping pixels. Our overlap fraction coefficient (OF) becomes the fraction of strictly positive pixels in the combined image over the number of pixels in the region of interest. Finally, to quantify the actual level of colocalization between two markers (e.g. R and G), we used the Manders Colocalization Coefficients (MCC) M1 and M2. M1 measures the fraction of the R marker in compartments that also contain the G marker, and M2, the fraction of the G marker in compartments that also contain the R marker. Lower-bound thresholds for pixel intensities were automatically determined using the Costes method. To measure the simultaneous overlap of our three, red, far-red and green markers (R, F, G), we first used the overlap image between marker R and marker F as defined above. We then measured the MCC colocalization parameter of this combined image against a green marker using the MCC formulae above, together with the Costes method to determine the thresholds. The scripts for the quantification of co-localization were written in the Python language and the code for Costes-adjusted MCC was taken verbatim from the CellProfiler code base. We analysed three independent experiments and between 2 and 5 cells for experiment. The Student’s t-test was subsequently used to

determine the difference in pixel overlap fraction between different experimental conditions in Fig 1 and 6, as indicated in the figure legends.

We have indicated in the figure legend of updated Fig 6d that we quantified three independent experiments and between 2 and 5 cells for experiment, as published (Smith, EMBO J, 2021). We have also changed the labelling of the graphs as suggested by the reviewer. See response to points 7 and 18 above for further details.

20. Fig 7a and c – how quantifications have been made in detail? what is N?

Quantification of the three cellular outputs can be found in the methods section and is summarized below:

Proliferation is measured as a percentage of cells with EdU incorporation. EdU is incorporated into newly synthesized DNA which can be visualised using a fluorescent azide.

Apoptosis was assessed using a CellEvent™ Caspase-3/7 Green Detection Reagent (Invitrogen), whereby cells with cleaved caspase have fluorescently labelled DNA. Fluorescence was measured at 502 nm excitation and 530 nm emission on a plate reader.

Autophagy was assessed using acridine orange, which fluoresces green when diffuse in the cytosol and fluoresces red when accumulated in acidic vesicles, such as autophagosomes. Excitation/emission wavelengths of 500/526 nm were used to measure intensity of diffuse acridine orange (non-specific) and 460/650 nm to assess autophagic staining. The ratio of these values represents stained autophagosomes.

The methods has been updated to explain this more clearly (from line 1672):

EdU Incorporation

Indicated cells were labelled with 20 µM 5-ethynyl-2'-deoxyuridine (EdU) for 4 h and processed following the manufacturer's protocol (Click-iT® EdU Alexa Fluor® 488 Imaging Kit, Thermo Fisher). EdU is incorporated into newly synthesised DNA, which can be visualized using a fluorescent azide. Prior to imaging cells were then stained with 5ng/ml Hoechst 3342 for 15 min.-Stained cells were analysed using a using a Leica microscope system. Statistical analysis was performed at the endpoint across repeats, as indicated in the Figure legends.

Cleaved caspase assay

Apoptosis was measured in cells receiving either- 24 h treatment with FGF10. Appropriately treated cells were incubated with 20 mM CellEvent™ Caspase-3/7 Green Detection Reagent (Invitrogen) made to 100X in PBS for 4 h in darkness then washed thoroughly in 1X PBS. This used a fluorogenic substrate for activated caspase 3/7, which is only cleaved by cleaved caspase, enabling DNA binding and fluorescence. Fluorescence was measured at 502 nm excitation and 530 nm emission. Statistical analysis was performed at the endpoint across repeats, as indicated in the Figure legends.

Autophagy

Populations of cells were assayed for autophagy using 5 mM Acridine Orange (Sigma) for 30 min after which excess was removed by thorough washing with 1X PBS. This fluorophore appears green when diffuse but is shifted to the red end of the spectrum when accumulated in acidic vesicles⁹². As such, excitation/emission wavelengths of 500/526 nm were used to measure intensity of diffuse acridine orange (non-specific) and 460/650 nm to assess autophagic staining. The ratio of these values represents stained autophagosomes. Statistical analysis was performed at the endpoint across repeats, as indicated in the Figure legends.

21. Fig 7c – autophagy/apoptosis/proliferation should be shown on separate graphs as these

are different tests and cannot be compared between each other on a single graph.

This has been changed and now corresponds to Figure 7c, Figure 7d and Figure 7e, respectively.

Figure legend now reads:

*Measurement of autophagy by acridine orange staining (c) cell proliferation by EdU incorporation (d), and cell apoptosis by cleaved caspase 3 activated dye (e) in T47D treated with the with FGFR inhibitor (FGFRi: PD173074), ULK1 inhibitor (ULK1i: ULK101), ULK1/2 inhibitor (ULK1/2i: SBI0206965), or mTOR inhibitor (mTORi: Rapamycin), stimulated or not with FGF10 for 2h. Data are presented as percentage compared to untreated cells. N = 6, p-value = < 0.001*** (one-way ANOVA with Tukey test).*

Minor points:

1. The Figures depicting analyses are in my opinion much too much detailed with limited significant contribution to the readability of the whole work for non MS experts. The authors could consider moving parts of MS data to supplement, leaving the most MS significant ones.

We have removed plots describing results from PCA from Figure 4; these are now found in the Supplementary Fig. 5b and 5d. We believe the figures that remain in the main panels are essential to communicate to MS- and proteomics-experts that our spatially resolved phosphoproteomics method is functional and a progression on previously published works. As the audience of Nature Communications is broad, we wanted to be able to communicate the technical aspect of our work to this particular audience, alongside the biological implications.

2. Typing errors:

P5/line 120 – vesicles for vesicles
P6/lines 174/175 – there is two times “presence”
P6 line 194 – localized for localize
P11 line 331 – lipidated instead of lappedated (present few times more throughout the text)

We have corrected this in the manuscript.

Congratulations on your impressive work,
With best regards,

Łukasz Opalinski

Reviewer #3 (Remarks to the Author):

Watson et al. describe an interesting tool to demonstrate spatio-temporal phosphorylation that occurs specifically at the RE. The authors put this into the context of FGFR2b recycling and the downstream effects of perturbed trafficking. The paper provides an interesting tool that could be valuable for distinguishing post-translational modifications dependent on specific localization.

The paper is well written and figures are well presented, however I have a few comments on the trafficking assays and analysis of downstream effects that are used to demonstrate the phenotype and consequences of perturbed trafficking, specifically autophagy.

We thank the reviewer for his or her detailed assessment of our work and we appreciate the reviewers' positive comments on our manuscript. We have addressed the points raised by the reviewer below.

Minor comments:

1. There are a number of typos throughout the manuscript, including a range of spellings of "lipidated"

We have corrected these mistakes.

2. For IF: In figure legends, please specify number of cells analysed from number of independent experiments, and for blots the number of repeats that were quantified (this should also be done), and the representative nature of the blots chosen. Statistical analysis also needs to be performed and described in the figure legend and indicated on the graphs.

We have added this information in the figure legends and in the Method section.

3. Terminology should be consistent throughout, especially in regard to internalization, cytoplasm, plasma membrane, binding, recycling. Recommend simplifying as one or other in both legends and figures.

We have updated the text and the figure legends.

4. Conditions in experiments should be referred to in respect to the experimental controls, and the control should be specified. Where "untreated" there should be a vehicle control, such as PBS or DMSO. Or mock transfected etc.

We have added this information in the figure legends and/or in the Method section.

5. Labelling of Western blots and IF should be made as efficient and clear as possible. There are also a few errors left over from assembling the figures, and typos in the labelling (Supp Fig 6c).

We have corrected these mistakes.

6. Graph presentation, especially in respect to x/y axes, should be changed for clarity. It is most informative for the reader to compare CT to treatments in each readout, ie Fig 7c. Present grouped or as separate graphs showing effects of CT vs treatments on autophagy, apoptosis, proliferation.

This has been changed and now corresponds to Figure 7c, Figure 7d and Figure 7e.

The separated graphs are shown below:

Figure legend now reads:

Measurement of autophagy by acridine orange staining (c) cell proliferation by EdU incorporation (d), and cell apoptosis by cleaved caspase 3 activated dye (e) in T47D treated with the with FGFR inhibitor (FGFRi: PD173074), ULK1 inhibitor (ULKi: ULK101), ULK1/2 inhibitor (ULK1/2i: SBI0206965), or mTOR inhibitor (mTORi: Rapamycin), stimulated or not with FGF10 for 2h. Data are presented as percentage compared to untreated cells. N = 6, p-value = < 0.001*** (one-way ANOVA with Tukey test).

7. Don't show inset table, simplify labelling to change terms used to quote the target of inhibition, and refer to original inhibitor names in materials and methods or figure legend.

This has been simplified and the name of the drugs included in the materials and methods.

8. ERK and mTOR activation is tested separately in Figs 1, 3, 6 and 7. These pathways are interlinked, and this could be commented on in the discussion.

We have added comments on the relationship between ERK and mTOR to the discussion as suggested (from line 1218). The corresponding section of the discussion can be found below:

RTK signalling and endocytosis have previously been linked to regulation of autophagy⁷³ and EGFR recycling has been shown to decrease in cells lacking autophagy regulators⁷⁴. Signals from growth factors are known to converge on the mTORC1 complexes at the lysosomal membrane to inhibit autophagy and catabolic processes¹⁹. Focusing on the FGFR family, the FGFR2b selective ligand FGF7 has been shown to induce autophagy in keratinocytes after 24 h stimulation⁷⁵ and FGF signalling regulates bone growth through autophagy⁴². However, within the 2 h timeframe used in our experiments, FGF7 fails to alter ULK1/mTOR signalling or the downstream autophagy response, in contrast to the responses achieved in FGF10-stimulated cells. Indeed, prior to our SRP approach, we had not associated recycling-dependent FGF10-FGFR2b signalling with enhanced mTOR activity. ERK is known to regulate mTOR activity, either indirectly through negative regulation of TSC complex or by direct phosphorylation of Raptor⁷⁶, while the regulatory relationship between ERK activity reduces AMPK activity in a context-dependent manner⁷⁷. It is interesting that ERK activity is comparable between FGF7 and FGF10. However ERK activation does not lead to

mTOR/ULK1 mediated suppression of autophagy downstream of FGF7. This would suggest that a role for ERK in regulating autophagy downstream of FGF7 or FGF10 would be independent of the level of ERK activation. Instead, the recycling endosomes could be required for co-ordinating ERK signalling downstream of membrane activation. The stark difference between FGF7 and FGF10 high lights the role of FGFR2b recycling as the regulator of the FGF10/ULK1/autophagy interplay. How this is orchestrated from the recycling endosomes remains however unclear. One possibility is the involvement of EGFR signalling, as we have recently shown that EGFR is phosphorylated downstream of FGF10/FGFR2b recycling at the recycling endosomes²² and EGFR signalling regulates autophagy⁷⁸ with EGFR trafficking requiring autophagy regulators⁷⁴. Alternatively, recycling endosomes and autophagosomes share signalling regulatory components that would require further investigations^{50,79}. Thus, a picture of recycling endosomes as a point of convergence for several signalling pathways and for coordination of long-term responses is clearly emerging. This information can be used to exploit recycling endosomes for nanomedicine, for instance for a better deliver of siRNA against specific signalling players⁸⁰.

Major comments:

1.Short introductory phrases should be included to describe experiments in the results sections, and a short description of what the results show. Some of the blots are large and complex, and the reader would benefit from being guided through what they should focus on. Ie. Immunofluorescence analysis of LC3 puncta in the presence or absence of xyz compared to CT, showed an increase in LC3 puncta, indicating an increase in basal autophagy in Y cells (Fig Xa). This is often absent.

We have updated the result section.

2.Materials and methods are not clear at times and lacking important information. I see that there are numerous references to a recent publication from the same group, but details for antibodies, and key relevant experimental details must be included.

We have added information as suggested by the reviewer. The antibodies are listed under "Plasmid, Antibodies and reagents". We have added details of how we performed the quantification in the "Immunofluorescence and quantification" section. We have added information on how experiments have been performed, and against which control at the beginning of the "Biochemical and functional assay section" and in appropriate result sections or figure legends as well. All original data has been provided.

3.One of my main concerns for this manuscript is the differentiation between antibody binding, vs recycling. The assay must be explained in more detail in the materials and methods, ie is binding performed at 4C? Is there an acid wash after binding and before fixation for analysis of recycled FGFR2b? In IF figures and description in the results, it is not clear that binding and recycling have been separated, however this is crucial for the experiment. Often, this may be performed by monitoring a modification that the protein undergoes while it is internalized. However in this case, this could also be done by stripping the membrane of non-internalised protein (acid wash) before then monitoring the amount of protein returned to the cell surface. Please provide an explanation of how these two phenotypes are distinguished from one another.

The internalization/recycling assay used in the updated Supplementary Fig 4b-4c and all the co-localization experiments shown in Figures 1, 3, 6 and Supplementary Figures 6, 8, and 9 have been previously described and used by the authors (and others) in different experimental conditions to study receptor trafficking (see, for instance, Francavilla, J Cell Biology, 2009:

Francavilla, Mol Cell, 2013; Francavilla, Nat Struct Mol Biol, 2016; Smith, EMBO J, 2021). We have added more information in the Methods Section, including details of the temperature used for the assays (4°C that allows the binding but not the internalization of the primary antibody, as reported). We did acidic washed samples before fixation as specified in the Method section.

Furthermore, we better described how we performed and quantified the internalization/recycling assay shown in Supplementary Fig 4b-4c in the updated methods and result sections (from line 1758) which now read:

Quantification of FGFR2b recycling, co-localization (pixel overlap fraction), and Expression Fraction (pixel proportion) was performed as recently described in detail²². Briefly, quantification of internalization and recycling was performed as follows. For each time point and each treatment, the presence (total) and the localization (cell surface versus internalized) of HA-FGFR2b or endogenous FGFR2b were assessed in at least seven randomly chosen fields. Approximately 100 cells per condition (both acidic-washed and not) were analyzed from three independent experiments. The results are expressed as the percentage of receptor-positive cells (green) over total cells (corresponding to DAPI-stained nuclei) and referred to the values obtained at time zero. Statistical analysis was performed across repeats, as indicated in the figure legends.

Quantification of Expression Fraction, Overlap Fraction and Co-localization was performed as follow. Images were pre-processed using an “À trous” wavelet band pass filter to reduce the contribution of high frequency speckled noise to the co-localization calculations. Pixel intensities were then normalized from the original 8-bit range [0,255] to [0,1]. To ensure that co-localization was only computed in well-determined regions of interest (ROI), we used the Fiji/ImageJ built-in ROI manager to create and record these regions (minimum two cells and up to five per biological replicates with N=3). To measure differences in expression over time or between conditions, we computed the fractions of expressed red marker R, green marker G, or far-red marker F, pixels over a region of interest. To quantify the overlap fraction between two (R and G) or three (R, F and G) markers, we first multiplied the (normalized) channel intensities together to compute a new image whose intensity increases to 1 where the markers strongly overlap and decreases or becomes null for non-overlapping pixels. Our overlap fraction coefficient (OF) becomes the fraction of strictly positive pixels in the combined image over the number of pixels in the region of interest. Finally, to quantify the actual level of colocalization between two markers (e.g. R and G), we used the Manders Colocalization Coefficients (MCC) M1 and M2. M1 measures the fraction of the R marker in compartments that also contain the G marker, and M2, the fraction of the G marker in compartments that also contain the R marker. Lower-bound thresholds for pixel intensities were automatically determined using the Costes method. To measure the simultaneous overlap of our three, red, far-red and green markers (R, F, G), we first used the overlap image between marker R and marker F as defined above. We then measured the MCC colocalization parameter of this combined image against a green marker using the MCC formulae above, together with the Costes method to determine the thresholds. The scripts for the quantification of co-localization were written in the Python language and the code for Costes-adjusted MCC was taken verbatim from the CellProfiler code base. We analysed three independent experiments and between 2 and 5 cells for experiment. The Student’s t-test was subsequently used to determine the difference in pixel overlap fraction between different experimental conditions in Fig 1 and 6, as indicated in the figure legends.

Quantification of LC3-, LAMP1-, and LC3/LAMP1-positive vesicles was performed manually using Image J. For N=3 independent experiment we analysed between 15 and 25 cells per image by adjusting the threshold of each channel to a value equal to 50 followed by particles counting. The number of LC3- or LAMP1-positive vesicles was divided by the number of DAPI-stained nuclei to obtain the ratio of autophagosomes and lysosomes, respectively. We

manually counted the number of LC3/LAMP1-positive vesicles using the merge image to determine the ratio of mature autophagosomes.

As highlighted in our response to reviewer 2 major point 8, the right panels labelled as “cytoplasm” in Supplementary Fig 4c (see updated figure below) how the samples after acidic wash, fixation and permeabilization, whereas panels labelled as “plasma membrane” show samples before acidic wash and not permeabilized. At time zero there is no FGFR detected in the cytoplasm of the unstimulated samples, but FGFR is detected in all the panels in the left (plasma membrane) which indicates that the antibody binds to FGFR at the plasma membrane but does not induce any internalization. FGFR signal was detected in the acidic washed samples on the right upon stimulation with FGF10 for 40 min, a condition where the signal was absent in the not acidic washed samples. This indicates that FGFR was internalized at this time point. At the same time point (40 min) unstimulated cells (control) shows the opposite: the FGFR signal was detected only in the not acidic washed samples which indicates lack of internalization. The surface staining of FGFR in FGF10-stimulated cells for 120 min indicates FGFR recycling back to the plasma membrane after internalization as there is no staining in the corresponding acidic washed cells on the right. In conclusion, the well-established (see also Di Guglielmo, Nat Cell Bio, 2003) internalization/recycling assay used in Supplementary Fig 4 clearly distinguish antibody binding and recycling in our experimental conditions.

b FGFR2b (top panels) and FGFR2b-APEX2 (middle and bottom panels) presence in the cytoplasm and at the plasma membrane in HeLa cells (bottom and middle panels, green) and T47D (bottom panels, red) untreated (UT) or stimulated with FGF10 or FGF7 for 0, 40 and 120 min. Nuclei are stained with DAPI (blue). Scale bar, 5 μ m. **c** Quantification of FGFR2b internalization and recycling in the three cell lines, showing the presence (total; top panel), recycled (cell surface; middle panel) and internalized (internalized; bottom panel) FGFR2b

*upon stimulation. Values represent median \pm standard deviation of at least three independent experiments where about 100 cells in total were counted for each condition, exact numbers indicated on graph. p-value < 0.05 *, p-value < 0.005 **, p-value < 0.0005 *** (one-way ANOVA, post-hoc tukey, significance compared to time 0 indicated) Representative images are shown in b.*

4. The inhibition of trafficking is not mentioned in the manuscript. Does the FGFR2b become blocked in EEs? Is this what is demonstrated by showing coloc with EEA1 in Fig 1a? Does it become degraded? It would be interesting to see the fate of this FGFR2b in regards to endosome/lysosome localization.

As highlighted in our response to reviewer 1 minor point 2 and reviewer 2 major point 9, the K44A mutant form of the GTPase dynamin2 (Dominant negative dynamin, DnDNM2 in our manuscript) is known to prevent the clathrin-mediated internalization of Receptor Tyrosine Kinases like Epidermal Growth Factor Receptor (EGFR) and FGFR (Vieira, Science, 1996; Sigismund, Dev Cell, 2008; Smith, EMBO J, 2021). We have previously shown that expressing DnDNM2 in the epithelial breast cancer cell line T47D inhibited FGFR2b internalization (Smith, EMBO J., 2021). Here, we show the same effect of DnDNM2 in inhibiting FGFR2b internalization in the epithelial cell line HeLa (Fig 1a-b). The updated Fig 1a shows the localization of FGFR2b at 0, 40 and 120 min in unstimulated cells or upon stimulation with FGF10 in three different experimental conditions. When cells express either Rab11-GFP or mutated Rab11-GFP (wtRAB11 and DnRAB11 in our manuscript) FGFR2b (red) localizes at the plasma membrane at all time points in untreated cells and at time 0 in stimulated cells as no signals was detected in the cytoplasm (right panels in Fig 1a), whereas FGFR2b localizes in RAB11- or DnRAB11-positive compartments in cells stimulated for 40 min with FGF10 as the red signal was detected only in the cytoplasm (right panel). In cells expressing RAB11, FGFR2b and RAB11 co-localize (yellow) which indicates that FGFR2b has been internalized and is in the recycling endosomes, as previously reported (Belleudi, Traffic, 2007, Francavilla, Mol Cell, 2013). However, in cells expressing DnRAB11, FGFR2b co-localizes with DnRAB11 and also with EEA1 (blue), a marker of early endosomes, which indicates receptor internalization (Francavilla, J Cell Biol, 2009). The white signal indicates that FGFR2b has been internalized but it is not in the recycling compartment. Interestingly, at 120 min stimulation FGFR2b remains in the early endosomes (white) in cells expressing DnRAB11, but not RAB11. On the other hand, when cells express DnDNM2 (green in the 12 bottom panels of Fig 1a), FGFR2b (red) is detected at the plasma membrane in all conditions and never detected in the cytoplasm, thus indicating lack of internalization. Furthermore, there is no co-localization between FGFR2b and the marker of internalization EEA1 in the cytoplasm of cells expressing DnDNM2. In conclusion, FGFR2b becomes blocked in EEA1-positive early endosomes when cells express DnRAB11 and at the plasma membrane when cells express DnDNM2.

The updated results section (from line 158) reads:

We transiently expressed (more than 80% of positive cells) Dynamin_K44A-eGFP (dominant negative Dynamin, DnDNM2) or eGFP-RAB11_S25N (dominant negative RAB11, DnRAB11), which are known to inhibit FGFR2b internalization and recycling to the plasma membrane respectively, in response to FGF10 stimulation for 40 min²². At this time point FGFR2b was localized in the recycling endosomes in cells expressing wild-type e-GFP-RAB11 (wild-type RAB11, wtRAB11) (Fig. 1a-b)^{13,22}. We also stimulated cells with FGF10 for 120 min to study the fate of FGFR2b at a longer time point. As shown for FGFR1³⁶, FGFR2b co-localized with the marker of early endosomes EEA1, and with DnRAB11 in cells expressing DnRAB11 and was not found at the plasma membrane upon 40 and also 120 min stimulation with FGF10 (Fig. 1a) These findings suggest that FGFR2b is trapped in EEA1/DnRAB11-positive vesicles.

When cells express DnDNM2 (green in the eight bottom panels of Fig 1a), FGFR2b (red) was detected at the plasma membrane at all time points and was never detected in the cytoplasm, thus indicating lack of internalization. Furthermore, there was no co-localization between FGFR2b and the marker of early endosomes EEA1 in the cytoplasm. The results of this experiment are quantified in Fig 1b. In conclusion, expressing DnDNM2 and DnRAB11 impair FGFR2b trafficking and will be used here to study trafficking-dependent changes in FGFR2b signalling in response to FGF10.

Fig. 1. FGFR2b activation is not affected by receptor sub-cellular localization. **a** FGFR2b (red) internalization (cytoplasm) and FGFR2b recycling (plasma membrane) in HeLa cells stably transfected with FGFR2b-HA (HeLa_FGFR2bST), expressing eGFP-RAB11a (wtRAB11), dominant negative eGFP-RAB11a_S25N (DnRAB11), or dominant negative dynamin-2_K44A-eGFP (DnDNM2) (green), and treated with FGF10 for 0, 40 and 120 min or left untreated (control). Early endosome antigen 1 (EEA1) (blue) is a marker for EEs²². Scale bar, 5 μ m. Single channels are shown on the right for FGF10-stimulated cells for 0, 40 and 120 min.. White arrowheads indicate co-localization or lack thereof. **b** Quantification of the co-localization of stimulated FGFR2b (red pixels) with GFP-tagged proteins (green pixels) indicated by red-green pixel overlap fraction (left panel). Quantification of the co-localization of FGFR2b (red pixels) with EEA1 (blue pixels) indicated by red-blue pixel overlap fraction (right panel). Representative images are shown in 1a. Values represent median \pm SD from N=3 where we analysed between 2 and 5 cells for each N; *** p-value < 0.0005 (students t-test)²².

FGF10

5. My other main concern is the quality of the IF images shown. Firstly, the figures are not clearly labelled, I believe that their may be CT v FGF10 labelling absent (ie Fig 1a). Images could be clarified by showing individual channels plus overlay, especially in the zoomed regions, which should be included for ALL conditions, including control. Zoomed panels must also be treated in the same way as the main image, not manipulated differently. Throughout the paper, zoomed images are manipulated too much, and the colocalisation is not clear. This is especially evident in Fig6, where it seems that gain/exposure of the images has been pushed too much during post-acquisition processing.

We apologise with the reviewer for unclarity about how we collected the IF images. We have performed the following changes to our manuscript:

1. We have updated Figure 1 and figure legend: see above.
2. We have changed the result related to Figure 1 (from line 158) as follows:

We transiently expressed (more than 80% of positive cells) Dynamin_K44A-eGFP (dominant negative Dynamin, DnDNM2) or eGFP-RAB11_S25N (dominant negative RAB11, DnRAB11), which are known to inhibit FGFR2b internalization and recycling to the plasma membrane respectively, in response to FGF10 stimulation for 40 min²². At this time point FGFR2b was localized in the recycling endosomes in cells expressing wild-type e-GFP-RAB11 (wild-type RAB11, wtRAB11) (Fig. 1a-b)^{13,22}. We also stimulated cells with FGF10 for 120 min to study the fate of FGFR2b at a longer time point. As shown for FGFR1³⁶, FGFR2b co-localized with the marker of early endosomes EEA1, and with DnRAB11 in cells expressing DnRAB11 and was not found at the plasma membrane upon 40 and also 120 min stimulation with FGF10 (Fig. 1a) These findings suggest that FGFR2b is trapped in EEA1/DnRAB11-positive vesicles. When cells express DnDNM2 (green in the eight bottom panels of Fig 1a), FGFR2b (red) was detected at the plasma membrane at all time points and was never detected in the cytoplasm, thus indicating lack of internalization. Furthermore, there was no co-localization between FGFR2b and the marker of early endosomes EEA1 in the cytoplasm. The results of this experiment are quantified in Fig 1b. In conclusion, expressing DnDNM2 and DnRAB11 impair FGFR2b trafficking and will be used here to study trafficking-dependent changes in FGFR2b signalling in response to FGF10.

3. We have updated Figure 6 and legend:

Fig. 6. Phosphorylated ULK1 recruitment at the recycling endosomes depends on FGFR2b recycling. *a, b* Immunoblot analysis ($N \geq 3$) with the indicated antibodies of HeLa_FGFR2bST_RAB11-APEX2 (*a*) or T47D transfected with RAB11-APEX2 (T47D_RAB11-APEX2) (*b*) stimulated with FGF10 for the indicated timepoints. Non proximal and proximal samples represent the supernatant and the pull-down following enrichment of biotinylated samples with streptavidin beads, respectively, and run against total lysates (total).

c Quantification of proximity ligation assay (PLA) puncta between FGFR2b and S638 pULK1 in HeLa_FGFR2bST cells treated with FGF10 40 mins compared to untreated (UT); p-value < 0.0005 *** (Students t-test) **d** Co-localization of FGFR2b-APEX2 (red) with phosphorylated ULK1 on S638 (blue) in T47D_FGFR2^{KO}_FGFR2b-APEXST transfected with RAB11 or GFP-DnRAB11 (green) and stimulated or not with FGF10 for 40 min as indicated. Scale bar, 5 µm. The white arrowhead indicates co-localization or lack thereof. **e** Quantification of the co-localization of FGFR2b (red pixels) with GFP-tagged proteins (green pixels) indicated by red-green pixel overlap fraction (top panel), of the co-localization of FGFR2b (red pixels) with GFP-tagged proteins (green) and with phosphorylated ULK1 (blue pixels) indicated by red-green-blue pixel overlap fraction (middle panel). The presence of phosphorylated ULK1 was determined by pixel proportion (see Methods) (bottom panel). Representative images are shown in 6c. Values represent median ± SD from N=3 where we analysed between 2 and 5 cells for each N; p-value < 0.005 **; p-value < 0.0005 *** (Students t-test). **f, g.** Immunoblot analysis (N>=3) with the indicated antibodies of HeLa_FGFR2bST (e) or T47D (f) transfected either with wtRAB11, DnRAB11, or DnDNM2 and left either untreated (UT) or treated with FGF10 for the indicated time points. **h** Quantification of the co-localization of FGFR2b (red pixels) with GFP-tagged proteins (green pixels) indicated by red-green pixel overlap fraction (first panel), of the co-localization of FGFR2b (red pixels) with GFP-tagged proteins (green) and with TTP (blue pixels) indicated by red-green-blue pixel overlap fraction (second panel), of the presence of TTP determined by pixel proportion (see Methods) (third panel). Representative images are shown in Supplementary Fig 8f. **i** Quantification of the co-localization of LAMP1 (red pixels) with TTP (blue pixels) indicated by red-blue pixel overlap fraction (first panel) and of the co-localization of GFP-tagged proteins (green) with TTP (blue pixels) indicated by green-blue pixel overlap fraction (second panel). Representative images are shown in Supplementary Fig 8g. Values represent median ± SD from N=3 where we analysed between 2 and 5 cells for each N; *** p-value<0.0005 (Student t-test). **j** Quantification of the co-localization of LAMP1 (red) with mTOR (blue) Indicated by red-blue pixel overlap fraction (right panel) and of the co-localization of GFP-tagged proteins (green) with mTOR (blue pixels) indicated by green-blue pixel overlap fraction (left panel). Representative images are shown in Supplementary Fig 8h. Values represent median ± SD from N=3 where we analysed between 2 and 5 cells for each N; *** p-value < 0.0005 (Students t-test). **k** Immunoblot analysis with indicated antibodies of T47D transfected with GFP or DnRAB11 and left either untreated (UT) or treated with FGF10 for the indicated time points.

4. We have changed the result section related to figure 6 starting from line 826.

6. The Lamp/mTOR imaging in Fig 6c/h does not look convincing. What is the IF protocol used? When performing IF for endo/lysosomes it can be tricky, especially when balancing conditions to visualize mTOR and additional markers. The protocol may need to be revisited further to optimize this staining, and show robust mTOR/Lamp1. Steps to optimize should include fixation, permeabilisation, blocking buffer. If optimisation has already been performed, please provide evidence of validation of IF. IF protocols should also be clearly explained in Materials and methods. As in the previous point, this will likely also be clarified by better image representation.

To detect the co-localization between LAMP1 and mTOR we used a published protocol including the use of the same primary antibody anti-mTOR (Cell Signalling, cat number 2983) (Nnah, Autophagy, 2019). The published protocol is in line with the protocol described in Material section:

Immunofluorescence staining was performed as previously described 22. To detect HA-FGFR2b or endogenous FGFR2 we incubated cells with 10 µg/ml of anti-HA (Covance) or anti-FGFR2 antibody (Cell Signalling) for 45 min with gentle agitation at 4C. The binding of the antibody did not activate receptor signalling in untreated cells nor induced receptor

internalization (see control cells in Fig 1, 3, and 6), as previously reported 22. After stimulation cells were incubated at 37°C for different time points. At each time point, non-permeabilized cells were either fixed to visualize the receptor on the cell surface (plasma membrane) or acid-washed in ice-cold buffer (50 mM glycine, pH 2.5) to remove surface-bound antibody. Acid-washed cells were then fixed and permeabilized to visualize the internalized receptor (cytoplasm). Finally, to detect the receptor cells were stained with AlexaFluor488-conjugated donkey anti-mouse or anti-rabbit (Jackson ImmunoResearch Laboratories). Nuclei were stained with DAPI. Coverslips were then mounted in mounting medium (Vectashield; Vector Laboratories).

For co-localization experiments, cells were fixed with 4% paraformaldehyde/2% sucrose for 10 min at room temperature, permeabilized with 0.02% saponin (Sigma) (except for the experiment looking at LAMP1-mTOR co-localization), blocked in 0.5% BsA and 0.5% Triton X-100 in PBS for 120 min at room temperature. treated with the indicated primary antibody for 60 min at 37 °C (overnight at 4C in case of mTOR/LAMP1 staining), and stained with AlexaFluor488 (or 568 or 647)-conjugated donkey anti-mouse or anti-rabbit. Samples expressing GFP-tagged proteins were kept in the dark. Nuclei were stained with DAPI. Coverslips were then mounted in mounting medium (Vectashield; Vector Laboratories).

We selected a picture better representing the result of the quantification which is shown now in Fig 6i and show separate channels in Supplementary Fig 8.

Supplementary Fig. 8 FGFR2b regulates mTOR and ULK1 signalling from the recycling endosomes. **a** Immunoblot analysis ($N \geq 3$) with the indicated antibodies of HeLa FGFR2b-APEX2ST (top) and T47D_FGFR2b^{KO}-FGFR2b-APEX2ST (bottom). Non proximal and proximal samples represent the supernatant and the pull-down following enrichment of biotinylated samples with streptavidin beads, respectively, and run against total lysates (total). **b** Representative images corresponding to quantification in Fig. 6c, from Proximity Ligation Assay between FGFR2b and S638 pULK1 (green) in T47D cells treated with FGF10 compared to untreated (UT). **c** Immunoblot analysis ($N=3$) with the indicated antibodies of T47D cells with siRNA-mediated knockdown of TTP or RCP, compared to siRNA control, treated with FGF10 for indicated time points **d** Immunoblot analysis ($N=3$) with the indicated antibodies of HeLa_FGFR2bST cells pre-treated with primaquine or Dynasore for 2 h followed by stimulation with FGF10 for the indicated time points. **e** Immunoblot analysis ($N=3$) with the indicated antibodies of T47D cells transfected with either GFP or DnRAB11 and stimulated with FGF7 for the indicated time points. **f** Co-localization of FGFR2b (red) with TTP (blue) in T47D_FGFR2^{KO}_FGFR2b-APEXST transfected with wtRAB11 or DnRAB11 (green) and stimulated or not with FGF10 for 40 min as indicated. Scale bar, 5 μ m. The white arrowhead indicates co-localization or lack thereof. Co-localization of LAMP1 (red) with either TTP (blue) (**g**) or mTOR (blue) (**h**) in T47D_FGFR2^{KO}_FGFR2b-APEXST transfected with wtRAB11 or DnRAB11 (green) and stimulated or not with FGF10 for 40 min as indicated. Scale bar, 5 μ m. The white arrowhead indicates co-localization or lack thereof. **i** Expression of indicated genes in HeLa FGFR2b or T47D transfected with wtRAB11, DnRAB11 or DnDNM2 or pre-incubated with rapamycin for 2 h followed by stimulation with FGF10 for 4 h. qPCR data are presented as heat map from $N=3$.

7. Fig 1a: Please confirm that zooms are consistent in all images. If not, put scale bars on separate images. Images should aim to represent the result shown in the quantification in the clearest way possible through cell choice and the representation. It seems that some of the cells are much smaller in certain conditions. Why is this only in the bottom panel? If this is a true representation, it should be commented on, and this must be taken into consideration in quantification.

We confirm that the zoom was the same in all images and we selected better representative images in certain conditions. (see updated Figure 1 above). The images corresponding to the nuclei of selected experimental conditions confirm consistency of the zoom. Furthermore, we took into account the size of the cytoplasm during quantification, as explained in the updated Method section (see text above).

8. Re image quantification. Should be represented as in Fig7, showing individual cell values. Method of quantification must be carefully selected. Please ensure that this method takes into consideration the changes in presence/intensity in the FGFR2b channel, as there is a big difference across conditions.

We have updated the graphs showing image quantification (see above Fig 1 and Fig 6) and described the procedure in the method section (see text above). The method for quantification has been previously published (Smith, EMBO J, 2021) and took into account the intensity of proteins in each channel, including FGFR2b.

9. It is not clear what quantification refers to in Fig 3d. It would be helpful to clarify which cells are shown in Fig 3c and which are quantified, standardize labelling throughout. I presume that Fig 3c upper panel is HeLa_FGFR2BST, while Fig 3c Lower panel is HeLa_FGFR2Bst-APEX2ST. If so, where is T47D...? If not included in main figs, please include imaging in supplementary. It is also unclear how this quantification has been done. Is quant of total FGFR2b done on cells that are not shown? Or is it calculated from PM + internalized? Why is quant here done differently to Fig 1? The quantification does not seem to reflect what is shown

in the images. Staining is almost absent in FGF10 40', however quantification shows that this is still about 90% of CT, this doesn't look like a convincing representation.

We apologize for the mistakes and for not having included enough information about the protocol and the method used for quantification of the internalization/recycling assay. We have used our previously published protocol and quantification methods for the internalization/recycling assay (Francavilla, *Journal Cell Biology*, 2009; Francavilla, *Mol Cell*, 2013; Francavilla, *Nat Struct Mol Biol*, 2016; Smith, *EMBO J*, 2021). For each experimental condition, this method counts how many cells over a total of about 100 express FGFR2b (green or red signal) at the plasma membrane or in the cytoplasm for each given time. We could differentiate the signal at the plasma membrane from the signal in the cytoplasm because, for each time point, cells stimulated with the same stimulus were either fixed or acidic washed, fixed and premetallized (see the Method section above). The results are shown as percentage of total amount of FGFR2b present at time zero. This method enables us to distinguish the total, cell surface and internalized amount of receptor, whereas the method used in Fig 1b and updated Supplementary Fig 4c quantifies the co-localization of FGFR2b with known markers of recycling or early endosomes.

There were a few labelling mistakes in Fig 3c-d (now Supplementary Fig 4b-c) which have been now corrected and that led to an incorrect interpretation of the results. Upon stimulation with FGF10 for 40 min we observed a decrease of the signal of FGFR2b at the plasma membrane, but a clear presence in the cytoplasm (images on the left and right of Supplementary Fig 4b) in all the tested cell lines. Indeed, the updated quantification in Supplementary Fig 4c shows that about 50% of FGFR2b was in the cytoplasm in FGF10-stimulated cells for 40 min (internalized FGFR2b, bottom right panel) and about 50% was still at the surface (cell surface FGFR2b, bottom left panel) for a total of 100% (total FGFR2b, top panel). On the contrary, in FGF7 stimulated cells FGFR2b was present in the cytoplasm at 40 min stimulation, but we detected only 25% of FGFR2b still at the cell surface, which indicates receptor degradation, as previously reported (Belleudi, *Traffic*, 2007; Francavilla, *Mol Cell*, 2013). Therefore, our conclusion that FGF10 induces FGFR2b recycling is consistent with previous publications and our interpretation of the experiments shown in Supplementary Fig 4b-c is correct. Updated result section starts at line 394 in the updated manuscript.

Furthermore, we have added images of T47D where FGFR2b is stained in red and we have performed the co-localization experiment shown in Fig 1 also in HeLa_FGFR2b-APEX2ST. The result of both assays is consistent with data shown in Supplementary Fig 4b-c and Fig 1a-b. The updated Figures and legend are copied below.

Fig. 3. APEX2 tagged-FGFR2b and RAB11a identifies compartment-specific signalling partners upon FGF10 stimulation. *a* Schematic underlying the Spatially Resolved Phosphoproteomics (SRP) approach. Panel 1 represents the trafficking of FGFR2b-APEX2 stimulated with FGF10 in *HeLa_FGFR2b-APEX2ST* and subsequent FGFR2b-APEX2 proximal phosphoproteome; panels 2 and 3 represent the localization of FGFR2b and of the APEX2-tagged proteins in cells expressing either FGFR2b and Rab11-APEX2 (*HeLa_FGFR2bST RAB11-APEX2*) or FGFR2b and GFP-APEX2 (*HeLa_FGFR2bST GFP-APEX2*) stimulated for 40 min with FGF10, and the proximal phosphoproteomes to the bait.

Panel 4 represents the phosphorylated events occurring at the RAB11- and FGFR2b-positive recycling endosomes upon 40 min FGF10 stimulation after subtracting cytosolic events using HeLa_FGFR2bST GFP-APEX2 proximal phosphoproteome. **b** Immunoblot analysis ($N \geq 3$) with the indicated antibodies of HeLa_FGFR2bST (right) or HeLa_FGFR2b-APEX2ST (left) stimulated with FGF10 for 1, 8, 40, 60, or 120 min or left untreated. **c** FGFR2b (red) internalization (cytoplasm) and FGFR2b recycling (plasma membrane) in HeLa_FGFR2b-APEX2ST, expressing eGFP-RAB11a (wtRAB11), dominant negative eGFP-RAB11a_S25N (DnRAB11), or dominant negative dynamin-2_K44A-eGFP (DnDNM2) (green), and treated with FGF10 for 0 and 40 min or left untreated (UT). Early endosome antigen 1 (EEA1) (blue) is a marker for early endosomes²². Scale bar, 5 μm . Single channels are shown on the right for FGF10-stimulated cells for 0 and 40 min.. White arrowheads indicate co-localization or lack thereof. **d** Quantification of the co-localization of stimulated FGFR2b (red pixels) with GFP-tagged proteins (green pixels) indicated by red-green pixel overlap fraction (top panel). Quantification of the co-localization of FGFR2b (red pixels) with EEA1 (blue pixels) indicated by red-blue pixel overlap fraction (bottom panel). Representative images are shown in c. Values represent median \pm SD from $N = 3$ where we analysed between 2 and 5 cells for each N ; *** p -value < 0.0005 (students t-test)²² **e** Immunoblot analysis ($N \geq 3$) with the indicated antibodies of input or biotinylated proteins enriched with Streptavidin beads from HeLa_FGFR2b-APEXST left untreated (UT) and treated either with H₂O₂ or with FGF10 for 1 and 8 min. **f** Schematic of RE-localised FGFR2b, following 40 min of FGF10 treatment. Both RAB11-APEX2 and RAB25 localize at the REs³⁶. **g** Immunoblot analysis ($N \geq 3$) with the indicated antibodies of input or biotinylated proteins enriched with Streptavidin beads from HeLa_FGFR2bST_RAB11-APEX2 stimulated with either H₂O₂ or with FGF10 for 40 min

Supplementary Fig. 4b-4c and corresponding figure legend are below:

b FGFR2b (top panels) and FGFR2b-APEX2 (middle and bottom panels) presence in the cytoplasm and at the plasma membrane in HeLa cells (bottom and middle panels, green) and T47D (bottom panels, red) untreated (UT) or stimulated with FGF10 or FGF7 for 0, 40 and 120 min. Nuclei are stained with DAPI (blue). Scale bar, 5 μ m. **c** Quantification of FGFR2b internalization and recycling in the three cell lines, showing the presence (total; top panel), recycled (cell surface; middle panel) and internalized (internalized; bottom panel) FGFR2b upon stimulation. Values represent median \pm standard deviation of at least three independent experiments where about 100 cells in total were counted for each condition, exact numbers indicated on graph. *p*-value < 0.05 *, *p*-value < 0.005 **, *p*-value < 0.0005 *** (one-way ANOVA, post-hoc tukey, significance compared to time 0 indicated) Representative images are shown in *b*.

10. The measurement of autophagy needs to be performed more precisely. The authors have looked at the acidic compartments of the cells, relating them to lysosomes. Firstly, the acridine orange images should be included in the manuscript. Secondly, to interpret autophagy data properly, autophagic flux must be measured. This should be performed through Western blotting of LC3 I-II conversion, measuring the flux by comparing treated conditions of perturbation (ie FGF10) +/- Bafilomycin A1 which blocks lysosomal degradation. If possible, an autophagy cargo should also be used to provide complementary data to LC3I-II, generic (but not always relevant) p62, or even FGFR2b (which would also contribute to the manuscript and show if FGFR2b is degraded).

We agree with the reviewer that acridine orange is a cruder measurement of autophagy although widely used. The analysis we undertook with acridine orange used plate reader measurements on a population basis rather than assessment of individual cells and images were therefore not acquired. To strengthen the evidence of suppression of autophagy by FGF10, we have added further experiments, which support FGF10 suppression of autophagy.

We used a FACs based approach where single cell measurements and images were unbiasedly analysed using Amnis® ImageStream®X Imaging Flow Cytometer, which can be found in Fig. 5h and Supplementary Fig. 6a-6b, as shown below with figure legends:

h Autophagy, measured using fluorescence activated cell sorting (FACS) of T47D cells in serum, left untreated (UT) or treated with FGF7 or FGF10 for 2h with autolysosomes stained with GFP, representative images and gating in Supplementary Fig. 6a-6b. Number of cells counted is indicated below graph, across a minimum of *N* = 4. *p*-value < 0.05 *, *p*-value < 0.0005 *** (one-way ANOVA with Tukey test)

b Representative images from Amnis® ImageStream®X Imaging Flow Cytometer analysis of T47D cells in serum, left untreated (UT) or treated with FGF7 or FGF10 for 2h with autolysosomes stained with GFP **c** Gating used in FACS analysis for quantification of high-GFP, high-autophagy cells shown in a, quantified in Fig. 5h

The updated results section is as shown below (from line 678):

To test whether FGF10-mediated FGFR2b recycling regulates autophagy, we assessed autophagy using four established methods: Fluorescence Activated Cell Sorting (FACS) analysis of cells with fluorescent staining of pre-autophagosome, autophagosomes and autolysosomes using a commercially available kit, acridine orange, widely used to stain lysosomes downstream of autophagy as a proxy for autophagy, western blotting of known markers for autophagy, and traditional immunofluorescence staining of autophagosomes, lysosomes and mature autolysosomes⁴¹. Both FACS analysis and acridine orange staining in HeLa-FGFR2bST, T47D and BT20 treated for 2 h with FGF10 and with FGF7 (as a negative control for FGFR2b recycling¹³) showed that FGF10 impaired autophagy compared to control in all cell lines, whereas FGF7 did not (Fig. 5h-i, Supplementary Fig 6a-b). Based on these results, we decided to use acridine orange staining to evaluate autophagy in our experimental conditions.

The details can be found in the methods as shown below (from line 1765):

FACS

Autophagy was assessed by Fluorescence Activated Cell Sorting (FACS) analysis using CYTO-ID® Autophagy detection kit (Enzo, ENZ-51031-0050), which labels pre-autophagosomes, autophagosomes and autolysosomes with a fluorescent dye, with minimal lysosomal staining. Samples were prepared following manufacturer's instructions. Briefly, T47D cells were seeded in 6-well plates and serum-starved overnight or kept in full media. Cells in PBS were left untreated (UT) or treated with FGF7 or FGF10 for 2h. Cells were then trypsinised, pelleted, washed in PBS and resuspended and incubated in CYTO-ID® green stain solution for 30 min, in dark at RT. Cells were pelleted, fixed in 4% formaldehyde for 20 min in dark and washed three times in PBS before being transferred to 96-well plate for imaging on Amnis® ImageStream®X Imaging Flow Cytometer and analysis using IDEAS® 6.0 software. Plots were gated to eliminate any aggregated cells. A second gate was added to count the single cell population. A third gate was then added to stratify the cell population with the highest GFP staining. The same gating was applied to each sample allowing a percentage of 'high GFP' cells to be calculated from each single cell population.

We used IF to look at LC3B-LAMP1 staining on individual cells (Supplementary Fig. 6d-e, 9a-9b).

d Co-localization analysis of LC3 (red) and LAMP1 (green) in HeLa_FGFR2bST (upper panels) and T47D (lower panels) grown in standard conditions (serum), in starvation medium (UT) or in starvation medium followed by stimulation with FGF7 or FGF10 for 2 hours. Scale bar, 30 μ m. Red arrowheads indicate LC3-positive vesicles (autophagosomes), white arrowheads indicate LAMP1-positive vesicles (lysosomes), and yellow arrowheads indicate LC3/LAMP1-positive vesicles (mature autolysosomes). **e** Quantification of the number of LC3-, LAMP1- or LC3/LAMP1-positive vesicles per nuclei (indicated below each graph). values represent median \pm st dev of N=3; p-value < 0.05 *, < 0.005 **, < 0.0005 *** (One-way ANOVA with Tukey)

The updated results section reads [from line 720]:

Finally, we observed an increase in LC3-positive autophagosomes in starved conditions and in FGF7, but not FGF10, stimulated HeLa-FGFR2bST and T47D cells (Supplementary Fig 6d-e). The number of LAMP1-positive lysosomes did not change in any condition, whereas the number of mature autolysosomes (LC3/LAMP1-positive vesicles) was higher in untreated compared to FGF7 stimulated cells and equal to zero upon FGF10 stimulation (Supplementary Fig 6 d-e), suggesting that starvation and FGF7 or FGF10 treatment differentially regulate the autophagy flux. The results from the four methods used to evaluate autophagy altogether suggest that autophagy regulation is FGFR2b-dependent and also requires FGFR2b recycling downstream of FGF10.

a Co-localization analysis of LC3 (red) and LAMP1 (green) in HeLa_FGFR2bST (upper panels) and T47D (lower panels) expressing wtRAB11 or DnRAB11 and stimulated with FGF10 for 2 hours after starvation (UT). Scale bar, 30 mm. Red arrowheads represent LC3-positive vesicles (autophagosomes), white arrowhead represent LAMP1-positive vesicles (lysosomes), and yellow arrowhead represent LC3/LAMP1-positive vesicles (mature autolysosomes). **b**. Quantification of the number of LC3, LAMP1- and LC3/LAMP1-positive vesicles per nuclei. Total number of cells is indicated below each graph. Values represent median \pm st dev of N=3. P-value < 0.005 ** P-value < 0.0005 ***

The updated results section reads [from line 1024]:

To Investigate how FGFR2b signalling partners at the recycling endosomes (e.g. ULK1) affected long-term FGFR2b responses during recycling, we tested the impact of impaired FGFR2b trafficking on FGF10-regulated responses. Firstly, we found that autophagy did not change or was slightly increased in FGF10-stimulated cells expressing DnRAB11, as shown by an increase in acridine orange staining in both HeLa-FGFR2bST and T47D (Fig. 7a). Furthermore, the number of LC3-positive autophagosomes and the number of LC3/LAMP1-positive mature autolysosomes, but not that of LAMP1-positive lysosomes, increased in cells expressing DnRAB11 compared to cells expressing wtRAB11 upon FGF10 stimulation (Supplementary Fig 9a-b).

We have expanded the western blot analysis to look at more markers of autophagy, including LC3B, BECLIN1, p-BECLIN1 and p62 (Fig. 5j-5k, 7b, Supplementary Fig. 9e). We have also expanded the western blot analysis related to mTOR suppression of autophagy with p-RAPTOR, RAPTOR, ULK1 and p-ULK1 (Fig. 6k, 7b, Supplementary Fig. 9b)

Fig. 5j-5k:

j Immunoblot analysis ($N \geq 3$) with the indicated antibodies of the effect of serum starvation and FGF treatment on autophagic markers in T47D. LC3B 2 is the lipidated form. Cells were untreated or treated with FGF7 or FGF10 after starvation. *k* Immunoblot analysis ($N \geq 3$) with the indicated antibodies of HeLa_FGFR2bST, T47D, and BT20 treated or not with FGF7 or FGF10 for 2 h.

Fig 6k:

Immunoblot analysis ($N=3$) with indicated antibodies of T47D transfected with GFP or DnRAB11 and left either untreated (UT) or treated with FGF10 for the indicated time points.

Fig 7b. Immunoblot analysis (N=3) with the indicated antibodies of HeLa_{FGFR2bST} transfected either with GFP or DnRAB11 and left either untreated (UT) or treated as indicated.

Supplementary Fig. 9e:

Immunoblot analysis with the indicated antibodies of T47D cells pre-treated for 2 h with FGFR inhibitor (FGFRi) PD173074, ULK1 inhibitor (ULK1i) ULK101, ULK1/2 inhibitor (ULK1/2i) SBI0206965, or mTOR inhibitor (mTORi) Rapamycin and stimulated with FGF10 for 2 h. The lysates relate to Fig. 7c-7e.

The updated results section from line 711 now reads:

As we starved cells before stimulation with FGFs and starvation is known to increase autophagy⁴³, we checked the levels of known autophagy markers in starved cells followed or not by stimulation with either serum (as control), FGF7 and FGF10 by western blots. The lipidated form (2) of the autophagosome-formation associated microtubule-associated

proteins 1A/1B light chain 3B (LC3B)⁴⁴ was suppressed to levels seen in serum-treated cells by FGF10 treatment alone (Fig. 5j). This FGF10-, but not FGF7-dependent decrease in the levels of lipidated LC3B (2) was seen in HeLa-FGFR2bST and BT20 cells as well, alongside a decrease in active BECLIN1 phosphorylated on S93 (Fig. 5k), another mediator of autophagosome formation and maturation^{41,45}. Similarly we found that p62 is stabilized under conditions of increased autophagy⁴¹.

We have also performed the suggested experiment using BafilomycinA which is included below. We measured autophagic flux in the presence of DMSO left UT, treated with FGF7 or FGF10 for 2h. Autophagy was measured using acridine orange (top panel) or using an Autophagy Assay Kit (Sigma-Aldrich, MAK138) (lower panel). We showed that FGF10 is unable to further suppress autophagic flux in the presence of Bafilomycin A. This suggests FGF10 suppresses autophagy through similar mechanisms as Bafilomycin A.

11. This should also be performed by IF in parallel, for autophagosomes and lysosomes. Commonly, LC3/Lamp1 co-staining or tandem-LC3 is performed in parallel with the IF conditions. From quantification of LC3 puncta, vs Lamp1 vs LAMP1+LC3 puncta, the number of autophagosomes, mature autolysosomes, and mature autolysosomes in blocked autophagic conditions, can be determined. Together, this will allow the author to demonstrate robustly the effect that the modulations have on autophagic flux.

We thank the reviewer for this suggestion, and we analysed by immunofluorescence the staining of LC3-, LAMP1-, and LC3/LAMP1-positive vesicles in growing cells, starved cells, and cells treated with FGF7 or FGF10 (the latter also in cells expressing DnRAB11). Our results show that FGF10 suppressed autophagy (no LC3 staining detected) in all the tested conditions except in the presence of DnRAB11. We have quantified the results by counting the number of LC3-, LAMP1-, or LC3/LAMP1-positive vesicles as explained in the updated Method section (see below). This confirms the importance of FGFR2b recycling for autophagy regulation and the results obtained using acridine orange (Fig 5, 7, Supplementary Fig 6 and 9). We have incorporated these findings in Supplementary Fig 6 ad 9 and in the result section.

The updates to Supplementary Fig. 6 and figure legend:

f Co-localization analysis of LC3 (red) and LAMP1 (green) in HeLa_FGFR2bST (upper panels) and T47D (lower panels) grown in standard conditions (serum), in starvation medium (UT) or in starvation medium followed by stimulation with FGF7 or FGF10 for 2 hours. Scale bar, 30 mm. Red arrowheads indicate LC3-positive vesicles (autophagosomes), white arrowheads indicate LAMP1-positive vesicles (lysosomes), and yellow arrowheads indicate LC3/LAMP1-positive vesicles (mature autolysosomes). **g** Quantification of the number of LC3-, LAMP1- or LC3/LAMP1-positive vesicles per nuclei (indicated below each graph). values represent median \pm st dev of N=3; p-value < 0.05 *, < 0.005 **, < 0.0005 *** (One-way ANOVA with Tukey)

The results (from line 720) now reads:

Finally, we observed an increase in LC3-positive autophagosomes in starved conditions and in FGF7, but not FGF10, stimulated HeLa-FGFR2bST and T47D cells (Supplementary Fig 6d-e). The number of LAMP1-positive lysosomes did not change in any condition, whereas the number of mature autolysosomes (LC3/LAMP1-positive vesicles) was higher in untreated compared to FGF7 stimulated cells and equal to zero upon FGF10 stimulation (Supplementary Fig 6d-e), suggesting that starvation and FGF7 or FGF10 treatment differentially regulate the autophagy flux.

Supplementary Fig. 9. Inhibiting FGFR2b recycling leads to dysregulated autophagy and an altered balance of proliferation and cell death. **a** Co-localization analysis of LC3 (red) and LAMP1 (green) in HeLa_FGFR2bST (upper panels) and T47D (lower panels) expressing wtRAB11 or DnRAB11 and stimulated with FGF10 for 2 hours after starvation (UT). Scale bar, 30 mm. Red arrowheads represent LC3-positive vesicles (autophagosomes), white arrowhead represent LAMP1-positive vesicles (lysosomes), and yellow arrowhead represent LC3/LAMP1-positive vesicles (mature autolysosomes). **b**. Quantification of the number of LC3, LAMP1- and LC3/LAMP1-positive vesicles per nuclei. Total number of cells is indicated below each graph. Values represent median \pm st dev of N=3. P-value < 0.0005 ***

The results (from line 1029) have been updated:

Furthermore, the number of LC3-positive autophagosomes and the number of LC3/LAMP1-positive mature autolysosomes, but not that of LAMP1-positive lysosomes, increased in cells expressing DnRAB11 compared to cells expressing wtRAB11 upon FGF10 stimulation (Supplementary Fig 9a-b).

We have updated the methods (from line 1859):

Quantification of LC3-, LAMP1-, and LC3/LAMP1-positive vesicles was performed manually using Image J. For N=3 independent experiment we analysed between 15 and 25 cells per image by adjusting the threshold of each channel to a value equal to 50 followed by particles counting. The number of LC3- or LAMP1-positive vesicles was divided by the number of DAPI-stained nuclei to obtain the ratio of autophagosomes and lysosomes, respectively. We manually counted the number of LC3/LAMP1-positive vesicles using the merge image to determine the ratio of mature autophagosomes.

REVIEWERS' COMMENTS

Reviewer #1 (Remarks to the Author):

Overall, the authors carefully addressed all my comments. However, the proteomics data presentation is generally redundant and not precise, especially for the following two issues:

1. In manuscript line 417~420 and Fig 5f, the author described the identification of ULK1 pS638 in proximal phosphoproteome and validated this phosphorylation sites and functions in further experiment. But in Table S6, the intensities of ULK1 pS638 were all "NA" in all the proximal samples. Similarly, the RRAS2 pS186 shown in Fig.5f also has no valid value in proximal sample. The authors should explain this key data seriously since ULK1 pS638 identified in the "proximal phosphoproteome" is the basis of the following biological findings.
2. Identification and repeatability of the individual protein or site data mentioned in following biological studies should be provided separately in SI figures to confirm the data quality.

Reviewer #2 (Remarks to the Author):

Dear Authors,

Thank you very much for detailed answers on all of my points and for improving your manuscript. Congratulations on your excellent work.

Reviewer #3 (Remarks to the Author):

The authors have made many changes to the manuscript, so I thank them for their effort. As a result, the paper has improved in clarity significantly. Thank you for additional effort, and I hope that this paper is well received.

Minor Comment 1:

Please make sure that the manuscript is thoroughly proof-read, there are some typos in the new parts of text too. Please also make sure to critically evaluate the English grammar, some errors are present such as “high light” (should be highlight) and “acidic wash” (acid wash is commonly used).

Standardise fig call outs: refer to main figs first then supp (Line 441)

Minor comment 4:

Fig 1 legend: “left untreated (control)” then labelled in figure UT. Make consistent, and explain any abbreviation used in figure. Ie. “Untreated Control (UT)”. As a note for general good practice, controls should not be untreated, but should always include the same volume of vehicle as the treated.

Line 772: You don’t include in the methods what the vehicle is for FGF10, this should be mentioned (also for other compounds)

Minor comment 6:

Y axis labels are not clear. % of autophagy does not explain what the graph is measuring. Likewise, “measurement by acridine orange staining”. What did you measure? Please be specific. Ie. Number of AO positive vesicles. Same with “Apoptosis by cleaved caspase-3”, should read “increase in cleaved caspase 3” (or caspase 3 levels)... then in the main text (ie line 366 for autophagy), include an explanation eg “apoptosis measured by increase in caspase-3 by x approach”. Once again, this is general good practice, and should be followed throughout.

Major point 3: Well determined ROI: Depending on what?

Major comment 3: Thank you for this detailed description. Looking back in the previous papers I see that you have tested also ubiquitination/degradation. I think that as a standard of good lab practice, these experiments should be run along-side any new conditions (including +/- Baf to inhibit autophagy as well as proteasomal degradation). But this is sufficient.

Major comment 5

The authors have now removed all zooms... I think that zooms are important to demonstrate the coloc described. My point was that zooms should be included for all conditions, to show the presence and absence of colocalization, as this is crucial to see that imaging has been performed robustly. The zoomed images shown must also be treated exactly as in the non-zoomed images. Ie., the exposure/gain/brightness etc etc must be the same in both zoomed and non-zoomed images. The images originally shown appeared to have brightness increased above that of the un-zoomed image, and the result did not clearly show the colocalisation. I think that the clarity is increased greatly by including zoomed areas, and they should thus be included.

Fig 6d: Please apply comments re. coloc here too. The top left point of coloc in FGF10 40’ spot coloc with pULK1 looks convincing. However, the low right arrow in the same condition and the arrow in

the bottom panel for DnRAB11 are not convincing, but may be with suitable zooms to demonstrate the colocalisation more clearly.

Legends:

The experimental approach shown in the figure should be included in each legend. (ie: "Fig 1. Scanning confocal of x cells shows...")

I don't think that information like "(EEA1 is a marker for EEs)" should be in the figure legends (esp. incl. references). Put in main text and remove from here. Make consistent throughout.

White arrowheads indicate colocalisation: I suggest that you use a different colour or sign for lack of colocalisation ie. Arrowheads vs full arrows, or arrows filled with white vs filled with black

Confirm all scales of images are equal, otherwise put separate scale bars for each image (120' FGF10 in FGFR2b wtRAB11 and 120' UT FGFR2b/DnRAB11 same as rest of images?). May be useful to put scale bars on all separate images.

The authors draw conclusions on the activation of autophagy from acridine orange staining and LC3 I-II levels. The authors also use inhibitors for early autophagy (ULK1, ULK1/2, mTOR) but they do not use late inhibitors to visualise the build-up of cargo/LC3

Fig 6. I presume wtRAB11 and DNRAB11 are GFP-tagged, thus the GFP lane represents these. Please mark it on the figure.

The authors address the IF for Lamp/mTOR, but say this is in Fig 6i – there is no IF here. However that shown in Supp Fig 8h looks very good.

In Fig legend for Supp Fig. 7, it is quoted: see Fig 7d inset table for inhibitor target. This is not there anymore

Point 10: Thank you for strengthening the autophagy results. I appreciate that you have added so many more autophagy assays to the paper, and I appreciate that these are very time consuming. The key element in any autophagy experiment is to show flux, which is done by adding a late autophagy inhibitor (inhibition of lysosomal degradation to accumulate autophagy machinery (LC3II) and cargo) eg. BafA1, where you compare control and experimental conditions in the presence and absence of BafA1. This will distinguish whether FGF10 indeed suppresses autophagy or whether it increases autophagic flux. Autophagosome or lysosome presence alone can cause confusion, as an increase in flux could also cause a decrease in autophagosome marker (ie. LC3II), as the autophagosomes mature and are degraded at a higher rate than normal, while suppression would also cause a decrease in autophagosome marker as autophagosomes are not formed. Including treatment with Baf clearly distinguishes the two: during autophagy suppression in the presence of Baf, there will be very little autophagosome staining (LC3II) while during activation, there would be a massive accumulation of LC3II, in the presence of Baf. (See Fig 5C in Klionsky et al (al. al. al.) 2012 Autophagy <https://doi.org/10.4161/auto.19496>. Or Fig 5 in Thomé et al. JCS 2016 doi.org/10.1242/jcs.195057)

This is most simply demonstrated by WB for LC3II (and p62, but not essential) as you have shown on page 57 of the rebuttal. However you have not added Baf or a late stage inhibitor. The AO measurements shown on page 58 of the rebuttal indeed shows that in the presence of Baf, acidification of lysosomes is lost, and that this should be exacerbated in the case that FGF10 induced autophagy, but as it is equal in Baf + FGF10, it does indicate that FGF10 does not do so. You seem to have taken a long and more complicated way around a simple question, and I would be much happier to see a simple western with all conditions +/- Baf, but these results do support your conclusions.

Point-by-point responses to reviewer's comments.

Our responses to reviewer comments are provided below in **blue font** after each comment and changes to the manuscript are visualized by *italic blue font*.

REVIEWER COMMENTS

Reviewer #1 (Remarks to the Author):

Overall, the authors carefully addressed all my comments. However, the proteomics data presentation is generally redundant and not precise, especially for the following two issues:

1. In manuscript line 417~420 and Fig 5f, the author described the identification of ULK1 pS638 in proximal phosphoproteome and validated this phosphorylation sites and functions in further experiment. But in Table S6, the intensities of ULK1 pS638 were all "NA" in all the proximal samples. Similarly, the RRAS2 pS186 shown in Fig.5f also has no valid value in proximal sample. The authors should explain this key data seriously since ULK1 pS638 identified in the "proximal phosphoproteome" is the basis of the following biological findings.

We appreciate the reviewer's detailed examination and criticism of our work. To improve clarity in the presentation of our proteomic findings we have removed the following panels: Fig. 4b, c, d, g and Fig 5a, d and combined panels 4j and 5c. Updated Figures 4 and 5 are provided below.

Fig. 4. Spatially resolved proteomics and phosphoproteomics reveal FGFR2b-dependent regulation of mTOR signalling and autophagy. **a** Workflow of the spatially resolved proteomics and phosphoproteomics experiments in HeLa cells expressing the indicated constructs. **b** Summarised data analysis pipeline of proximal phosphoproteome data. **c** Cluster analysis of the proximal phosphoproteome from the indicated conditions normalized to the proximal phosphoproteome of HeLa_FGFR2bST GFP-APEX2 for each timepoint. Phosphorylated sites upregulated at 40 min stimulation with FGF10 in both

HeLa_FGFR2bST RAB11-APEX2 and HeLa_FGFR2b-APEX2ST RAB11 are marked as the FGFR2b Recycling Proximal Signalling Cluster. d Overlap of proteins and phosphorylated proteins detected in the proximal proteome and phosphoproteome samples, respectively. *e* Distribution of the phosphorylated sites *s*, 77,4% of which were found in the FGFR2b Recycling Proximal Signalling Cluster. *f* Overlap between the phosphorylated sites upregulated in the global phosphoproteome upon FGF10 stimulation and the phosphorylated sites upregulated in the FGFR2b Recycling Proximal Signalling cluster from the proximal phosphoproteome (Fig. 4c). *g* Phosphorylated sites identified on FGFR2 and EGFR in the global (blue light) or in the proximal phosphoproteome (red) or in both (blue), and in the phosphoproteome from HeLa_FGFR2bST cells expressing GFP, GFP-DnRAB11 or GFP-DnDNM2 (Fig. 2a). Light blue with green border indicates phosphorylated sites found in internalization response clusters and dark blue with green border indicates sites found in recycling response clusters (Fig. 2d). *h* KEGG pathway enrichment (calculated with Fishers Exact Test and FDR adjustment) of the phosphorylated sites found in the FGF10 global phosphoproteome (blue light), FGFR2b Recycling Proximal Signalling Cluster (red) and among the phosphorylated sites on proteins quantified at the proteome level from *e* (orange). Source data are provided as a Source Data file.

Fig. 5. FGFR2b regulates mTOR signalling and autophagy from the recycling endosomes. *a* Subnetwork of proteins annotated to mTOR pathway or autophagy based on KEGG analysis from Fig. 4h. Node colouring indicates whether the phosphorylated protein or the phosphorylated sites from the KEGG term “autophagy” were found in

global, proximal phosphoproteome or both. Sites and proteins also quantified in Supplementary Data 2 have a green border. **b** Immunoblot analysis (N=3 independent biological replicates) with indicated antibodies of candidate phosphorylated proteins from subnetwork (Fig. 5a). T47D were transfected with RAB11-APEX2 (T47D_RAB11-APEX2) stimulated with FGF10 for the indicated time points. UT, treatment with vehicle as control. Non proximal and proximal samples represent the supernatant and the pull-down following enrichment of biotinylated samples with streptavidin beads, respectively, and run against total lysates (total). **c** Autophagy measured using fluorescence activated cell sorting (FACS) of T47D cells in serum, treated with vehicle (UT) or with FGF7 and FGF10 for 2h. Representative images and gating are shown in in Supplementary Fig. 6b-c. Number of cells counted is indicated below graph. N = 3 independent biological replicates. p-value < 0.05 *, p-value < 0.0005 *** (one-way ANOVA with Tukey test) **d** Autophagy (measured by staining of autolysosomes with acridine orange) of HeLa_FGFR2bST, T47D, and BT20 treated with vehicle (UT) or with FGF7 or FGF10. N = 3 independent biological replicates where at least 6 treated wells of cells were counted. p-value < 0.001*** (one-way ANOVA with Tukey test). **e** Immunoblot analysis (N>=3 independent biological replicates) with the indicated antibodies of the effect of serum starvation and FGF treatment on autophagic markers in T47D. LC3B 2 is the lipidated form. Cells were treated with vehicle (UT), FGF7 or FGF10. **f** Immunoblot analysis (N>=3 independent biological replicates) with the indicated antibodies of HeLa_FGFR2bST, T47D, and BT20 treated or not with FGF7 or FGF10 for 2 h. Source data are provided as a Source Data file.

We thank the reviewer for pointing out the mistake in Supplementary Table 6 (now SupplementaryData6). To investigate the invalid values, we went back to our analysis pipeline and found that we had mistakenly performed imputation of the data at the wrong step in our pipeline, as detailed in Figure for Reviewer 1.1. We profusely apologise for this genuine mistake.

*Post filtering for low confidence identifications/contaminants and normalisation.

Figure for Reviewer 1.1: Details of the incorrect imputation process and the corrected process.

Although S638 phosphorylation on ULK1 did not come up as significant in the proximal phosphoproteome anymore, but other phosphorylated sites on ULK1 did (SupplementaryData6), the conclusions of our combined SRP and biochemical approach regarding autophagy being repressed downstream of FGF10 signalling via S638 phosphorylated ULK1 recruited to the recycling endosomes have not changed for the following reasons:

1. After correcting the imputation error, we found an increase in the number of significantly regulated sites identified in the FGFR2b Recycling Proximal Signalling Cluster (Figure for Reviewers 1.2). This was likely due to the use of FDR adjustment, which is sensitive to skewed p -value distribution, at the correct step of the data analysis.

Figure for Reviewer 1.2

2. The overlap of proteins and phosphorylated proteins (detected in the proximal proteome and phosphoproteome samples) is now higher than in the previous analysis (588 in the overlap), and a substantial number are still only found in the set labelled 'Proteins with Phospho(STY) Sites'. Therefore, our conclusion related to the importance of performing the double enrichment step to reveal spatially resolved, phosphorylated signalling partners of FGFR2b was not altered by the corrected analysis (Figure for Reviewer 1.3).

Figure for Reviewer 1.3

3. Of the overlapping 588 proteins described above, 77.4% were found in the FGFR2b Recycling Proximal Signalling cluster. This is comparable to the 71.3% found in the original analysis and creates a proportionally larger subset of phosphorylated proteins to compare to as a high-confidence subset representing phosphorylated proteins proximal to FGFR2b at RAB11-positive recycling endosomes (Figure for Reviewer 1.4).

Figure for Reviewer 1.4

4. Our assessment of whether expression of APEX2 tags altered the quantification of the phosphoproteome in Supplementary Fig. 5j also does not change. In the UT and FGF10-treated samples, only 2 and 109 phosphorylated sites respectively were found to be statistically different due to the APEX2 tags, which was still well within the 5% chance of statistical error (Figure for Reviewer 1.5).

Original Supplementary Fig. 5j:

Comparison of Global STY	Test	FDR < 0.05
GFP-APEX2 UT, FGFR2-APEX2 UT	T-test + FDR adjustment	1 / 11799 quantified sites
GFP-APEX2 40', FGFR2-APEX2 40', RAB11-APEX2 40	One-way ANOVA + FDR adjustment	96 / 11799 quantified sites

Amended Supplementary Fig. 5j:

Comparison of Global STY	Test	FDR < 0.05
GFP-APEX2 UT, FGFR2-APEX2 UT	T-test + FDR adjustment	2 / 11799 quantified sites
GFP-APEX2 40', FGFR2-APEX2 40', RAB11-APEX2 40	One-way ANOVA + FDR adjustment	109 / 11799 quantified sites

Figure for Reviewer 1.5

5. The size of the overlap between the phosphorylated sites upregulated in the global phosphoproteome upon FGF10 stimulation and the phosphorylated sites upregulated in the FGFR2b Recycling Proximal Signalling cluster from the proximal phosphoproteome remains comparable: in the original analysis it was 69 and in the amended analysis it is 107. Our conclusion remains that we have enriched for two very different populations of FGFR2b signalling partners in the global and proximal phosphoproteome (Figure for Reviewer 1.6).

Figure for Reviewer 1.6

6. FGFR2 and EGFR phosphorylated sites has the same pattern of regulation before and after error corrections except for EGFR_S991 which was no longer identified as a significantly regulated site neither in the proximal nor in the global phosphoproteome (Figure for Reviewer 1.7).

Figure for Reviewer 1.7

7. We performed the KEGG enrichment analyses between the different proximal and global profiles together and found that there were 6 terms in common between the two proximal profiles, most likely being the pathways regulated by FGFR2b at the recycling endosomes (Figure for Reviewer 1.8). Among these pathways we identified autophagy, thus confirming our previous findings. mTOR signalling was enriched in both the proximal and the global phosphoproteome. We therefore maintain our hypothesis that mTOR signalling is integrated at the global and proximal level, before converging to regulate autophagy in proximity of the recycling endosomes, most likely via ULK1 phosphorylation and regulation.

Original (was Fig. 4j)

Original (was Fig.5c)

Amended (Fig. 4h)

Figure for Reviewer 1.8

8. The subnetwork which included proteins annotated to mTOR pathway or autophagy based on KEGG terms is now larger and more connected. A few nodes from the original figure, including the RRAS2 and ULK1 sites highlighted by the reviewer, are no longer present in the proximal phosphoproteome, whilst a further 22 phosphorylated sites have been added (Figure for Reviewer 1.9 and 1.10). However, the phosphorylated site S863 on RPTOR - which is upstream ULK1 phosphorylation on S638 (Zachari, M. & Ganley, I. G. The mammalian ULK1 complex and autophagy initiation. *Essays Biochem* 61, 585-596, doi:10.1042/EBC20170021, 2017) and then

we have validated in the previous version of the manuscript - still belongs to the proximal phosphoproteome, together with other phosphorylated sites of the mTOR pathway upstream of ULK1 regulation.

Figure for Reviewer 1.9

Original (was Fig. 5f)

Amended (Fig. 5a):

Figure for Reviewer 1.10

9. The SRP approach picks up a clear correlation between autophagy, mTOR signalling (including phosphorylated ULK1 on different sites) and recycling endosomes (updated Figures 4, 5) both before and after the error correction. Several biological assays showed regulation of autophagy downstream of FGF10 in a recycling-, mTOR-and ULK1-dependent manner (updated Figures 5, 6), thus confirming our initial conclusions.
10. Having found the phosphorylation of the mTOR regulator RPTOR in the proximal phosphoproteome downstream of FGFR2b signalling (Figure 5a) prompted us to look at the phosphorylation of ULK1 on S638, which is widely studied as regulator of autophagy downstream of mTOR signalling (Zachari, M. & Ganley, I. G. The mammalian ULK1 complex and autophagy initiation. *Essays Biochem* 61, 585-596, doi:10.1042/EBC20170021, 2017). We validated S638 phosphorylated ULK1 as recruited to recycling endosomes in FGF10 stimulated cells for the first time and upstream of autophagy regulation using several assays (Figure 7). Together with the lack of regulation of ULK1 phosphorylation on S638 in the global phosphoproteome (Figure 5 a-b, SupplementaryData6), our data shows that correcting the mistake has not changed the validity of our conclusions.

The text has been updated as follows from line 373:

“ To reveal the phosphorylated interactome of FGFR2b when localized at the recycling endosomes we normalized the log₂ transformed data from the control and from the FGF10-treated FGFR2b-APEX2 and RAB11-APEX2 samples against the corresponding time points of the GFP-APEX2 samples (Fig. 4b, Supplementary Fig. 5k). Hierarchical clustering of the normalized data revealed a cluster of phosphorylated sites enriched in both the FGFR2b-APEX2 and the RAB11-APEX2 samples treated with FGF10, hereby the FGFR2b Recycling Proximal Signalling Cluster (Fig. 4c). We noticed an overlap of 588 proteins between the phosphorylated proteins identified in the proximal phosphoproteome and the proteins identified in the proximal proteome which would most likely represent phosphorylated FGFR2b partners at the recycling endosomes (Fig. 4d). The relatively small overlap (588 over 1099 proteins with phosphorylated sites) may indicate the importance of performing the double enrichment step to reveal spatially resolved, phosphorylated signalling partners of the bait of interest. Interestingly, of the 961 phosphorylated sites on the 588 overlap proteins, 77.4% (743) was also found in the FGFR2b Recycling Proximal Signalling Cluster (Fig. 4e). *Furthermore, when we compared the FGFR2b Recycling Proximal Signalling Cluster with the FGF10-regulated phosphorylated sites from the global phosphoproteome, we found only a small overlap of 107 phosphorylated sites (Fig. 4f). FGFR2 and EGFR were found phosphorylated in this overlap (Fig. 4g). One of the catalytic sites of FGFR2 (Y656)²⁴ was also identified as part of the internalization response cluster (Fig. 2), corroborating the role of this site for FGFR2 trafficking⁴⁰. Interestingly, T693 on EGFR was found phosphorylated only in the proximal phosphoproteome (Fig. 4g), consistent with its role in regulating FGFR2b recycling at the recycling endosomes⁴¹. These findings, altogether, indicate that the SRP approach*

capably distinguished the FGFR2b proximal phosphoproteome enriched at RAB11-positive endosomes from the FGFR2b global phosphoproteome.

KEGG pathway enrichment analysis of the FGF10 global phosphoproteome, FGFR2b Recycling Proximal Signalling Cluster, and the subset of the latter overlapping with the proximal proteome (orange in Fig 4h) revealed six terms specifically enriched in the FGFR2b proximal datasets, among which autophagy. Interestingly, mTOR signalling pathways, which suppresses autophagy¹⁹, was enriched in both the global and proximal FGFR2b phosphoproteome (Fig 4h). We therefore hypothesised that mTOR signalling may be integrating at the global and the proximal level downstream of FGFR2b activation, before converging to regulate autophagy in the proximity of the recycling endosomes during FGFR2b trafficking.

FGFR2b recycling suppresses mTOR/ULK1-dependent autophagy

To investigate the link between FGFR2b proximal signalling partners and autophagy regulation downstream of mTOR signalling, we created a sub-network by extracting those proteins annotated to either autophagy or mTOR signalling pathway in KEGG (Fig. 4h). We found several components upstream of mTOR, including RAF1, MAP2K2, RPS6, as well as the mTOR subunits RPTOR and RICTOR, and several proteins known to regulate autophagy via mTOR signalling, among which SGK1, SQSTM1 (also known as p62), TSC1, and the kinase ULK1⁴² (Fig. 5a). A subset of candidates within this network, spanning the proximal and global phosphoproteome were confirmed by immunoblot analysis in T47D to match the patterns identified by the SRP approach (Fig. 5b, Supplementary Fig 6a, Supplementary Data 6). Interestingly, RPTOR phosphorylated at S863 was spatially restricted at the recycling endosomes (Fig 5b), confirming the link between recycling endosomes and autophagy regulation downstream of FGF10/FGFR2b signalling. “

The text has been updated as follows from line 942:

“As we identified ULK1 and RPTOR phosphorylation in the proximal phosphoproteome (Fig. 5a), we next investigated whether ULK1 phosphorylated downstream of mTOR – for instance on S638 which is known to suppress autophagy⁵¹ - localized at the recycling endosomes during FGFR2b recycling. In both HeLa-FGFR2bST and T47D cells ULK1 was recruited and phosphorylated on S638 in proximity of both FGFR2b and RAB11 as shown upon streptavidin beads enrichment of biotinylated proteins followed by immunoblotting (Figure 7a-b, Supplementary Fig 7a).”

2. Identification and repeatability of the individual protein or site data mentioned in following biological studies should be provided separately in SI figures to confirm the data quality.

Bar charts for the following phosphorylated sites, which were regulated in the global, proximal, or both the phosphoproteomes and which we validate by immunoblotting in Figure 5b, have been provided in Supplementary Fig 6a.

Supplementary Fig 6. FGFR2b signalling and autophagy. a Bar graph showing the log₂ intensity of selected phosphorylated sites and proteins from Supplementary Data 6. Global p-value < 0.05 * (two-sided permutation t-test with FDR adjustment); proximal p-value < 0.05 *, (One-way ANOVA with Tukey).

Reviewer #2 (Remarks to the Author):

Dear Authors,

Thank you very much for detailed answers on all of my points and for improving your manuscript. Congratulations on your excellent work.

We appreciate the reviewers' positive comments on our manuscript.

Reviewer #3 (Remarks to the Author):

The authors have made many changes to the manuscript, so I thank them for their effort. As a result, the paper has improved in clarity significantly. Thank you for additional effort, and I hope that this paper is well received.

We thank the reviewer for his/her detailed assessment of our work, and we appreciate the reviewers' positive comments on our manuscript. We have addressed the remaining concerns of the reviewers below and have changed the manuscript accordingly.

Minor Comment 1:

Please make sure that the manuscript is thoroughly proof-read, there are some typos in the new parts of text too. Please also make sure to critically evaluate the English grammar, some errors are present such as "high light" (should be highlight) and "acidic wash" (acid wash is commonly used). Standardise fig call outs: refer to main figs first then supp (Line 441)

We have checked the manuscript to the best of our ability.

Minor comment 4: Fig 1 legend: “left untreated (control)” then labelled in figure UT. Make consistent, and explain any abbreviation used in figure. I.e. “Untreated Control (UT)”. As a note for general good practice, controls should not be untreated, but should always include the same volume of vehicle as the treated.

We have explained in all the figure legends that UT corresponds to samples treated with vehicle as control. The appropriate vehicle controls are described in Methods in the appropriate section.

Line 772: You don’t include in the methods what the vehicle is for FGF10, this should be mentioned (also for other compounds)

We have added this information in Methods.

Minor comment 6: Y axis labels are not clear. % of autophagy does not explain what the graph is measuring. Likewise, “measurement by acridine orange staining”. What did you measure? Please be specific. I.e. Number of AO positive vesicles. Same with “Apoptosis by cleaved caspase-3”, should read “increase in cleaved caspase 3” (or caspase 3 levels)... then in the main text (ie line 366 for autophagy), include an explanation eg “apoptosis measured by increase in caspase-3 by x approach”. Once again, this is general good practice, and should be followed throughout.

We have updated the text, figures, and figure legends as suggested.

Major point 3: Well determined ROI: Depending on what?

We removed “well determined” from the Methods, as it is explained in the following sentence what we mean by ROI in regions (cells) and how we used the ImageJ ROI function.

Major comment 3: Thank you for this detailed description. Looking back in the previous papers I see that you have tested also ubiquitination/degradation. I think that as a standard of good lab practice, these experiments should be run along-side any new conditions (including +/- Baf to inhibit autophagy as well as proteasomal degradation). But this is sufficient.

Thank you for your positive comment.

Major comment 5. The authors have now removed all zooms... I think that zooms are important to demonstrate the colocalization described. My point was that zooms should be included for all conditions, to show the presence and absence of colocalization, as this is crucial to see that imaging has been performed robustly. The zoomed images shown must also be treated exactly as in the non-zoomed images. I.e., the exposure/gain/brightness

etc etc must be the same in both zoomed and non-zoomed images. The images originally shown appeared to have brightness increased above that of the un-zoomed image, and the result did not clearly show the colocalisation. I think that the clarity is increased greatly by including zoomed areas, and they should thus be included. Fig 6d: Please apply comments re. coloc here too. The top left point of coloc in FGF10 40' spot coloc with pULK1 looks convincing. However, the low right arrow in the same condition and the arrow in the bottom panel for DnRAB11 are not convincing, but may be with suitable zooms to demonstrate the coloc more clearly.

We have added Zoom images corresponding to the region indicated by the arrowheads in Figures 1a, 3c, updated 7d, updated Supplementary Fig 7 f-h. The Zoom images have been treated as the non-zoomed ones.

Fig. 1. FGFR2b activation is not affected by receptor sub-cellular localization. a Representative confocal image of the presence of FGFR2b (red) in the cytoplasm and of FGFR2b recycling to the plasma membrane in HeLa cells stably transfected with FGFR2b-HA (HeLa_FGFR2bST), expressing eGFP-RAB11a (wtRAB11), dominant negative eGFP-RAB11a_S25N (DnRAB11), or dominant negative dynamin-2_K44A-eGFP (DnDNM2) (green), and treated with FGF10 for 0, 40 and 120 min.. UT, treatment with vehicle as control). Early endosome antigen 1 (EEA1) is in blue. Scale bar, 5 μ m. Zoomed images of the regions indicated by the arrowheads (scale bar, 50 μ m) and single channels for FGF10-stimulated cells for 0, 40 and 120 min. are shown in the inset and on the right, respectively. White arrowheads indicate co-localization and pink arrowheads indicate lack of co-localization. **b** Quantification of the co-localization of stimulated FGFR2b (red pixels) with GFP-tagged proteins (green pixels) indicated by red-green pixel overlap fraction (left panel). Quantification of the co-localization of FGFR2b (red pixels) with EEA1 (blue pixels) indicated by red-blue pixel overlap fraction (right panel). Representative images are shown in 1a. Values represent median \pm SD from N=3 where we analysed between 2 and 5 cells for each N; *** p -value < 0.0005 (one-sided students t-test)²². **c** Immunoblot analysis (N \geq 3 independent biological replicates) with the indicated antibodies of HeLa_FGFR2bST cells expressing GFP, DnRAB11 or DnDNM2 treated with FGF10 for 0, 8 and 40 min. UT, treatment with vehicle as control. Source data are provided as Source Data file.

Fig. 3. APEX2 tagged-FGFR2b and RAB11a identifies compartment-specific signalling partners upon FGF10 stimulation. **a** Schematic underlying the Spatially Resolved Phosphoproteomics (SRP) approach. **b** Immunoblot analysis ($N \geq 3$ independent biological replicates) with the indicated antibodies of HeLa_FGFR2bST (right) or HeLa_FGFR2b-APEX2ST (left) stimulated with FGF10 for 0, 1, 8, 40, 60, or 120 min. UT, treatment with vehicle as control. **c** Representative confocal images of FGFR2b (red) internalization in the cytoplasm and FGFR2b recycling to the plasma membrane in HeLa_FGFR2b-APEX2ST, expressing eGFP-RAB11a (wtRAB11), dominant negative eGFP-RAB11a_S25N (DnRAB11), or dominant negative dynamin-2_K44A-eGFP (DnDNM2) (green), and treated with FGF10 for 0 and 40 min. UT, treatment with vehicle as control. Early endosome antigen 1 (EEA1) is visualized in blue.²². Scale bar, 5 μ m. Zoomed images of the region indicated by the arrowheads (scale bar, 50 μ m) and single channels *t* for FGF10-stimulated cells for 0 and 40 min. are shown in the inset and on the right, respectively. White arrowheads indicate co-localization and pink arrowheads indicate lack of co-localization. **d** Quantification of the co-localization of stimulated FGFR2b (red pixels) with GFP-tagged proteins (green pixels) indicated by red-green pixel overlap fraction (top panel). Quantification of the co-localization of FGFR2b (red pixels) with EEA1 (blue pixels) indicated by red-blue pixel overlap fraction (bottom panel). Representative images are shown in *c*. Values represent median \pm SD from $N=3$ independent biological replicates where we analysed between 2 and 5 cells for each *N*; *** *p*-value < 0.0005 (one-sided students *t*-test). **e** Immunoblot analysis ($N \geq 3$ independent biological replicates) with the indicated antibodies of input or biotinylated proteins enriched with Streptavidin beads from HeLa_FGFR2b-APEXST treated with vehicle (UT) or treated with H₂O₂ or with FGF10 for 1, and 8 min. **f** Schematic of RE-localised FGFR2b, following 40 min of FGF10 treatment. Both RAB11-APEX2 and RAB25 localize at the recycling endosomes³⁷. **g** Immunoblot analysis ($N \geq 3$ independent biological replicates) with the indicated antibodies of input or biotinylated proteins enriched with Streptavidin beads from HeLa_FGFR2bST_RAB11-APEX2 stimulated with either H₂O₂ or with FGF10 for 0 and 40 mins. UT, treatment with vehicle as control. Source data are provided as a Source Data file.

Fig. 7. Phosphorylated ULK1 is recruited at the recycling endosomes. a, b, f, g, k Immunoblot analysis with the indicated antibodies of HeLa_FGFR2bST_RAB11-APEX2 (a), T47D transfected with RAB11-APEX2 (T47D_RAB11-APEX2) (b), HeLa_FGFR2bST (f) or T47D (g) transfected either with wtRAB11, DnRAB11, or DnDNM2 (f, g), T47D transfected with GFP or DnRAB11 (k) stimulated with FGF10 for the indicated time points. Non proximal and proximal samples represent the supernatant and the pull-down following enrichment of biotinylated samples with streptavidin beads, respectively, and run against total lysates (total) (a, b). N \geq 3 independent biological replicates. UT, treatment with vehicle as control. **c** Quantification of Proximity Ligation Assay (PLA) puncta between FGFR2b and S638 pULK1 in HeLa_FGFR2bST cells treated with vehicle (UT) or FGF10 for 40 mins.; *p*-value < 0.0005 *** (one-sided Students t-test) N = 9 independent biological replicates. **d** Representative confocal images of co-localization between FGFR2b-APEX2 (red) and phosphorylated ULK1 on S638 (blue) in T47D_FGFR2^{KO}_FGFR2b-APEXST transfected with RAB11 or GFP-DnRAB11 (green) and stimulated with FGF10 for 40 min. Scale bar, 5 μ m. Inset, zoomed images of the region indicated by the arrowheads (scale bar, 50 μ m). The white arrowhead indicates co-localization, and the pink arrowheads indicate lack of co-localization. **e, h, j, i** Quantification of the presence (pixel proportion) and of the co-localization (pixel overlap fraction) of the indicated proteins. Values represent median \pm SD from N=3 independent biological replicates where we analysed between 2 and 5 cells for each N; *** *p*-value<0.0005 (one-sided Student t-test). Representative images are shown in Fig. 7d (e), Supplementary Fig. 7f (h), Supplementary Fig. 7g (i), Supplementary Fig. 7h (j). Source data are provided as a Source Data file.

Supplementary Fig. 7. FGFR2b regulates mTOR and ULK1 signalling from the recycling endosomes. **a** Immunoblot analysis (N \geq 3 independent biological replicates) with the indicated antibodies of HeLa FGFR2b-APEX2ST (top) and T47D_FGFR2b^{KO}-FGFR2b-APEX2ST (bottom). Non proximal and proximal samples represent the supernatant and the pulldown following enrichment of biotinylated samples with streptavidin beads, respectively, and run against total lysates (total). UT, treatment with vehicle as control. **b** Representative confocal images corresponding to quantification in Fig. 7c, from Proximity Ligation Assay (PLA) between FGFR2b and S638 pULK1 (green) in T47D cells treated with FGF10 compared to vehicle (UT) Scale bar, 10 μ m. **c** Immunoblot analysis (N=3 independent biological replicates) with the indicated antibodies of T47D cells with siRNA-mediated knockdown of TTP or RCP, compared to siRNA control, treated with FGF10 for indicated time points **d** Immunoblot analysis (N=3 independent biological replicates) with the indicated antibodies of HeLa_FGFR2bST cells pre-treated with primaquine or Dynasore for 2 h followed by stimulation with FGF10 for the indicated time points. **e** Immunoblot analysis (N=3 independent biological replicates) with the indicated antibodies of T47D cells transfected with either GFP or DnRAB11 and stimulated with vehicle (UT) or FGF7 for the indicated time points. **f** Representative confocal images of the o-localization between FGFR2b (red) and TTP (blue) in T47D_FGFR2^{KO}_FGFR2b-APEXST transfected with wtRAB11 or DnRAB11 (green) and stimulated with vehicle (UT) or FGF10 for 40 min as indicated. Scale bar, 5 μ m. Zoomed images of the region indicated by the arrowheads are shown in the inset (scale bar, 50 μ m). The white arrowhead indicates co-localization, and the pink arrowheads indicate lack of co-localization. Representative confocal images of the co-localization between LAMP1 (red) and either TTP (blue) (**g**) or mTOR (blue) (**h**) in T47D_FGFR2^{KO}_FGFR2b-APEXST transfected with wtRAB11 or DnRAB11 (green) and treated with vehicle (UT) or with FGF10 for 40 min as indicated. Scale bar, 5 μ m. Zoomed images corresponding to the region indicated by the arrowheads are shown in the inset (scale bar, 50 μ m). The white arrowhead indicates co-localization, and the pink arrowheads indicate lack of co-localization. N=3 independent biological replicates. **i** Expression of indicated genes in HeLa FGFR2b or T47D transfected with wtRAB11, DnRAB11 or DnDNM2 or pre-incubated with rapamycin for 2 h followed by stimulation with FGF10 for 4 h. qPCR data are presented as heat map from N= 3 independent biological replicates. Source data are provided as a Source Data file.

Legends:

The experimental approach shown in the fig should be included in each legend. (ie: "Fig 1. Scanning confocal of x cells shows..." I don't think that information like "(EEA1) is a marker for EEs" should be in the fig legends (esp. incl. references). Put in main text and remove from here. Make consistent throughout.

We have changed this in all the appropriate figure legends.

White arrow heads indicate colocalisation: I suggest that you use a different colour or sign for lack of coloc ie. Arrowheads vs full arrows, or arrows filled with white vs filled with black

We have used white arrowheads to indicate co-localization and pink arrowheads to indicate lack of co-localization in Figures 1a, 3c, 7d, Supplementary Fig 7 f-h.

Confirm all scales of images are equal, otherwise put separate scale bars for each image (120' FGF10 in FGFR2b wtRAB11 and 120' UT FGFR2b/DnRAB11 same as rest of images?). May be useful to put scale bars on all separate images.

We confirm that the scale bar is the same for all images although certain nuclei look bigger. See also previous publications for similar images including Smith et al., EMBO J, 2021.

The authors draw conclusions on the activation of autophagy from acridine orange staining and LC3 I-II levels. The authors also use inhibitors for early autophagy (ULK1, ULK1/2, mTOR) but they do not use late inhibitors to visualise the build-up of cargo/LC3.

We thank the reviewer for his/her comment. We confirm that we mainly focused on early autophagy events based on the findings of the phosphoproteomics approach.

Fig 6. I presume wtRAB11 and DNRAB11 are GFP-tagged, thus the GFP lane represents these. Please mark it on the figure.

We have changed this in Figures 1a, 3c, 7d, Supplementary Fig 7 f-h.

The authors address the IF for Lamp/mTOR, but say this is in Fig 6i – there is no IF here. However that shown in Sup Fig 8h looks very good.

In the result section describing these results we referred both to the IF (Supplementary Fig 7h) and the quantification of the IF (Fig 7i).

In Fig legend for Supp Fig. 7, it is quoted: see Fig 7d inset table for inhibitor target. This is not there anymore

We have changed this.

Point 10: Thank you for strengthening the autophagy results. I appreciate that you have added so many more autophagy assays to the paper, and I appreciate that these are very time consuming. The key element in any autophagy experiment is to show flux, which is done by adding a late autophagy inhibitor (inhibition of lysosomal degradation to accumulate autophagy machinery (LC3II) and cargo) eg. BafA1, where you compare control and experimental conditions in the presence and absence of BafA1. This will distinguish whether FGF10 indeed suppresses autophagy or whether it increases autophagic flux. Autophagosome or lysosome presence alone can cause confusion, as an increase in flux could also cause a decrease in autophagosome marker (ie. LC3II), as the autophagosomes mature and are degraded at a higher rate than normal, while suppression would also cause a decrease in autophagosome marker as

autophagosomes are not formed. Including treatment with Baf clearly distinguishes the two:

during autophagy suppression in the presence of Baf, there will be very little autophagosome staining (LC3II) while during activation, there would be a massive accumulation of LC3II, in the presence of Baf. (See Fig 5C in Klionsky et al (al. al. al.) 2012 Autophagy <https://doi.org/10.4161/auto.19496>. Or Fig 5 in Thomé et al. JCS 2016 doi.org/10.1242/jcs.195057)

This is most simply demonstrated by WB for LC3II (and p62, but not essential) as you have shown on page 57 of the rebuttal. However you have not added Baf or a late stage inhibitor. The AO measurements shown on page 58 of the rebuttal indeed shows that in the presence of Baf, acidification of lysosomes is lost, and that this should be exacerbated in the case that FGF10 induced autophagy, but as it is equal in Baf + FGF10, it does indicate that FGF10 does not do so. You seem to have taken a long and more complicated way around a simple question, and I would be much happier to see a simple western with all conditions +/- Baf, but these results do support your conclusions.

We thank the reviewer for pointing out that our new experiments support the conclusion that FGF10 can regulate autophagy initiation from the recycling endosomes during FGFR2b recycling.